# Learning curves of generic features maps for realistic datasets with a teacher-student model

**Bruno Loureiro**
IdePHICS, EPFL, Lausanne

**Cédric Gerbelot**
Lab. de Physique de l'École Normale Supérieure, Paris

**Hugo Cui**
SPOC, EPFL, Lausanne

**Sebastian Goldt**
SISSA, Trieste

**Florent Krzakala**
IdePHICS, EPFL, Lausanne

**Marc Mézard**
École Normale Supérieure, Paris

**Lenka Zdeborová**
SPOC, EPFL, Lausanne

## Abstract

Teacher-student models provide a framework in which the typical-case performance of high-dimensional supervised learning can be described in closed form. The assumptions of Gaussian i.i.d. input data underlying the canonical teacher-student model may, however, be perceived as too restrictive to capture the behaviour of realistic data sets. In this paper, we introduce a Gaussian covariate generalisation of the model where the teacher and student can act on different spaces, generated with fixed, but generic feature maps. While still solvable in a closed form, this generalization is able to capture the learning curves for a broad range of realistic data sets, thus redeeming the potential of the teacher-student framework. Our contribution is then two-fold: first, we prove a rigorous formula for the asymptotic training loss and generalisation error. Second, we present a number of situations where the learning curve of the model captures the one of a *realistic data set* learned with kernel regression and classification, with out-of-the-box feature maps such as random projections or scattering transforms, or with pre-learned ones - such as the features learned by training multi-layer neural networks. We discuss both the power and the limitations of the framework.

## 1 Introduction

Teacher-student models are a popular framework to study the high-dimensional asymptotic performance of learning problems with synthetic data, and have been the subject of intense investigations spanning three decades [1–7]. In the wake of understanding the limitations of classical statistical learning approaches [8–10], this direction is witnessing a renewal of interest [10–15]. However, this framework is often assuming the input data to be Gaussian i.i.d., which is arguably too simplistic to be able to capture properties of realistic data. In this paper, we redeem this line of work by defining a Gaussian covariate model where the teacher and student act on different Gaussian correlated spaces with arbitrary covariance. We derive a rigorous asymptotic solution of this model generalizing the formulas found in the above mentioned classical works.

We then put forward a theory, supported by universality arguments and numerical experiments, that this model captures learning curves, i.e. the dependence of the training and test errors on the number of samples, for a generic class of feature maps applied to realistic datasets. These maps can be deterministic, random, or even learnt from the data. This analysis thus gives a unified framework to describe the learning curves of, for example, kernel regression and classification, the analysis of

35th Conference on Neural Information Processing Systems (NeurIPS 2021).

feature maps – random projections [16], neural tangent kernels [17], scattering transforms [18] – as well as the analysis of transfer learning performance on data generated by generative adversarial networks [19]. We also discuss limits of applicability of our results, by showing concrete situations where the learning curves of the Gaussian covariate model differ from the actual ones.

**Model definition —** The Gaussian covariate teacher-student model is defined via two vectors $\boldsymbol{u} \in \mathbb{R}^p$ and $\boldsymbol{v} \in \mathbb{R}^d$, with correlation matrices $\Psi \in \mathbb{R}^{p \times p}, \Omega \in \mathbb{R}^{d \times d}$ and $\Phi \in \mathbb{R}^{p \times d}$, from which we draw $n$ independent samples:

$$\begin{bmatrix} \boldsymbol{u}^\mu \\ \boldsymbol{v}^\mu \end{bmatrix} \in \mathbb{R}^{p+d} \underset{\text{i.i.d.}}{\sim} \mathcal{N} \left( 0, \begin{bmatrix} \Psi & \Phi \\ \Phi^\top & \Omega \end{bmatrix} \right), \qquad \mu = 1, \cdots, n. \tag{1}$$

The *labels* $y^\mu$ are generated by a **teacher** function that is only using the vectors $\boldsymbol{u}^\mu$:

$$y^\mu = f_0 \left( \frac{1}{\sqrt{p}} \boldsymbol{\theta}_0^\top \boldsymbol{u}^\mu \right), \tag{2}$$

where $f_0 : \mathbb{R} \to \mathbb{R}$ is a function that may include randomness such as, for instance, an additive Gaussian noise, and $\boldsymbol{\theta}_0 \in \mathbb{R}^p$ is a vector of teacher-weights with finite norm which can be either random or deterministic. Learning is performed by the **student** with weights $\boldsymbol{w}$ via empirical risk minimization that has access only to the features $\boldsymbol{v}^\mu$:

$$\hat{\boldsymbol{w}} = \arg\min_{\boldsymbol{w} \in \mathbb{R}^d} \left[ \sum_{\mu=1}^n g \left( \frac{\boldsymbol{w}^\top \boldsymbol{v}^\mu}{\sqrt{d}}, y^\mu \right) + r(\boldsymbol{w}) \right], \tag{3}$$

where $r$ and $g$ are proper, convex, lower-semicontinuous functions of $\boldsymbol{w} \in \mathbb{R}^d$ (e.g. $g$ can be a logistic or a square loss and $r$ a $\ell_p$ $(p=1,2)$ regularization). The key quantities we want to compute in this model are the *averaged training and generalisation errors* for the estimator $\boldsymbol{w}$,

$$\mathcal{E}_{\text{train.}}(\boldsymbol{w}) \equiv \frac{1}{n} \sum_{\mu=1}^n g \left( \frac{\boldsymbol{w}^\top \boldsymbol{v}^\mu}{\sqrt{d}}, y^\mu \right) \quad \text{and} \quad \mathcal{E}_{\text{gen.}}(\boldsymbol{w}) \equiv \mathbb{E} \left[ \hat{g} \left( \hat{f} \left( \frac{\boldsymbol{v}_{\text{new}}^\top \boldsymbol{w}}{\sqrt{d}} \right), f_0 \left( \frac{\boldsymbol{u}_{\text{new}}^\top \boldsymbol{\theta}_0}{\sqrt{p}} \right) \right) \right]. \tag{4}$$

where $g$ is the loss function in eq. (3), $\hat{f}$ is a prediction function (e.g. $\hat{f} = \text{sign}$ for a classification task), $\hat{g}$ is a performance measure (e.g. $\hat{g}(\hat{y}, y) = (\hat{y} - y)^2$ for regression or $\hat{g}(\hat{y}, y) = \mathbb{P}(\hat{y} \neq y)$ for classification) and $(\boldsymbol{u}_{\text{new}}, \boldsymbol{v}_{\text{new}})$ is a fresh sample from the joint distribution of $\boldsymbol{u}$ and $\boldsymbol{v}$.

Our two **main technical contributions** are:

(C1) In Theorems 1 & 2, we give a rigorous closed-form characterisation of the properties of the estimator $\hat{\boldsymbol{w}}$ for the Gaussian covariate model (1), and the corresponding training and generalisation errors in the high-dimensional limit. We prove our result using Gaussian comparison inequalities [20]; (C2) We show how the same expression can be obtained using the replica method from statistical physics [21]. This is of additional interest given the wide range of applications of the replica approach in machine learning and computer science [22]. In particular, this allows to put on a rigorous basis many results previously derived with the replica method.

**Towards realistic data —** In the second part of our paper, we argue that the above Gaussian covariate model (1) is generic enough to capture the learning behaviour of a broad range of realistic data. Let $\{\boldsymbol{x}^\mu\}_{\mu=1}^n$ denote a data set with $n$ independent samples on $\mathcal{X} \subset \mathbb{R}^D$. Based on this input, the **features** $\boldsymbol{u}, \boldsymbol{v}$ are given by (potentially) elaborated transformations of $\boldsymbol{x}$, i.e.

$$\boldsymbol{u} = \boldsymbol{\varphi}_t(\boldsymbol{x}) \in \mathbb{R}^p \quad \text{and} \quad \boldsymbol{v} = \boldsymbol{\varphi}_s(\boldsymbol{x}) \in \mathbb{R}^d \tag{5}$$

for given centred feature maps $\boldsymbol{\varphi}_t : \mathcal{X} \to \mathbb{R}^p$ and $\boldsymbol{\varphi}_s : \mathcal{X} \to \mathbb{R}^d$, see Fig. 1. Uncentered features can be taken into account by shifting the covariances, but we focus on the centred case to lighten notation.

The Gaussian covariate model (1) is exact in the case where $\boldsymbol{x}$ are Gaussian variables and the feature maps $(\boldsymbol{\varphi}_s, \boldsymbol{\varphi}_s)$ preserve the Gaussianity, for example linear features. In particular, this is the case for $\boldsymbol{u} = \boldsymbol{v} = \boldsymbol{x}$, which is the widely-studied vanilla teacher-student model [24]. The interest of the model (1) is that it also captures a range of cases in which the feature maps $\boldsymbol{\varphi}_t$ and $\boldsymbol{\varphi}_s$ are deterministic, or even learnt from the data. The covariance matrices $\Psi$, $\Phi$, and $\Omega$ then represent different aspects of the data-generative process and learning model. The student (3) then corresponds to the last layer of the learning model. These observation can be distilled into the following conjecture:

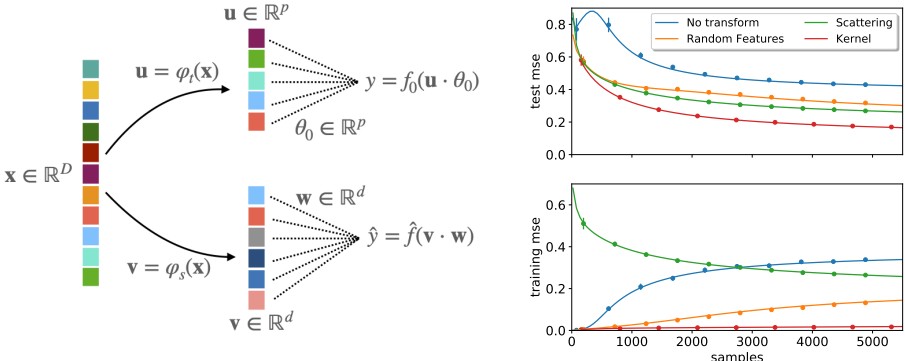

Figure 1: **Left:** Given a data set $\{\boldsymbol{x}^\mu\}_{\mu=1}^n$, teacher $\boldsymbol{u} = \boldsymbol{\varphi}_t(\boldsymbol{x})$ and student maps $\boldsymbol{v} = \boldsymbol{\varphi}_t(\boldsymbol{x})$, we assume $[\boldsymbol{u}, \boldsymbol{v}]$ to be jointly Gaussian random variables and apply the results of the Gaussian covariate model (1). **Right:** Illustration on real data, here ridge regression on even vs odd MNIST digits, with regularisation $\lambda = 10^{-2}$. Full line is theory, points are simulations. We show the performance with no feature map (blue), random feature map with $\sigma = \mathrm{erf}$ & Gaussian projection (orange), the scattering transform with parameters $J = 3, L = 8$ [18] (green), and of the limiting kernel of the random map [23] (red). The covariance $\Omega$ is empirically estimated from the full data set, while the other quantities appearing in the Theorem 1 are expressed directly as a function of the labels, see Section 3.4. Simulations are averaged over 10 independent runs.

**Conjecture 1.** *(Gaussian equivalent model) For a wide class of data distributions $\{\boldsymbol{x}^\mu\}_{\mu=1}^n$, and features maps $\boldsymbol{u} = \boldsymbol{\varphi}_t(\boldsymbol{x}), \boldsymbol{v} = \boldsymbol{\varphi}_s(\boldsymbol{x})$, the generalisation and training errors of estimator (3) are asymptotically captured by the equivalent Gaussian model (1), where $[\boldsymbol{u}, \boldsymbol{v}]$ are jointly Gaussian variables, and thus by the closed-form expressions of Theorem 1.*

The second part of our **main contributions** are:

(C3) In Sec. 3.3 we show that the theoretical predictions from (C1) captures the learning curves in non-trivial cases, e.g. when input data are generated using a trained generative adversarial network, while extracting both the feature maps from a neural network trained on real data.

(C4) In Sec. 3.4, we show empirically that for ridge regression the asymptotic formula of Theorem 1 can be applied *directly* to real data sets, even though the Gaussian hypothesis is not satisfied. This universality-like property is a consequence of Theorem 3 and is illustrated in Fig. 1 (right) where the real learning curve of several features maps learning the odd-versus-even digit task on MNIST is compared to the theoretical prediction.

**Related work —** Rigorous results for teacher-student models: The Gaussian covariate model (1) contains the vanilla teacher-student model as a special case where one takes $\boldsymbol{u}$ and $\boldsymbol{v}$ *identical*, with unique covariance matrix $\Omega$. This special case has been extensively studied in the statistical physics community using the heuristic replica method [1–3, 24, 25]. Many recent rigorous results for such models can be rederived as a special case of our formula, e.g. refs. [10–15, 26–29]. Numerous of these results are based on the same proof technique as we employed here: the Gordon's Gaussian min-max inequalities [20, 30, 31]. The asymptotic analysis of kernel ridge regression [32], of margin-based classification [33] also follow from our theorem. See also Appendix A.6 for the details on these connections. Other examples include models of the double descent phenomenon [34]. Closer to our work is the recent work of [35] on the random feature model. For ridge regression, there are also precise predictions thanks to random matrix theory [12, 36–41]. A related set of results was obtained in [42] for orthogonal random matrix models. The main technical novelty of our proof is the handling of a generic loss and regularisation, not only ridge, representing convex empirical risk minimization, for both classification and regression, with the generic correlation structure of the model (1).

Gaussian equivalence: A similar Gaussian conjecture has been discussed in a series of recent works, and some authors proved partial results in this direction [11, 12, 28, 35, 43–46]. Ref. [45] analyses a special case of the Gaussian model (corresponding to $\boldsymbol{\varphi}_t = \mathrm{id}$ here), and proves a Gaussian equivalence theorem (GET) for feature maps $\boldsymbol{\varphi}_s$ given by single-layer neural networks with fixed weights. They also show that for Gaussian data $\boldsymbol{x} \sim \mathcal{N}(\boldsymbol{0}, \mathrm{I}_D)$, feature maps of the form $\boldsymbol{v} = \sigma(\mathbf{W}\boldsymbol{x})$ (with some technical restriction on the weights) led to the jointly-Gaussian property for the two scalars $(\boldsymbol{v} \cdot \boldsymbol{w}, \boldsymbol{u} \cdot \boldsymbol{\theta}_0)$ for *almost* any vector $\boldsymbol{w}$. However, their stringent assumptions on random teacher weights limited the scope of applications to unrealistic label models. A related line of work

discussed similar universality through the lens of random matrix theory [47–49]. In particular, Seddik et al. [50] showed that, in our notations, vectors $[\boldsymbol{u}, \boldsymbol{v}]$ obtained from Gaussian inputs $\boldsymbol{x} \sim \mathcal{N}(\mathbf{0}, I_D)$ with Lipschitz feature maps satisfy a concentration property. In this case, again, one can expect the two scalars $(\boldsymbol{v} \cdot \boldsymbol{w}, \boldsymbol{u} \cdot \boldsymbol{\theta}_0)$ to be jointly Gaussian with high-probability on $\boldsymbol{w}$. Remarkably, in the case of random feature maps, [46] could go beyond this central-limit-like behavior and established the universality of the Gaussian covariate model (1) for the actual learned weights $\hat{\boldsymbol{w}}$.

## 2 Main technical results

Our main technical result is a closed-form expression for the asymptotic training and generalisation errors (4) of the Gaussian covariate model introduced above. We start by presenting our result in the most relevant setting for the applications of interest in Section 3, which is the case of the $\ell_2$ regularization. Next, we briefly present our result in larger generality, which includes non-asymptotic results for non-separable losses and regularizations.

We start by defining key quantities that we will use to characterize the estimator $\hat{\boldsymbol{w}}$. Let $\Omega = S^\top \text{diag}(\omega_i) S$ be the spectral decomposition of $\Omega$. Let:

$$\rho \equiv \frac{1}{d}\boldsymbol{\theta}_0^\top \Psi \boldsymbol{\theta}_0 \in \mathbb{R}, \qquad\qquad \bar{\boldsymbol{\theta}} \equiv \frac{S\Phi^\top \boldsymbol{\theta}_0}{\sqrt{\rho}} \in \mathbb{R}^d \qquad (6)$$

and define the joint empirical density $\hat{\mu}_d$ between $(\omega_i, \bar{\theta}_i)$:

$$\hat{\mu}_d(\omega, \bar{\theta}) \equiv \frac{1}{d}\sum_{i=1}^d \delta(\omega - \omega_i)\delta(\bar{\theta} - \bar{\theta}_i). \qquad (7)$$

Note that $\Phi^\top \boldsymbol{\theta}_0$ is the projection of the teacher weights on the student space, and therefore $\bar{\boldsymbol{\theta}}$ is the rotated projection on the basis of the student covariance, rescaled by the teacher variance. Together with the student eigenvalues $\omega_i$, these are relevant statistics of the model, encoded here in the joint distribution $\hat{\mu}_d$.

**Assumptions —** Consider the *high-dimensional* limit in which the number of samples $n$ and the dimensions $p, d$ go to infinity with fixed ratios:

$$\alpha \equiv \frac{n}{d}, \text{ and } \gamma \equiv \frac{p}{d}. \qquad (8)$$

Assume that the covariance matrices $\Psi, \Omega$ are positive-definite and that the Schur complement of the block covariance in equation (1) is positive semi-definite. Additionally, the spectral distributions of the matrices $\Phi, \Psi$ and $\Omega$ converge to distributions such that the limiting joint distribution $\mu$ is well-defined, and their maximum singular values are bounded with high probability as $n, p, d \to \infty$. Finally, regularity assumptions are made on the loss and regularization functions mainly to ensure feasibility of the minimization problem. We assume that the cost function $r + g$ is coercive, i.e. $\lim_{\|\boldsymbol{w}\|_2 \to +\infty}(r + g)(\boldsymbol{w}) = +\infty$ and that the following scaling condition holds : for all $n, d \in \mathbb{N}, \boldsymbol{z} \in \mathbb{R}^n$ and any constant $c > 0$, there exist a finite, positive constant $C$, such that, for any standard normal random vectors $\boldsymbol{h} \in \mathbb{R}^d$ and $\boldsymbol{g} \in \mathbb{R}^n$:

$$\|\boldsymbol{z}\|_2 \leqslant c\sqrt{n} \implies \sup_{\boldsymbol{x}\in\partial g(\boldsymbol{z})} \|\boldsymbol{x}\|_2 \leqslant C\sqrt{n}, \quad \frac{1}{d}\mathbb{E}\left[r(\boldsymbol{h})\right] < +\infty, \quad \frac{1}{n}\mathbb{E}\left[g(\boldsymbol{g})\right] < +\infty \qquad (9)$$

The relevance of these assumptions in a supervised machine learning context is discussed in Appendix B.1. We are now in a position to state our result.

**Theorem 1.** *(Closed-form asymptotics for $\ell_2$ regularization) In the asymptotic limit defined above, the training and generalisation errors (4) of the estimator $\hat{\boldsymbol{w}} \in \mathbb{R}^d$ solving the empirical risk minimisation problem in eq. (3) with $\ell_2$ regularization $r(\boldsymbol{w}) = \frac{\lambda}{2}||\boldsymbol{w}||_2^2$ verify:*

$$\mathcal{E}_{\text{train.}}(\hat{\boldsymbol{w}}) \xrightarrow[d\to\infty]{P} \mathbb{E}_{s,h\sim\mathcal{N}(0,1)}\left[g\left(prox_{V^\star g(.,f_0(\sqrt{\rho}s))}\left(\frac{m^\star}{\sqrt{\rho}}s + \sqrt{q^\star - \frac{m^{\star 2}}{\rho}}h\right), f_0(\sqrt{\rho}s)\right)\right]$$

$$\mathcal{E}_{\text{gen.}}(\hat{\boldsymbol{w}}) \xrightarrow[d\to\infty]{P} \mathbb{E}_{(\nu,\lambda)}\left[\hat{g}\left(\hat{f}(\lambda), f_0(\nu)\right)\right] \qquad (10)$$

*where prox stands for the proximal operator defined as*

$$prox_{Vg(.,y)}(x) = \arg\min_z \{g(z,y) + \frac{1}{2V}(x-z)^2\} \tag{11}$$

*and where $(\nu, \lambda)$ are jointly Gaussian scalar variables:*

$$(\nu, \lambda) \sim \mathcal{N}\left(0, \begin{bmatrix} \rho & m^\star \\ m^\star & q^\star \end{bmatrix}\right), \tag{12}$$

*and the overlap parameters $(V^\star, q^\star, m^\star)$ are prescribed by the unique fixed point of the following set of self-consistent equations:*

$$
\begin{cases}
V = \mathbb{E}_{(\omega, \bar{\theta}) \sim \mu}\left[\frac{\omega}{\lambda + \hat{V}\omega}\right] \\
m = \frac{\hat{m}}{\sqrt{\gamma}}\mathbb{E}_{(\omega, \bar{\theta}) \sim \mu}\left[\frac{\bar{\theta}^2}{\lambda + \hat{V}\omega}\right] \\
q = \mathbb{E}_{(\omega, \bar{\theta}) \sim \mu}\left[\frac{\hat{m}^2\bar{\theta}^2\omega + \hat{q}\omega^2}{(\lambda + \hat{V}\omega)^2}\right]
\end{cases}
,
\begin{cases}
\hat{V} = \frac{\alpha}{V}(1 - \mathbb{E}_{s,h \sim \mathcal{N}(0,1)}[f'_g(V,m,q)]) \\
\hat{m} = \frac{1}{\sqrt{\rho\gamma}}\frac{\alpha}{V}\mathbb{E}_{s,h \sim \mathcal{N}(0,1)}\left[sf_g(V,m,q) - \frac{m}{\sqrt{\rho}}f'_g(V,m,q)\right] \\
\hat{q} = \frac{\alpha}{V^2}\mathbb{E}_{s,h \sim \mathcal{N}(0,1)}\left[\left(\frac{m}{\sqrt{\rho}}s + \sqrt{q - \frac{m^2}{\rho}}h - f_g(V,m,q)\right)^2\right]
\end{cases}
\tag{13}
$$

*where we defined the scalar random functions $f_g(V,m,q) = prox_{Vg(.,f_0(\sqrt{\rho}s))}(\rho^{-1/2}ms + \sqrt{q - \rho^{-1}m^2}h)$ and $f'_g(V,m,h) = prox'_{Vg(.,f_0(\sqrt{\rho}s))}(\rho^{-1/2}ms + \sqrt{q - \rho^{-1}m^2}h)$ as the first derivative of the proximal operator.*

*Proof*: This result is a consequence of Theorem 2, whose proof can be found in appendix B.

The parameters of the model $(\boldsymbol{\theta}_0, \Omega, \Phi, \Psi)$ only appear trough $\rho$, eq. (6), and the asymptotic limit $\mu$ of the joint distribution eq. (7) and $(f_0, \hat{f}, g, \lambda)$. One can easily iterate the above equations to find their fixed point, and extract $(q^*, m^*)$ which appear in the expressions for the training and generalisation errors $(\mathcal{E}^\star_{\text{train}}, \mathcal{E}^\star_{\text{gen}})$, see eq. (4). Note that $(q^*, m^*)$ have an intuitive interpretation in terms of the estimator $\hat{\boldsymbol{w}} \in \mathbb{R}^d$:

$$q^\star \equiv \frac{1}{d}\hat{\boldsymbol{w}}^\top \Omega \hat{\boldsymbol{w}}, \qquad\qquad m^\star \equiv \frac{1}{\sqrt{dp}}\boldsymbol{\theta}_0^\top \Phi \hat{\boldsymbol{w}} \tag{14}$$

Or in words: $m^\star$ is the correlation between the estimator projected in the teacher space, while $q^\star$ is the reweighted norm of the estimator by the covariance $\Omega$. The parameter $V^*$ also has a concrete interpretation : it parametrizes the deformation that must be applied to a Gaussian field specified by the solution of the fixed point equations to obtain the asymptotic behaviour of $\hat{\mathbf{z}}$. It prescribes the degree of non-linearity given to the linear output by the chosen loss function. This is coherent with the robust regression viewpoint, where one introduces non-square losses to deal with the potential non-linearity of the generative model. $\hat{V}^*$ plays a similar role for the estimator $\hat{\mathbf{w}}$ through the proximal operator of the regularisation, see Theorem 4 and 5 in the Appendix. Two cases are of particular relevance for the experiments that follow. The first is the case of *ridge regression*, in which $f_0(x) = \hat{f}(x)$ and both the loss $g$ and the performance measure $\hat{g}$ are taken to be the *mean-squared error* $\text{mse}(y, \hat{y}) = \frac{1}{2}(y - \hat{y})^2$, and the asymptotic errors are given by the simple closed-form expression:

$$\mathcal{E}^\star_{\text{gen}} = \rho + q^\star - 2m^\star, \qquad\qquad \mathcal{E}^\star_{\text{train}} = \frac{\mathcal{E}^\star_{\text{gen}}}{(1 + V^\star)^2}, \tag{15}$$

The second case of interest is the one of a binary classification task, for which $f_0(x) = \hat{f}(x) = \text{sign}(x)$, and we choose the performance measure to be the *classification error* $\hat{g}(y, \hat{y}) = \mathbb{P}(y \neq \hat{y})$. In the same notation as before, the asymptotic generalisation error in this case reads:

$$\mathcal{E}^\star_{\text{gen}} = \frac{1}{\pi}\cos^{-1}\left(\frac{m^\star}{\sqrt{\rho q^\star}}\right), \tag{16}$$

while the training error $\mathcal{E}^\star_{\text{train}}$ depends on the choice of $g$ - which we will take to be the logistic loss $g(y, x) = \log(1 + e^{-xy})$ in all of the binary classification experiments.

As mentioned above, this paper includes stronger technical results including finite size corrections and precise characterization of the distribution of the estimator $\hat{\boldsymbol{w}}$, for generic, non-separable loss and

regularization $g$ and $r$. This type of distributional statement is encountered for special cases of the model in related works such as [28, 29, 51]. Define $\mathcal{V} \in \mathbb{R}^{n \times d}$ as the matrix of concatenated samples used by the student. Informally, in high-dimension, the estimator $\hat{w}$ and $\hat{z} = \frac{1}{\sqrt{d}} \mathcal{V} \hat{w}$ roughly behave as non-linear transforms of Gaussian random variables centered around the teacher vector $\boldsymbol{\theta}_0$ (or its projection on the covariance spaces) as follows:

$$\boldsymbol{w}^* = \Omega^{-1/2} \underset{\frac{1}{\hat{V}^*} r(\Omega^{-1/2}.)}{\text{prox}} \left( \frac{1}{\hat{V}^*} (\hat{m}^* \boldsymbol{t} + \sqrt{\hat{q}^*} \boldsymbol{g}) \right), \boldsymbol{z}^* = \underset{V^* g(.,\boldsymbol{z})}{\text{prox}} \left( \frac{m^*}{\sqrt{\rho}} \boldsymbol{s} + \sqrt{q^* - \frac{(m^*)^2}{\rho}} \boldsymbol{h} \right).$$

where $\boldsymbol{s}, \boldsymbol{h} \sim \mathcal{N}(0, \mathrm{I}_n)$ and $\boldsymbol{g} \sim \mathcal{N}(0, \mathrm{I}_d)$ are random vectors independent of the other quantities, $\boldsymbol{t} = \Omega^{-1/2} \Phi^\top \boldsymbol{\theta}_0$, $\boldsymbol{y} = \boldsymbol{f}_0 \left( \sqrt{\rho} \boldsymbol{s} \right)$, and $(V^*, \hat{V}^*, q^*, \hat{q}^*, m^*, \hat{m}^*)$ is the unique solution to the fixed point equations presented in Lemma 12 of appendix B. Those fixed point equations are the generalization of (13) to generic, non-separable loss function and regularization. The formal concentration of measure result can then be stated in the following way:

**Theorem 2.** *(Non-asymptotic version, generic loss and regularization) Under Assumption (B.1), consider any optimal solution $\hat{w}$ to 3. Then, there exist constants $C, c, c' > 0$ such that, for any Lipschitz function $\phi_1 : \mathbb{R}^d \to \mathbb{R}$, and separable, pseudo-Lipschitz function $\phi_2 : \mathbb{R}^n \to \mathbb{R}$ and any $0 < \epsilon < c'$:*

$$\mathbb{P}\left( \left| \phi_1 \left( \frac{\hat{w}}{\sqrt{d}} \right) - \mathbb{E}\phi_1 \left( \frac{\boldsymbol{w}^*}{\sqrt{d}} \right) \right| \geqslant \epsilon \right) \leqslant \frac{C}{\epsilon^2} e^{-cn\epsilon^4}, \mathbb{P}\left( \left| \phi_2 \left( \frac{\hat{z}}{\sqrt{n}} \right) - \mathbb{E}\phi_2 \left( \frac{\boldsymbol{z}^*}{\sqrt{n}} \right) \right| \geqslant \epsilon \right) \leqslant \frac{C}{\epsilon^2} e^{-cn\epsilon^4}.$$

Note that in this form, the dimensions $n, p, d$ still appear explicitly, as we are characterizing the convergence of the estimator's distribution for large but finite dimension. The clearer, one-dimensional statements are recovered by taking the $n, p, d \to \infty$ limit with separable functions and an $\ell_2$ regularization. Other simplified formulas can also be obtained from our general result in the case of an $\ell_1$ penalty, but since this breaks rotational invariance, they do look more involved than the $\ell_2$ case. From Theorem 2, one can deduce the expressions of a number of observables, represented by the test functions $\phi_1, \phi_2$, characterizing the performance of $\hat{w}$, for instance the training and generalization error. A more detailed statement, along with the proof, is given in appendix B.

## 3   Applications of the Gaussian model

We now discuss how the theorems above are applied to characterise the learning curves for a range of concrete cases. We present a number of cases – some rather surprising – for which Conjecture 1 seems valid, and point out some where it is not. An out-of-the-box iterator for all the cases studied hereafter is provided in the GitHub repository for this manuscript at `https://github.com/IdePHICS/GCMProject`.

### 3.1   Random kitchen sink with Gaussian data

If we choose random feature maps $\boldsymbol{\varphi}_s(\boldsymbol{x}) = \sigma(\mathrm{F}\boldsymbol{x})$ for a random matrix F and a chosen scalar function $\sigma$ acting component-wise, we obtain the random kitchen sink model [16]. This model has seen a surge of interest recently, and a sharp asymptotic analysis was provided in the particular case of uncorrelated Gaussian data $\boldsymbol{x} \sim \mathcal{N}(\boldsymbol{0}, \mathrm{I}_D)$ and $\boldsymbol{\varphi}_t(\boldsymbol{x}) = \boldsymbol{x}$ in [11, 12] for ridge regression and generalised by [43, 46] for generic convex losses. Both results can be framed as a Gaussian covariate model with:

$$\Psi = \mathrm{I}_p, \qquad \Phi = \kappa_1 \mathrm{F}^\top, \qquad \Omega = \kappa_0^2 \mathbf{1}_d \mathbf{1}_d^\top + \kappa_1^2 \frac{\mathrm{F}\mathrm{F}^\top}{d} + \kappa_\star^2 \mathrm{I}_d, \qquad (17)$$

where $\mathbf{1}_d \in \mathbb{R}^d$ is the all-one vector and the constants $(\kappa_0, \kappa_1, \kappa_\star)$ are related to the non-linearity $\sigma$:

$$\kappa_0 = \mathbb{E}_{z \sim \mathcal{N}(0,1)} [\sigma(z)], \quad \kappa_1 = \mathbb{E}_{z \sim \mathcal{N}(0,1)} [z\sigma(z)], \quad \kappa_\star = \sqrt{\mathbb{E}_{z \sim \mathcal{N}(0,1)} [\sigma(z)^2] - \kappa_0^2 - \kappa_1^2}. \quad (18)$$

In this case, the averages over $\mu$ in eq. (13) can be directly expressed in terms of the Stieltjes transform associated with the spectral density of $\mathrm{F}\mathrm{F}^\top$. Note, however, that our present framework can accommodate more involved random sinks models, such as when the teacher features are also a random feature model or multi-layer random architectures.

### 3.2   Kernel methods with Gaussian data

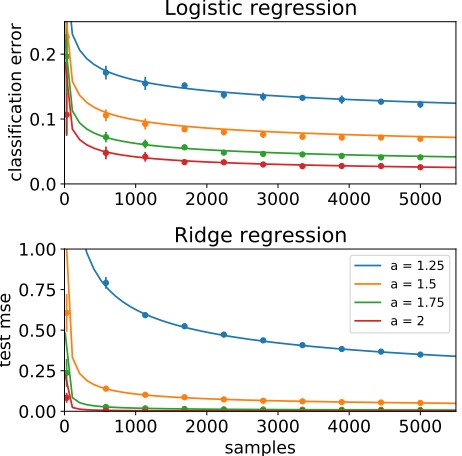

**Figure 2:** Learning in kernel space: Teacher and student live in the same (Hilbert) feature space $\boldsymbol{v} = \boldsymbol{u} \in \mathbb{R}^d$ with $d \gg n$, and the performance only depends on the relative decay between the student spectrum $\omega_i = d\, i^{-2}$ (the capacity) and the teacher weights in feature space $\theta_{0i}^2 \omega_i = d\, i^{-a}$ (the source). Top: a task with sign teacher (in kernel space), fitted with a max-margin support vector machine (logistic regression with vanishing regularisation [52]). Bottom: a task with linear teacher (in kernel space) fitted via kernel ridge regression with vanishing regularisation. Points are simulation that matches the theory (lines). Simulations are averaged over 10 independent runs.

Another direct application of our formalism is to kernel methods. Kernel methods admit a dual representation in terms of optimization over feature space [53]. The connection is given by Mercer's theorem, which provides an eigen-decomposition of the kernel and of the target function in the feature basis, effectively mapping kernel regression to a teacher-student problem on feature space. The classical way of studying the performance of kernel methods [54, 55] is then to directly analyse the performance of convex learning in this space. In our notation, the teacher and student feature maps are equal, and we thus set $p = d, \Psi = \Phi = \Omega = \mathrm{diag}(\omega_i)$ where $\omega_i$ are the eigenvalues of the kernel and we take the teacher weights $\boldsymbol{\theta}_0$ to be the decomposition of the target function in the kernel feature basis.

There are many results in classical learning theory on this problem for the case of ridge regression (where the teacher is usually called "the source" and the eigenvalues of the kernel matrix the "capacity", see e.g. [54, 56]). However, these are worst case approaches, where no assumption is made on the true distribution of the data. In contrast, here we follow a *typical case* analysis, assuming Gaussianity in feature space. Through Theorem 1, this allows us to go beyond the restriction of the ridge loss. An example for logistic loss is in Fig. 2.

For the particular case of kernel ridge regression, Th. 1 provides a rigorous proof of the formula conjectured in [32]. App. A.6 presents an explicit mapping to their results. Hard-margin Support Vector Machines (SVMs) have also been studied using the heuristic replica method from statistical physics in [57, 58]. In our framework, this corresponds to the *hinge loss* $g(x, y) = \max(0, 1 - yx)$ when $\lambda \to 0^+$. Our theorem thus puts also these works on rigorous grounds, and extends them to more general losses and regularization.

### 3.3 GAN-generated data and learned teachers

To approach more realistic data sets, we now consider the case in which the input data $\boldsymbol{x} \in \mathcal{X}$ is given by a generative neural network $\boldsymbol{x} = \mathcal{G}(\boldsymbol{z})$, where $\boldsymbol{z}$ is a Gaussian i.i.d. latent vector. Therefore, the covariates $[\boldsymbol{u}, \boldsymbol{v}]$ are the result of the following Markov chain:

$$\boldsymbol{z} \underset{\mathcal{G}}{\mapsto} \boldsymbol{x} \in \mathcal{X} \underset{\boldsymbol{\varphi}_t}{\mapsto} \boldsymbol{u} \in \mathbb{R}^p, \qquad\qquad \boldsymbol{z} \underset{\mathcal{G}}{\mapsto} \boldsymbol{x} \in \mathcal{X} \underset{\boldsymbol{\varphi}_s}{\mapsto} \boldsymbol{v} \in \mathbb{R}^d. \qquad (19)$$

With a model for the covariates, the missing ingredient is the teacher weights $\boldsymbol{\theta}_0 \in \mathbb{R}^p$, which determine the label assignment: $y = f_0(\boldsymbol{u}^\top \boldsymbol{\theta}_0)$. In the experiments that follow, we fit the teacher weights *from the original data set in which the generative model $\mathcal{G}$ was trained*. Different choices for the fitting yield different teacher weights, and the quality of label assignment can be accessed by the performance of the fit on the test set. The set $(\boldsymbol{\varphi}_t, \boldsymbol{\varphi}_s, \mathcal{G}, \boldsymbol{\theta}_0)$ defines the data generative process. For predicting the learning curves from the iterative eqs. (13) we need to sample from the spectral measure $\mu$, which amounts to estimating the *population* covariances $(\Psi, \Phi, \Omega)$. This is done from the generative process in eq. (19) with a Monte Carlo sampling algorithm. This pipeline is explained in detail in Appendix D. An open source implementation of the algorithms used in the experiments is available online at `https://github.com/IdePHICS/GCMProject`.

Fig. 3 shows an example of the learning curves resulting from the pipeline discussed above in a logistic regression task on data generated by a GAN trained on CIFAR10 images. More concretely,

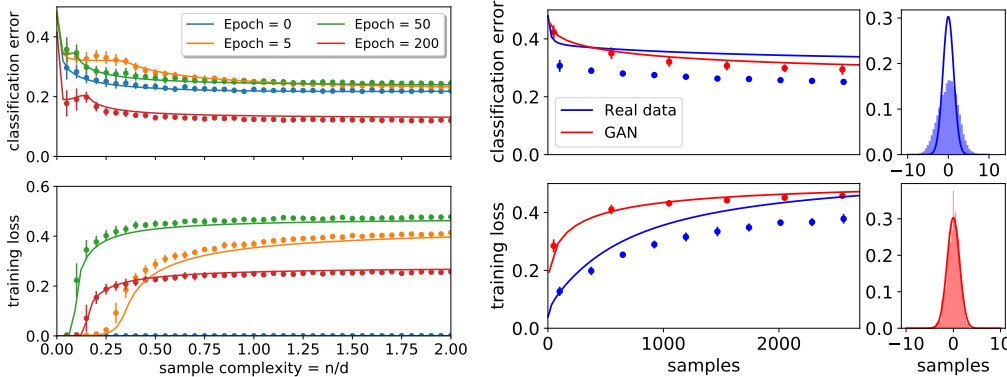

Figure 3: **Left:** generalisation classification error (top) and (unregularised) training loss (bottom) vs the sample complexity $\alpha = n/d$ for logistic regression on a learned feature map trained on dcGAN-generated CIFAR10-like images labelled by a teacher fully-connected neural network (see Appendix D.1 for architecture details), with vanishing $\ell_2$ regularisation. The different curves compare featured maps at different epochs of training. The theoretical predictions based on the Gaussian covariate model (full lines) are in very good agreement with the actual performance (points). **Right:** Test classification error (top) and (unregularised) training loss, (bottom) for logistic regression as a function of the number of samples $n$ for an animal vs not-animal binary classification task with $\ell_2$ regularization $\lambda = 10^{-2}$, comparing real CIFAR10 grey-scale images (blue) with dcGAN-generated CIFAR10-like gray-scale images (red). The real-data learning curve was estimated, just as in Figs. 4 from the population covariances on the full data set, and it is not in agreement with the theory in this case. On the very right we depict the histograms of the variable $\frac{1}{\sqrt{d}} v^\top \hat{w}$ for a fixed number of samples $n = 2d = 2048$ and the respective theoretical predictions (solid line). Simulations are averaged over 10 independent runs.

we used a pre-trained five-layer deep convolutional GAN (dcGAN) from [59], which maps $100$ dimensional i.i.d. Gaussian noise into $k = 32 \times 32 \times 3$ realistic looking CIFAR10-like images: $\mathcal{G} : z \in \mathbb{R}^{100} \mapsto x \in \mathbb{R}^{32 \times 32 \times 3}$. To generate labels, we trained a simple fully-connected four-layer neural network on the *real* CIFAR10 data set, on a odd ($y = +1$) vs. even ($y = -1$) task, achieving $\sim 75\%$ classification accuracy on the test set. The teacher weights $\theta_0 \in \mathbb{R}^p$ were taken from the last layer of the network, and the teacher feature map $\varphi_t$ from the three previous layers. For the student model, we trained a completely independent fully connected 3-layer neural network on the dcGAN-generated CIFAR10-like images and took snapshots of the feature maps $\varphi_s^i$ induced by the 2-first layers during the first $i \in \{0, 5, 50, 200\}$ epochs of training. Finally, once $(\mathcal{G}, \varphi_t, \varphi_s^i, \theta_0)$ have been fixed, we estimated the covariances $(\Psi, \Phi, \Omega)$ with a Monte Carlo algorithm. Details of the architectures used and of the training procedure can be found in Appendix. D.1.

Fig. 3 depicts the resulting learning curves obtained by training the last layer of the student. Interestingly, the performance of the feature map at epoch 0 (random initialisation) beats the performance of the learned features during early phases of training in this experiment. Another interesting behaviour is given by the separability threshold of the learned features, i.e. the number of samples for which the training loss becomes larger than 0 in logistic regression. At epoch 50 the learned features are separable at lower sample complexity $\alpha = n/d$ than at epoch 200 - even though in the later the training and generalisation performances are better.

## 3.4 Learning from real data sets

**Applying teacher/students to a real data set —** Given that the learning curves of realistic-looking inputs can be captured by the Gaussian covariate model, it is fair to ask whether the same might be true for *real data sets*. To test this idea, we first need to cast the real data set into the teacher-student formalism, and then compute the covariance matrices $\Omega, \Psi, \Phi$ and teacher vector $\theta_0$ required by model (1).

Let $\{\boldsymbol{x}^\mu, y^\mu\}_{\mu=1}^{n_{\text{tot}}}$ denote a real data set, e.g. MNIST or Fashion-MNIST for concreteness, where $n_{\text{tot}} = 7 \times 10^4$, $\boldsymbol{x}^\mu \in \mathbb{R}^D$ with $D = 784$. Without loss of generality, we can assume the data is centred. To generate the teacher, let $\boldsymbol{u}^\mu = \boldsymbol{\varphi}_t(\boldsymbol{x}^\mu) \in \mathbb{R}^p$ be a feature map such that data is invertible in feature space, i.e. that $y^\mu = \boldsymbol{\theta}_0^\top \boldsymbol{u}^\mu$ for some teacher weights $\boldsymbol{\theta}_0 \in \mathbb{R}^p$, which should be computed from the samples. Similarly, let $\boldsymbol{v}^\mu = \boldsymbol{\varphi}_s(\boldsymbol{x}^\mu) \in \mathbb{R}^d$ be a feature map we are interested in studying. Then, we can estimate the population covariances $(\Psi, \Phi, \Omega)$ empirically from the *entire* data set as:

$$\Psi = \sum_{\mu=1}^{n_{\text{tot}}} \frac{\boldsymbol{u}^\mu \boldsymbol{u}^{\mu\top}}{n_{\text{tot}}}, \qquad \Phi = \sum_{\mu=1}^{n_{\text{tot}}} \frac{\boldsymbol{u}^\mu \boldsymbol{v}^{\mu\top}}{n_{\text{tot}}}, \qquad \Omega = \sum_{\mu=1}^{n_{\text{tot}}} \frac{\boldsymbol{v}^\mu \boldsymbol{v}^{\mu\top}}{n_{\text{tot}}}. \qquad (20)$$

At this point, we have all we need to run the self-consistent equations (13). The issue with this approach is that there is not a unique teacher map $\boldsymbol{\varphi}_t$ and teacher vector $\boldsymbol{\theta}_0$ that fit the true labels. However, we can show that *all interpolating linear teachers are equivalent*:

**Theorem 3.** *(Universality of linear teachers) For any teacher feature map $\boldsymbol{\varphi}_t$, and for any $\boldsymbol{\theta}_0$ that interpolates the data so that $y^\mu = \boldsymbol{\theta}_0^\top \boldsymbol{u}^\mu \; \forall \mu$, the asymptotic predictions of model (1) are equivalent. Proof.* It follows from the fact that the teacher weights and covariances only appear in eq. (13) through $\rho = \frac{1}{p} \boldsymbol{\theta}_0^\top \Psi \boldsymbol{\theta}_0$ and the projection $\Phi^\top \boldsymbol{\theta}_0$. Using the estimation (20) and the assumption that it exists $y^\mu = \boldsymbol{\theta}_0^\top \boldsymbol{u}^\mu$, one can write these quantities directly from the labels $y^\mu$:

$$\rho = \frac{1}{n_{\text{tot}}} \sum_{\mu=1}^{n_{\text{tot}}} (y^\mu)^2, \qquad \Phi^\top \boldsymbol{\theta}_0 = \frac{1}{n_{\text{tot}}} \sum_{\mu=1}^{n_{\text{tot}}} y^\mu \boldsymbol{v}^\mu. \qquad (21)$$

For linear interpolating teachers, results are thus independent of the choice of the teacher. □

Although this result might seen surprising at first sight, it is quite intuitive. Indeed, the information about the teacher model only enters the Gaussian covariate model (1) through the statistics of $\boldsymbol{u}^\top \boldsymbol{\theta}_0$. For a linear teacher $f_0(x) = x$, this is precisely given by the labels.

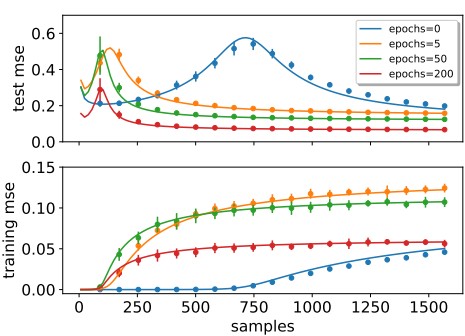

Figure 4: Test and training mean-squared errors eqs. (15) as a function of the number of samples $n$ for ridge regression. The Fashion-MNIST data set, with vanishing regularisation $\lambda = 10^{-5}$. In this plot, the student feature map $\boldsymbol{\varphi}_s$ is a 3-layer fully-connected neural network with $d = 2352$ hidden neurons trained on the full data set with the square loss. Different curves correspond to the feature map obtained at different stages of training. Simulations are averaged over 10 independent runs. Further details on the simulations are described in Appendix D.1

**Ridge Regression with linear teachers —**
We now test the prediction of model (1) on real data sets, and show that it is surprisingly effective in predicting the learning curves, at least for the ridge regression task. We have trained a 3-layer fully connected neural network with ReLU activations on the full Fashion-MNIST data set to distinguish clothing used above vs. below the waist [60]. The student feature map $\boldsymbol{\varphi}_s : \mathbb{R}^{784} \to \mathbb{R}^d$ is obtained by removing the last layer, see Appendix D.1 for a detailed description. In Fig. 4 we show the test and training errors of the ridge estimator on a sub-sample of $n < n_{\text{tot}}$ on the Fashion-MNIST images. We observe remarkable agreement between the learning curve obtained from simulations and the theoretical prediction by the matching Gaussian covariate model. Note that for the square loss and for $\lambda \ll 1$, the worst performance peak is located at the point in which the linear system becomes invertible. Curiously, Fig. 4 shows that the fully-connected network progressively learns a low-rank representation of the data as training proceeds. This can be directly verified by counting the number of zero eigenvalues of $\Omega$, which go from a full-rank matrix to a matrix of rank 380 after 200 epochs of training.

Fig. 1 (right) shows a similar experiment on the MNIST data set, but for different out-of-the-box feature maps, such as random features and the scattering transform [61], and we chose the number of random features $d = 1953$ to match the number of features from the scattering transform. Note the

characteristic double-descent behaviour [9, 25, 62], and the accurate prediction of the peak where the interpolation transition occurs. We note in Appendix D.1 that for both Figs. 4 and 1, for a number of samples $n$ closer to $n_{\text{tot}}$ we start to see deviations between the real learning curve and the theory. This is to be expected since in the teacher-student framework the student can, in principle, express the same function as the teacher if it recovers its weights exactly. Recovering the teacher weights becomes possible with a large training set. In that case, its test error will be zero. However, in our setup the test error on real data remains finite even if more training data is added, leading to the discrepancy between teacher-student learning curve and real data, see Appendix D.1 for further discussion.

Why is the Gaussian model so effective for describing learning with data that are *not* Gaussian? The point is that ridge regression is sensitive only to second order statistics, and not to the full distribution of the data. It is a classical property (see Appendix E) that the training and generalisation errors are only a function of the spectrum of the *empirical* and *population* covariances, and of their products. Random matrix theory teaches us that such quantities are very robust, and their asymptotic behaviour is universal for a broad class of distributions of $[\boldsymbol{u}, \boldsymbol{v}]$ [49, 63–65]. The asymptotic behavior of kernel matrices has indeed been the subject of intense scrutiny [11, 47, 48, 50, 66, 67]. Indeed, a universality result akin to Theorem 3 was noted in [41] in the specific case of kernel methods. We thus expect the validity of model (1) for ridge regression, with a linear teacher, to go way beyond the Gaussian assumption.

**Beyond ridge regression —** The same strategy fails beyond ridge regression and mean-squared test error. This suggests a limit in the application of model (1) to real (non-Gaussian) data to the universal linear teacher. To illustrate this, consider the setting of Figs. 4, and compare the model predictions for the binary classification error instead of the $\ell_2$ one. There is a clear mismatch between the simulated performance and prediction given by the theory (see Appendix D.1) due to the fact that the classification error does not depends only on the first two moments.

We present an additional experiment in Fig. 3. We compare the learning curves of logistic regression on a classification task on the *real* CIFAR10 images with the real labels versus the one on dcGAN-generated CIFAR10-like images and teacher generated labels from Sec. 3.3. While the Gaussian theory captures well the behaviour of the later, it fails on the former. A histogram of the distribution of the product $\boldsymbol{u}^\top \hat{\boldsymbol{w}}$ for a fixed number of samples illustrates well the deviation from the prediction of the theory with the real case, in particular on the tails of the distribution. The difference between GAN generated data (that fits the Gaussian theory) and real data is clear. Given that for classification problems there exists a number of choices of "sign" teachers and feature maps that give the exact same labels as in the data set, an interesting open question is: *is there a teacher that allows to reproduce the learning curves more accurately*? This question is left for future works.

## Acknowledgements

We thank Romain Couillet, Cosme Louart, Loucas Pillaud-Vivien, Matthieu Wyart, Federica Gerace, Luca Saglietti and Yue Lu for discussions. We are grateful to Kabir Aladin Chandrasekher, Ashwin Pananjady and Christos Thrampoulidis for pointing out discrepancies in the finite size rates and insightful related discussions. We acknowledge funding from the ERC under the European Union's Horizon 2020 Research and Innovation Programme Grant Agreement 714608-SMiLe, and from the French National Research Agency grants ANR-17-CE23-0023-01 PAIL.

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
