## Supplementary Material to *"Learning curves of generic features maps for realistic datasets with a teacher-student model"*

## A   Main result from the replica method

In this appendix we derive the formula for the performance of the Gaussian covariate model from a heuristic replica analysis. The computation closely follows the recent developments in [14, 43]. We refer to [3, 21, 22] for an introduction to this remarkable heuristic (but seemingly never failing) approach.

**The data:**   First, let's recall the definition of our model. Consider synthetic labelled data $(\boldsymbol{v}, y) \in \mathbb{R}^d \times \mathbb{R}$ drawn independently from a joint distribution with density:

$$p_{\boldsymbol{\theta}_0}(\boldsymbol{v}, y) = \int_{\mathbb{R}^p} \mathrm{d}\boldsymbol{u} \; P_0(y|\boldsymbol{u}^\top \boldsymbol{\theta}_0) \mathcal{N}(\boldsymbol{u}, \boldsymbol{v}; \boldsymbol{0}, \Sigma) \tag{22}$$

where $P_0$ is a given likelihood on $\mathbb{R}$, $\boldsymbol{\theta}_0 \in \mathbb{R}^p$ is a fixed vector of parameters and $\Sigma$ is a correlation matrix given by:

$$\Sigma = \begin{bmatrix} \Psi & \Phi \\ \Phi^\top & \Omega \end{bmatrix} \in \mathbb{R}^{(p+d) \times (p+d)} \tag{23}$$

for symmetric positive semi-definite matrices $\Psi$ and $\Omega$ and $\Phi \in \mathbb{R}^{p \times d}$. In its simplest form, which we will mostly be using in the applications, we take the likelihood $P_0(y|x) = \delta(y - f_0(x))$ to be a deterministic function with $f_0 : \mathbb{R} \to \mathbb{R}$ a non-linearity, e.g. $f_0(x) = \mathrm{sign}(x)$ to generate binary labels.

**The task:**   In our analysis, we are interested in the training and generalisation performance of a linear classifier $\hat{y} = f_{\boldsymbol{w}}(\boldsymbol{v}) = \hat{f}\left(\boldsymbol{w}^\top \boldsymbol{v}\right)$ trained on $n$ independent samples $\mathcal{D} = \{(\boldsymbol{v}^\mu, y^\mu)\}_{\mu=1}^n$ from $p_{\boldsymbol{\theta}_0}$ by minimising the regularised empirical risk:

$$\hat{\boldsymbol{w}} = \underset{\boldsymbol{w} \in \mathbb{R}^d}{\arg\min} \left[ \sum_{\mu=1}^n g\left(y^\mu, \boldsymbol{w}^\top \boldsymbol{v}^\mu\right) + \frac{\lambda}{2} ||\boldsymbol{w}||_2^2 \right], \tag{24}$$

where $\lambda > 0$ is the regularisation strength. We define the *sample complexity* $\alpha = n/d$ and the *aspect ratio* $\gamma = p/d$.

**Gibbs minimisation:**   As it was proven in Theorem 4 of the main manuscript, the asymptotic performance of the estimator in eq. (24) is fully characterised by the following scalar parameters:

$$\rho = \frac{1}{p} \boldsymbol{\theta}_0^\top \Psi \boldsymbol{\theta}_0, \qquad\qquad m^\star = \frac{1}{\sqrt{pd}} \boldsymbol{\theta}_0^\top \Phi \hat{\boldsymbol{w}}, \qquad\qquad q^\star = \frac{1}{d} \hat{\boldsymbol{w}}^\top \Omega \hat{\boldsymbol{w}} \tag{25}$$

The replica method is precisely a heuristic tool allowing us to circumvent the high-dimensional estimation problem defined in eq. (24) and giving us direct access to $(m^\star, q^\star)$.

The starting point is to define the following Gibbs measure over weights $\boldsymbol{w} \in \mathbb{R}^d$:

$$\mu_\beta(\mathrm{d}\boldsymbol{w}) = \frac{1}{\mathcal{Z}_\beta} e^{-\beta \left[ \sum_{\mu=1}^n g\left(y^\mu, \boldsymbol{w}^\top \boldsymbol{v}^\mu\right) + \frac{\lambda}{2} \sum_{i=1}^d w_i^2 \right]} \mathrm{d}\boldsymbol{w} = \frac{1}{\mathcal{Z}_\beta} \underbrace{\prod_{\mu=1}^n e^{-\beta g\left(y^\mu, \boldsymbol{w}^\top \boldsymbol{v}^\mu\right)}}_{P_g} \underbrace{\prod_{i=1}^d e^{-\frac{\beta\lambda}{2} w_i^2} \mathrm{d}w_i}_{P_w} \tag{26}$$

where $\mathcal{Z}_\beta$, known as the *partition function*, is a constant normalising the Gibbs measure $\mu_\beta$:

$$\mathcal{Z}_\beta = \int_{\mathbb{R}^d} \left( \prod_{i=1}^d \mathrm{d}w_i \right) e^{-\frac{\beta\lambda}{2} w_i^2} \prod_{\mu=1}^n e^{-\beta g\left(y^\mu, \boldsymbol{w}^\top \boldsymbol{v}^\mu\right)} \tag{27}$$

Note that $P_g$ and $P_w$ can be interpreted as a (unormalised) likelihood and prior distribution respectively. In the limit $\beta \to \infty$, the measure $\mu_\beta$ concentrates around solutions of the minimisation in eq. (24). The aim in the replica method is to compute the free energy density, defined as:

$$\beta f_\beta = -\lim_{d \to \infty} \frac{1}{d} \mathbb{E}_{\mathcal{D}} \log \mathcal{Z}_\beta. \tag{28}$$

## A.1 Replica computation of the free energy

The average in eq. (28) is not straightforward due to the logarithm term. The replica method consists of computing it using the following trick to get rid of the logarithm:

$$\log \mathcal{Z}_\beta = \lim_{r \to 0^+} \frac{1}{r} \partial_r \mathcal{Z}_\beta^r \tag{29}$$

**Averaging**

Applying the trick above, the computation of the free energy density boils down to the evaluation of the averaged replicated partition function:

$$
\begin{aligned}
\mathbb{E}_{\mathcal{D}} \mathcal{Z}_\beta^r &= \prod_{\mu=1}^{n} \mathbb{E}_{(\boldsymbol{v}^\mu, y^\mu)} \prod_{a=1}^{r} \int_{\mathbb{R}^d} P_w(\mathrm{d}\boldsymbol{w}^a) P_g \left( y^\mu \Big| \frac{\boldsymbol{v}^\mu \cdot \boldsymbol{w}^a}{\sqrt{d}} \right) \\
&= \prod_{\mu=1}^{n} \int_{\mathbb{R}} \mathrm{d}y^\mu \int_{\mathbb{R}^p} P_{\boldsymbol{\theta}_0}(\mathrm{d}\boldsymbol{\theta}_0) \int_{\mathbb{R}^{d \times r}} \left( \prod_{a=1}^{r} P_w(\mathrm{d}\boldsymbol{w}^a) \right) \underbrace{\mathbb{E}_{\boldsymbol{u}^\mu, \boldsymbol{v}^\mu} \left[ P_0 \left( y^\mu \Big| \frac{\boldsymbol{u}^\mu \cdot \boldsymbol{\theta}_0}{\sqrt{p}} \right) \prod_{a=1}^{r} P_g \left( y^\mu \Big| \frac{\boldsymbol{v}^\mu \cdot \boldsymbol{w}^a}{\sqrt{d}} \right) \right]}_{(\star)}
\end{aligned}
\tag{30}
$$

Note that in the above we included an average over the parameters $\boldsymbol{\theta}_0 \in \mathbb{R}^p$. The case in which $\boldsymbol{\theta}_0$ is a fixed vector can be recovered by choosing a point mass $P_{\boldsymbol{\theta}_0} = \delta_{\boldsymbol{\theta}_0}$. Focusing on the average term in brackets:

$$
\begin{aligned}
(\star) &= \mathbb{E}_{(\boldsymbol{u}, \boldsymbol{v})} \left[ P_0 \left( y^\mu \Big| \frac{\boldsymbol{u}^\mu \cdot \boldsymbol{\theta}_0}{\sqrt{p}} \right) \prod_{a=1}^{r} P_g \left( y^\mu \Big| \frac{\boldsymbol{v}^\mu \cdot \boldsymbol{w}^a}{\sqrt{d}} \right) \right] \\
&= \int_{\mathbb{R}} \mathrm{d}\nu_\mu P_0 \left( y | \nu_\mu \right) \int_{\mathbb{R}^r} \left( \prod_{a=1}^{r} \mathrm{d}\lambda_\mu^a P_g(y^\mu | \lambda_\mu^a) \right) \underbrace{\mathbb{E}_{(\boldsymbol{u}^\mu, \boldsymbol{v}^\mu)} \left[ \delta \left( \nu_\mu - \frac{\boldsymbol{u}^\mu \cdot \boldsymbol{\theta}_0}{\sqrt{p}} \right) \prod_{a=1}^{r} \delta \left( \lambda_\mu^a - \frac{\boldsymbol{v}^\mu \cdot \boldsymbol{w}^a}{\sqrt{d}} \right) \right]}_{P(\nu, \lambda)}
\end{aligned}
$$

Note that the term in brackets defines the joint density over $(\nu_\mu, \lambda_\mu^a)$. It is easy to check that these are Gaussian random variables with zero mean and covariance matrix given by:

$$\Sigma^{ab} = \begin{pmatrix} \rho & m^a \\ m^a & Q^{ab} \end{pmatrix}. \tag{31}$$

where the so-called overlap parameters $(\rho, m^a, Q^{ab})$ are related to the weights $\boldsymbol{\theta}_0, \boldsymbol{w}$:

$$\rho \equiv \mathbb{E}\left[\nu_\mu^2\right] = \frac{1}{p} \boldsymbol{\theta}_0^\top \Psi \boldsymbol{\theta}_0, \quad m^a \equiv \mathbb{E}\left[\lambda_\mu^a \nu_\mu\right] = \frac{1}{\sqrt{pd}} \boldsymbol{\theta}_0^\top \Phi \boldsymbol{w}^a, \quad Q^{ab} \equiv \mathbb{E}\left[\lambda_\mu^a \lambda_\mu^b\right] = \frac{1}{d} \boldsymbol{w}^{a\top} \Omega \boldsymbol{w}^b$$

We can therefore write the averaged replicated partition function as:

$$
\begin{aligned}
\mathbb{E}_{\mathcal{D}} \mathcal{Z}_\beta^r &= \prod_{\mu=1}^{n} \int \mathrm{d}y^\mu \int_{\mathbb{R}^p} P_{\boldsymbol{\theta}_0}(\mathrm{d}\boldsymbol{\theta}_0) \int_{\mathbb{R}^{d \times r}} \left( \prod_{a=1}^{r} P_w(\mathrm{d}\boldsymbol{w}^a) \right) \int_{\mathbb{R}} \mathrm{d}\nu_\mu P_0(y^\mu | \nu_\mu) \times \\
&\quad \times \int_{\mathbb{R}^r} \left( \prod \mathrm{d}\lambda_\mu^a P_g \left( y^\mu | \lambda_\mu^a \right) \right) \mathcal{N}(\nu_\mu, \lambda_\mu^a; \boldsymbol{0}, \Sigma^{ab})
\end{aligned}
\tag{32}
$$

**Rewriting as a saddle-point problem**

The next step is to free the overlap parameters by introducing delta functions:

$$1 \propto \int_{\mathbb{R}} \mathrm{d}\rho \, \delta\left(p\rho - \boldsymbol{\theta}_0^\top \Psi \boldsymbol{\theta}_0\right) \int_{\mathbb{R}^r} \prod_{a=1}^r \mathrm{d}m^a \, \delta\left(\sqrt{pd}m^a - \boldsymbol{\theta}_0^\top \Phi \boldsymbol{w}^a\right)$$

$$\times \int_{\mathbb{R}^{r \times r}} \prod_{1 \le a \le b \le r} \mathrm{d}Q^{ab} \, \delta\left(dQ^{ab} - \boldsymbol{w}^{a\top} \Omega \boldsymbol{w}^b\right)$$

$$= \int_{\mathbb{R}} \frac{\mathrm{d}\rho \mathrm{d}\hat{\rho}}{2\pi} e^{i\hat{\rho}\left(p\rho - \boldsymbol{\theta}_0^\top \Psi \boldsymbol{\theta}_0\right)} \int_{\mathbb{R}^r} \prod_{a=1}^r \frac{\mathrm{d}m^a \mathrm{d}\hat{m}^a}{2\pi} e^{i \sum_{a=1}^r \hat{m}^a \left(\sqrt{pd}m^a - \boldsymbol{\theta}_0^\top \Phi \boldsymbol{w}^a\right)} \times$$

$$\times \int_{\mathbb{R}^{r \times r}} \prod_{1 \le a \le b \le r} \frac{\mathrm{d}Q^{ab} \mathrm{d}\hat{Q}^{ab}}{2\pi} e^{i \sum_{1 \le a \le b \le r} \hat{Q}^{ab}\left(dQ^{ab} - \boldsymbol{w}^{a\top} \Omega \boldsymbol{w}^b\right)} \tag{33}$$

Inserting this in eq. (32) allow us to rewrite:

$$\mathbb{E}_{\mathcal{D}} \mathcal{Z}_\beta^r = \int_{\mathbb{R}} \frac{\mathrm{d}\rho \mathrm{d}\hat{\rho}}{2\pi} \int_{\mathbb{R}^r} \prod_{a=1}^r \frac{\mathrm{d}m^a \mathrm{d}\hat{m}^a}{2\pi} \int_{\mathbb{R}^{r \times r}} \prod_{1 \le a \le b \le r} \frac{\mathrm{d}Q^{ab} \mathrm{d}\hat{Q}^{ab}}{2\pi} e^{d\Phi^{(r)}} \tag{34}$$

where we have absorbed a $-i$ factor in the integrals (this won't matter since we will look to the saddle-point) and defined the potential:

$$\Phi^{(r)} = -\gamma\rho\hat{\rho} - \sqrt{\gamma} \sum_{a=1}^r m^a \hat{m}^a - \sum_{1 \le a \le b \le r} Q^{ab} \hat{Q}^{ab} + \alpha \Psi_y^{(r)}(\rho, m^a, Q^{ab}) + \Psi_w^{(r)}(\hat{\rho}, \hat{m}^a, \hat{Q}^{ab}) \tag{35}$$

where we recall that $\alpha = n/d$, $\gamma = p/d$ and:

$$\Psi_w^{(r)} = \frac{1}{d} \log \int_{\mathbb{R}^p} P_{\boldsymbol{\theta}_0}(\mathrm{d}\boldsymbol{\theta}_0) \int_{\mathbb{R}^{d \times r}} \prod_{a=1}^r P_w(\mathrm{d}\boldsymbol{w}^a) e^{\hat{\rho}\boldsymbol{\theta}_0^\top \Psi \boldsymbol{\theta}_0 + \sum_{a=1}^r \hat{m}^a \boldsymbol{\theta}_0^\top \Phi \boldsymbol{w}^a + \sum_{1 \le a \le b \le r} \hat{Q}^{ab} \boldsymbol{w}^{a\top} \Omega \boldsymbol{w}^b} \tag{36}$$

$$\Psi_y^{(r)} = \log \int_{\mathbb{R}} \mathrm{d}y \int_{\mathbb{R}} \mathrm{d}\nu \, P_0(y|\nu) \int \prod_{a=1}^r \mathrm{d}\lambda^a P_g(y|\lambda^a) \, \mathcal{N}(\nu, \lambda^a; \boldsymbol{0}, \Sigma^{ab}) \tag{37}$$

In the high-dimensional limit where $d \to \infty$ while $\alpha = n/d$ and $\gamma = p/d$ stay finite, the integral in eq. (34) concentrate around the values of the overlaps that extremise $\Phi^{(r)}$, and therefore we can write:

$$\beta f_\beta = -\lim_{r \to 0^+} \frac{1}{r} \mathbf{extr} \, \Phi^{(r)}\left(\hat{\rho}, \hat{m}^a, \hat{Q}^{ab}; \rho, m^a, Q^{ab}\right) \tag{38}$$

**Replica symmetric ansatz**

In order to proceed with the $r \to 0^+$ limit, we restrict the extremisation above to the following replica symmetric ansatz:

$$\begin{aligned}
m^a &= m, & \hat{m}^a &= \hat{m}, & \text{for } a = 1, \dots, r \\
Q^{aa} &= r, & \hat{Q}^{aa} &= -\frac{1}{2}\hat{r}, & \text{for } a = 1, \dots, r \\
Q^{ab} &= q, & \hat{Q}^{ab} &= \hat{q}, & \text{for } 1 \le a < b \le r
\end{aligned} \tag{39}$$

Inserting this ansatz in eq. (35) allows us to explicitly take the $r \to 0^+$ limit for each term. The first three terms are straightforward to obtain. The limit of $\Psi_y^{(r)}$ is cumbersome, but it common to many replica computations for the generalised linear likelihood $P_g$. We refer the curious reader to Appendix C of [43] or to Appendix IV of [14] for details, and write the final result here:

$$\Psi_y \equiv \lim_{r \to 0^+} \frac{1}{r} \Psi_w^{(r)} = \mathbb{E}_{\xi \sim \mathcal{N}(0,1)} \left[ \int_{\mathbb{R}} \mathrm{d}y \, \mathcal{Z}_0\left(y, \frac{m}{\sqrt{q}}\xi, \rho - \frac{m^2}{q}\right) \log \mathcal{Z}_g(y, \sqrt{q}\xi, V) \right] \tag{40}$$

where we have defined $V = r - q$ and:

$$\mathcal{Z}_{g/0}(y, \omega, V) = \mathbb{E}_{x \sim \mathcal{N}(\omega, V)} \left[ P_{g/0}(y|x) \right] . \tag{41}$$

Note that as in [43], the consistency condition of the zeroth order term in the free energy fix $\rho = \mathbb{E}_{\boldsymbol{\theta}_0} \left[ \frac{1}{p} \boldsymbol{\theta}_0^\top \Psi \boldsymbol{\theta}_0 \right]$ and $\hat{\rho} = 0$. On the other hand, the limit of the prior term here is exactly as the one discussed in Appendix C of [45], and is given by:

$$\Psi_w \equiv \lim_{r \to 0^+} \frac{1}{r} \Psi_w^{(r)} = \frac{1}{d} \mathbb{E}_{\xi, \boldsymbol{\theta}_0} \log \int_{\mathbb{R}^d} P_w (\mathrm{d}\boldsymbol{w}) \, e^{-\frac{\hat{V}}{2} \boldsymbol{w}^\top \Omega \boldsymbol{w} + \boldsymbol{w}^\top \left( \hat{m} \Phi^\top \boldsymbol{\theta}_0 + \hat{q} \Omega^{1/2} \boldsymbol{\xi} \right)} . \tag{42}$$

**Summary**

The replica symmetric free energy density is simply given by:

$$f_\beta = \operatorname*{extr}_{q, m, \hat{q}, \hat{m}} \left\{ -\frac{1}{2} r \hat{r} - \frac{1}{2} q \hat{q} + \sqrt{\gamma} \, m \hat{m} - \alpha \Psi_y(r, m, q) - \Psi_w(\hat{r}, \hat{m}, \hat{q}) \right\} \tag{43}$$

where

$$\Psi_w = \lim_{d \to \infty} \frac{1}{d} \mathbb{E}_{\xi, \boldsymbol{\theta}_0} \log \int_{\mathbb{R}^d} P_w (\mathrm{d}\boldsymbol{w}) \, e^{-\frac{\hat{V}}{2} \boldsymbol{w}^\top \Omega \boldsymbol{w} + \boldsymbol{w}^\top \left( \hat{m} \Phi^\top \boldsymbol{\theta}_0 + \hat{q} \Omega^{1/2} \boldsymbol{\xi} \right)}$$

$$\Psi_y = \mathbb{E}_{\xi \sim \mathcal{N}(0,1)} \left[ \int_{\mathbb{R}} \mathrm{d}y \, \mathcal{Z}_0 \left( y, \frac{m}{\sqrt{q}} \xi, \rho - \frac{m^2}{q} \right) \log \mathcal{Z}_g(y, \sqrt{q}\xi, V) \right] \tag{44}$$

## A.2 Ridge regression and fixed weights

For an $\ell_2$-regularisation term, we have:

$$P_w(\mathrm{d}\boldsymbol{w}) = \frac{1}{(2\pi)^{d/2}} e^{-\frac{\beta\lambda}{2} \|\boldsymbol{w}\|_2^2} \mathrm{d}\boldsymbol{w} \tag{45}$$

where we have included a convenient constant, and therefore:

$$\int_{\mathbb{R}^d} P_w(\mathrm{d}\boldsymbol{w}) e^{-\frac{\hat{V}}{2} \boldsymbol{w}^\top \Omega \boldsymbol{w} + \boldsymbol{w}^\top \left( \hat{m} \Phi^\top \boldsymbol{\theta}_0 + \sqrt{\hat{q}} \Omega^{1/2} \boldsymbol{\xi} \right)} = \int_{\mathbb{R}^d} \frac{\mathrm{d}\boldsymbol{w}}{(2\pi)^{p/2}} e^{-\frac{1}{2} \boldsymbol{w}^\top \left( \beta\lambda \mathrm{I}_d + \hat{V}\Omega \right) \boldsymbol{w} + \boldsymbol{w}^\top \left( \hat{m} \Phi^\top \boldsymbol{\theta}_0 + \sqrt{\hat{q}} \Omega^{1/2} \boldsymbol{\xi} \right)}$$

$$= \frac{\exp \left( \frac{1}{2} \left( \hat{m} \Phi^\top \boldsymbol{\theta}_0 + \sqrt{\hat{q}} \Omega^{1/2} \boldsymbol{\xi} \right)^\top \left( \beta\lambda \mathrm{I}_d + \hat{V}\Omega \right)^{-1} \left( \hat{m} \Phi^\top \boldsymbol{\theta}_0 + \sqrt{\hat{q}} \Omega^{1/2} \boldsymbol{\xi} \right)^\top \right)}{\sqrt{\det \left( \beta\lambda \mathrm{I}_d + \hat{V}\Omega \right)}} \tag{46}$$

taking the log and using $\log \det = \operatorname{tr} \log$, up to the limit:

$$\Psi_w = \frac{1}{2d} \mathbb{E}_{\xi, \boldsymbol{\theta}_0} \left[ \left( \hat{m} \Phi^\top \boldsymbol{\theta}_0 + \sqrt{\hat{q}} \Omega^{1/2} \boldsymbol{\xi} \right)^\top \left( \beta\lambda \mathrm{I}_d + \hat{V}\Omega \right)^{-1} \left( \hat{m} \Phi^\top \boldsymbol{\theta}_0 + \sqrt{\hat{q}} \Omega^{1/2} \boldsymbol{\xi} \right) \right]$$

$$- \frac{1}{2d} \operatorname{tr} \log \left( \beta\lambda \mathrm{I}_d + \hat{V}\Omega \right) \tag{47}$$

Defining the shorthand $\mathrm{A} = \left( \beta\lambda \mathrm{I}_d + \hat{V}\Omega \right)^{-1}$, we can now take the averages over $\boldsymbol{\xi}$ explicitly:

$$\mathbb{E}_{\boldsymbol{\xi}} \left[ \left( \hat{m} \Phi^\top \boldsymbol{\theta}_0 + \sqrt{\hat{q}} \Omega^{1/2} \boldsymbol{\xi} \right)^\top \mathrm{A} \left( \hat{m} \Phi^\top \boldsymbol{\theta}_0 + \hat{q} \Omega^{1/2} \boldsymbol{\xi} \right) \right] = \hat{m}^2 \boldsymbol{\theta}_0^\top \Phi \mathrm{A} \Phi^\top \boldsymbol{\theta}_0 + \hat{q} \operatorname{tr} \Omega^{1/2} \mathrm{A} \Omega^{1/2} \tag{48}$$

Putting together, up to the limit:

$$\Psi_w = -\frac{1}{2d} \operatorname{tr} \log \left( \beta\lambda \mathrm{I}_d + \hat{V}\Omega \right) + \frac{1}{2d} \operatorname{tr} \left( \hat{m}^2 \Phi^\top \boldsymbol{\theta}_0 \boldsymbol{\theta}_0^\top \Phi + \hat{q}\Omega \right) \left( \beta\lambda \mathrm{I}_d + \hat{V}\Omega \right)^{-1} \tag{49}$$

## A.3 Taking the $\beta \to \infty$ limit

Finally, in order to take the $\beta \to \infty$ limit explicitly, we note that under the rescaling

$$V \to \beta^{-1}V \qquad\qquad q \to q \qquad\qquad m \to m$$
$$\hat{V} \to \beta\hat{V} \qquad\qquad \hat{q} \to \beta^2\hat{q} \qquad\qquad \hat{m} \to \beta\hat{m}. \tag{50}$$

The potential $\Psi_w$ has a trivial limit:

$$\lim_{\beta\to\infty} \frac{1}{\beta}\Psi_w = -\frac{1}{2d}\operatorname{tr}\log\left(\lambda\mathrm{I}_d + \hat{V}\Omega\right) + \frac{1}{2d}\operatorname{tr}\left(\hat{m}^2\Phi^\top\boldsymbol{\theta}_0\boldsymbol{\theta}_0^\top\Phi + \hat{q}\Omega\right)\left(\lambda\mathrm{I}_d + \hat{V}\Omega\right)^{-1} \tag{51}$$

while $\Psi_y$ requires more attention. Since $\mathcal{Z}_0$ only depends on $(q, m)$, it is invariant under the rescaling. On the other hand, we have that:

$$\mathcal{Z}_g(y, \sqrt{q}\xi, V) = \sqrt{\beta}\int \frac{\mathrm{d}x}{\sqrt{2\pi V}}e^{-\beta\left[\frac{(x-\sqrt{q}\xi)^2}{2V} + g(y,x)\right]} \underset{\beta\to\infty}{=} e^{-\beta\mathcal{M}_{Vg(y,\cdot)}(\sqrt{q}\xi)} \tag{52}$$

where $\mathcal{M}$ is the Moreau envelope associated to the loss $g$:

$$\mathcal{M}_{\tau g(y,\cdot)}(x) = \inf_{z\in\mathbb{R}}\left[\frac{(z-x)^2}{2\tau} + g(y,z)\right] \tag{53}$$

and therefore:

$$\lim_{\beta\to\infty}\frac{1}{\beta}\Psi_y = -\mathbb{E}_{\xi\sim\mathcal{N}(0,1)}\left[\int\mathrm{d}y\,\mathcal{Z}_0\left(y, \frac{m}{\sqrt{q}}\xi, \rho - \frac{m^2}{q}\right)\mathcal{M}_{Vg(y,\cdot)}\left(\sqrt{q}\xi\right)\right] \tag{54}$$

The zero temperature therefore is simply given by:

$$\lim_{\beta\to\infty} f_\beta = \operatorname*{extr}_{V,q,m,\hat{V},\hat{q},\hat{m}}\left\{-\frac{1}{2}\left(q\hat{V} - \hat{q}V\right) + \sqrt{\gamma}\,m\hat{m} + \alpha\mathbb{E}_{\xi\sim\mathcal{N}(0,1)}\left[\int\mathrm{d}y\,\mathcal{Z}_0\,\mathcal{M}_{Vg(y,\cdot)}\right]\right.$$
$$\left. -\frac{1}{2d}\operatorname{tr}\left(\hat{m}^2\Phi^\top\boldsymbol{\theta}_0\boldsymbol{\theta}_0^\top\Phi + \hat{q}\Omega\right)\left(\lambda\mathrm{I}_d + \hat{V}\Omega\right)^{-1}\right\} \tag{55}$$

## A.4 Saddle-point equations

To solve the extremisation problem defined by eq. (55), we search for vanishing gradient points of the potential. This lead to a set of self-consistent *saddle-point* equations:

$$\begin{cases} \hat{V} = -\alpha\mathbb{E}_\xi\left[\int_\mathbb{R}\mathrm{d}y\,\mathcal{Z}_0\,\partial_\omega f_g\right] \\ \hat{q} = \alpha\mathbb{E}_\xi\left[\int_\mathbb{R}\mathrm{d}y\,\mathcal{Z}_0 f_g^2\right] \\ \hat{m} = \frac{\alpha}{\sqrt{\gamma}}\mathbb{E}_\xi\left[\int_\mathbb{R}\mathrm{d}y\,\partial_\omega\mathcal{Z}_0\,f_g\right] \end{cases} \qquad \begin{cases} V = \frac{1}{d}\operatorname{tr}\left(\lambda\mathrm{I}_d + \hat{V}\Omega\right)^{-1}\Omega \\ q = \frac{1}{d}\operatorname{tr}\left[\left(\hat{q}\Omega + \hat{m}^2\Phi^\top\boldsymbol{\theta}_0\boldsymbol{\theta}_0^\top\Phi\right)\Omega\left(\lambda\mathrm{I}_d + \hat{V}\Omega\right)^{-2}\right] \\ m = \frac{1}{\sqrt{\gamma}}\frac{\hat{m}}{p}\operatorname{tr}\Phi^\top\boldsymbol{\theta}_0\boldsymbol{\theta}_0^\top\Phi\left(\lambda\mathrm{I}_d + \hat{V}\Omega\right)^{-1} \end{cases} \tag{56}$$

where $f_g(y, \omega, V) = -\partial_\omega\mathcal{M}_{Vg(y,\cdot)}(\omega)$, which can also be obtained from the proximal operator

$$\operatorname{prox}_{Vg(y,\cdot)}(\omega) = \arg\min_{z\in\mathbb{R}}\left[\frac{(z-\omega)^2}{2V} + g(y,z)\right] \tag{57}$$

using the envelope theorem $\mathcal{M}'_{\tau f}(x) = \tau^{-1}\left(x - \operatorname{prox}_{\tau f}(x)\right)$. A python implementation of the saddle-point equations for the losses discussed below is available in `https://github.com/IdePHICS/GCMProject`

## A.5 Examples

We now discuss a couple of examples in which the equations above simplify.

**Ridge regression:** Consider a ridge regression task with $f_0(x) = \hat{f}(x) = x$, loss $g(y, x) = \frac{1}{2}(y - x)^2$ and choose $\hat{g}(y, x) = \frac{1}{2}(y - x)^2$. In this case, our model is closely related to the mismatched models in [12] and [26]. In the first, labels are generated in a higher-dimensional space which contains the features as a subspace, and can be mapped to our model in the case $p > d$ by defining the projection of the teacher weights in the student space $\Phi^\top \boldsymbol{\theta}_0 \in \mathbb{R}^d$ and its orthogonal complement $(\Phi^\top \boldsymbol{\theta}_0)^\perp \in \mathbb{R}^{p-d}$. In the second, the teacher acts on an orthogonal subset of the features, and can be mapped with a similar construction to our model in the case $p < d$. These two cases were studied for specific linear tasks, such as ridge and random features regression, with the covariances modelling structure in the data. Conceptually, our model differs slightly in the sense that any additional fixed feature layer, e.g. random projections or a pre-trained feature map, is also contained in the convariances.

For the linear task, the asymptotic training and generalisation errors read:

$$\mathcal{E}^\star_{\text{train.}} = \frac{\rho + q^\star - 2m^\star}{(1 + V^\star)^2}, \qquad\qquad \mathcal{E}^\star_{\text{gen.}} = \rho + q^\star - 2m^\star \tag{58}$$

where $\rho = \frac{1}{p} \boldsymbol{\theta}_0^\top \Psi \boldsymbol{\theta}_0$ and $(V^\star, q^\star, m^\star)$ are the fixed point of the following set of self-consistent equations:

$$
\begin{cases}
\hat{V} = \frac{\alpha}{1+V} \\
\hat{q} = \alpha \frac{\rho+q-2m}{(1+V)^2} \\
\hat{m} = \frac{1}{\sqrt{\gamma}} \frac{\alpha}{1+V}
\end{cases}
, \qquad
\begin{cases}
V = \frac{1}{d} \operatorname{tr} \left( \lambda \mathrm{I}_d + \hat{V}\Omega \right)^{-1} \Omega \\
q = \frac{1}{d} \operatorname{tr} \left[ \left( \hat{q}\Omega + \hat{m}^2 \Phi^\top \boldsymbol{\theta}_0 \boldsymbol{\theta}_0^\top \Phi \right) \Omega \left( \lambda \mathrm{I}_d + \hat{V}\Omega \right)^{-2} \right] \\
m = \frac{1}{\sqrt{\gamma}} \frac{\hat{m}}{d} \operatorname{tr} \Phi^\top \boldsymbol{\theta}_0 \boldsymbol{\theta}_0^\top \Phi \left( \lambda \mathrm{I}_d + \hat{V}\Omega \right)^{-1}
\end{cases}
. \tag{59}
$$

Note that quite interestingly we have the following relationship between the training and generalisation error:

$$\mathcal{E}^\star_{\text{train.}} = \frac{\mathcal{E}^\star_{\text{gen.}}}{(1 + V^\star)^2}. \tag{60}$$

This give us an interesting interpretation of $V^\star$ as parametrising the variance gap between the generalisation and training error[1]. In particular, note that $V^\star$ only depends on the spectrum of the population covariance, since it is the solution of:

$$V = \int \nu_\Omega(\mathrm{d}\omega) \frac{\omega}{\lambda + \frac{\alpha\omega}{1+V}} \tag{61}$$

where $\nu_\Omega$ is the spectral density of $\Omega$.

**Binary classification**

For a binary classification task, we tak $f_0(x) = \hat{f}(x) = \operatorname{sign}(\mathrm{x}) \in \{-1, 1\}$. Our equations generalise the ones derived [33] in the specific case of $d = p$ and $\Psi = \mathrm{I}_d$, $\boldsymbol{\theta}_0 \sim \mathcal{N}(\mathbf{0}, \mathrm{I}_d)$. For binary classification, the asymptotic classification error $\mathcal{E}_{\text{gen.}}(\hat{w}) = \mathbb{P}\left( y \neq \operatorname{sign}(\hat{\mathrm{w}}^\top \mathrm{u}) \right)$ can be explicitly writen is terms of the overlaps as:

$$\mathcal{E}^\star_{\text{gen.}} = \frac{1}{\pi} \cos^{-1} \left( \frac{m^\star}{\sqrt{\rho q^\star}} \right). \tag{62}$$

where again $(q^\star, m^\star)$ are solutions of the self-consistent saddle-point equations. The teacher measure is given by:

$$\mathcal{Z}_0(y, \omega, V) = \frac{\delta_{y,1} + \delta_{y,-1}}{2} \left( 1 + \operatorname{erf}\left( \frac{y\omega}{\sqrt{2V}} \right) \right) \tag{63}$$

The explicit form of the equation depends on the choice of the loss function, three of which are of particular interest:

---

[1]We thank Stéphane d'Ascoli for bringing this relation to our attention.

**Square-loss:** As in the ridge case, for $g(y, x) = \frac{1}{2}(y - x)^2$ the saddle-point equations simplify considerably:

$$
\begin{cases}
\hat{V} = \frac{\alpha}{1+V} \\
\hat{q} = \alpha \frac{1+q-2m\sqrt{\frac{2}{\pi\rho}}}{(1+V)^2} \\
\hat{m} = \sqrt{\frac{2}{\pi\rho}} \frac{\alpha}{1+V}
\end{cases}
,
\qquad
\begin{cases}
V = \frac{1}{d} \operatorname{tr} \left(\lambda \mathrm{I}_d + \hat{V}\Omega\right)^{-1} \Omega \\
q = \frac{1}{d} \operatorname{tr} \left[\left(\hat{q}\Omega + \hat{m}^2 \Phi^\top \boldsymbol{\theta}_0 \boldsymbol{\theta}_0^\top \Phi\right) \Omega \left(\lambda \mathrm{I}_d + \hat{V}\Omega\right)^{-2}\right] \\
m = \frac{1}{\sqrt{\gamma}} \frac{\hat{m}\gamma}{d} \operatorname{tr} \Phi^\top \boldsymbol{\theta}_0 \boldsymbol{\theta}_0^\top \Phi \left(\lambda \mathrm{I}_d + \hat{V}\Omega\right)^{-1}
\end{cases}
. \tag{64}
$$

Similarly, the asymptotic training error also admits a simple expression:

$$
\mathcal{E}_{\text{train.}}^\star = \frac{1}{4} \frac{1 + q^\star - 2m^\star \sqrt{\frac{2}{\pi\rho}}}{(1 + V^\star)^2} \tag{65}
$$

**Logistic regression:** Different from the previous cases, for logistic loss $g(y, x) = \log\left(1 + e^{-yx}\right)$ the equations for $(\hat{V}, \hat{q}, \hat{m})$ cannot be integrated explicitly, since the proximal operator doesn't admit a closed form solution. Instead, $f_g$ can be found by solving the following self-consistent equation:

$$
f_g = \frac{y}{1 + e^{y(Vf_g + \omega)}}. \tag{66}
$$

**Soft-margin regression:** Another useful case in which the proximal operator has a closed form solution is for the hinge loss $g(y, x) = \max(0, 1 - yx)$. In this case:

$$
f_g(y, \omega, V) =
\begin{cases}
y & \text{if } \omega y < 1 - V \\
\frac{y - \omega}{V} & \text{if } 1 - V < \omega y < 1 \\
0 & \text{otherwise}
\end{cases}
, \quad
\partial_\omega f_g(y, \omega, V) =
\begin{cases}
-\frac{1}{V} & \text{if } 1 - V < \omega y < 1 \\
0 & \text{otherwise}
\end{cases}
\tag{67}
$$

Again, the equations cannot be integrated explicitly. Note that in the limit $\lambda \to 0$, both the logistic and soft-margin solutions converge to the max-margin estimator

### A.6  Relation to previous models

**Random features:** The feature map for random features learning can be written as:

$$
\Phi_{\mathrm{F}} : \boldsymbol{u} \in \mathbb{R}^p \mapsto \boldsymbol{v} = \sigma\left(\frac{1}{\sqrt{k}} \mathbf{F} \boldsymbol{u}\right) \in \mathbb{R}^d \tag{68}
$$

where $\boldsymbol{u} \in \mathbb{R}^p$ is the original data, $\mathbf{F} \in \mathbb{R}^{d \times k}$ is a chosen random projection matrix and $\sigma : \mathbb{R} \to \mathbb{R}$ is a chosen non-linearity acting component-wise in $\mathbb{R}^d$, see [16]. Random features learning has attracted a lot of interest recently, and has been studied in [11, 33, 35, 43] in the case of Gaussian data $\boldsymbol{u} \sim \mathcal{N}(\mathbf{0}, \mathrm{I}_p)$. Our model encompasses all of these works, and in the case of Gaussian data the covariance $(\Psi, \Omega, \Phi)$ can be explicitly related to the projection matrix F:

$$
\Psi = \mathrm{I}_p, \qquad \Phi = \kappa_1 \mathbf{F}, \qquad \Omega = \kappa_0^2 \mathbf{1}_d \mathbf{1}_d^\top + \kappa_1^2 \frac{\mathbf{F}\mathbf{F}^\top}{d} + \kappa_\star^2 \mathrm{I}_d \tag{69}
$$

where $\mathbf{1}_d \in \mathbb{R}^d$ is the all-ones vector and the constants $(\kappa_0, \kappa_1, \kappa_\star)$ are related to $\sigma$ as:

$$
\kappa_0 = \mathbb{E}_{z \sim \mathcal{N}(0,1)}\left[\sigma(z)\right], \quad \kappa_1 = \mathbb{E}_{z \sim \mathcal{N}(0,1)}\left[z\sigma(z)\right], \quad \kappa_\star = \sqrt{\mathbb{E}_{z \sim \mathcal{N}(0,1)}\left[\sigma(z)^2\right] - \kappa_0^2 - \kappa_1^2} \tag{70}
$$

These relations hold asymptotically, and rely on the *Gaussian equivalence theorem* (GET), see [45] for a proof.

**Generative models:** In [45], a similar Gaussian covariate model was used to study the performance of random feature regression on data generated from pre-trained generative models:

$$
\boldsymbol{v} = \mathcal{G}(\boldsymbol{u}) \in \mathbb{R}^d, \qquad\qquad \boldsymbol{u} \sim \mathcal{N}(\mathbf{0}, \mathrm{I}_p). \tag{71}
$$

where $\mathcal{G} : \mathbb{R}^p \to \mathbb{R}^d$ is a generative network mapping the latent space $\mathbb{R}^p$ to the input space $\mathbb{R}^d$ (e.g. a pre-trained GAN). Labels were generated directly in the latent space $\mathbb{R}^p$ using a generalised linear model on random weights: $y = f_0\left(\boldsymbol{\theta}_0^\top \boldsymbol{u}\right)$ with $\boldsymbol{u} \sim \mathcal{N}(\boldsymbol{0}, \mathrm{I}_p)$. A *Gaussian Equivalence Principle* (GEP) stating that the asymptotic generalisation and training performances of this model are fully captured by second order statistics was conjectured and shown to hold numerically for different choices of generative models $\mathcal{G}$. Indeed, this model is a particular case of ours when $\Psi = \mathrm{I}_p$ and $\boldsymbol{\theta}_0 \sim \mathcal{N}(\boldsymbol{0}, \mathrm{I}_p)$. Assuming that the GEP holds, our model therefore can be seen as a generalisation of [45] to structured teachers. For instance, in Section 3.3 of the main we show several cases in which the teacher $\boldsymbol{u} = \tilde{\mathcal{G}}(\boldsymbol{c})$ for a latent vector $\boldsymbol{c} \sim \mathcal{N}(\boldsymbol{0}, \mathrm{I}_k)$ and a pre-trained map $\tilde{\mathcal{G}}$ that can include a generative model and a fixed feature map (e.g. random features, scattering transform, pre-learned neural network, etc.). Also, it is important to stress that our model also account for the case in which the teacher weights $\boldsymbol{\theta}_0 \in \mathbb{R}^p$ are fixed, and therefore can be also learned.

**Kernel methods:** Let $\mathcal{H}$ be a Kernel Reproducing Hilbert space (RKHS) associated to a given kernel $K$ and $\mathcal{D} = \{\boldsymbol{x}^\mu, y^\mu\}_{\mu=1}^n$ be a labelled data set with $\boldsymbol{x} \sim p_x$ independently, and set $\mathcal{X} = \mathrm{supp}(\mathrm{p_x})$. In Kernel regression, the aim is to solve:

$$\min_{f \in \mathcal{H}} \left[ \frac{1}{2} \sum_{\mu=1}^n \left( y^\mu - f(\boldsymbol{x}^\mu) \right)^2 + \frac{\lambda}{2} ||f||_{\mathcal{H}}^2 \right] \tag{72}$$

where $|| \cdot ||_{\mathcal{H}}$ is the norm induced by the scalar product in $\mathcal{H}$. An alternative representation of this problem is given by the feature decomposition of the kernel given by Mercer's theorem:

$$K(\boldsymbol{x}, \boldsymbol{x}') = \sum_{i=1}^\infty \omega_i e_i(\boldsymbol{x}') e_i(\boldsymbol{x}) \tag{73}$$

where $\omega_i$ and $e_i(\boldsymbol{x})$ are the eigenvalues and eigenvectors associated with the kernel:

$$\int_{\mathbb{R}^k} p_x(\mathrm{d}\boldsymbol{x}') K(\boldsymbol{x}, \boldsymbol{x}') e_i(\boldsymbol{x}') = \omega_i e_i(\boldsymbol{x}) \tag{74}$$

Note that $\{e_i(\boldsymbol{x})\}_i^\infty$ form an orthonormal basis of the space of square-integrable functions $L^2(\mathcal{X})$ (with respect to the standard scalar product of $L^2$). It is also convenient to define the feature map $\varphi_i(\boldsymbol{x}) = \sqrt{\omega_i} e_i(\boldsymbol{x})$, which is an orthonormal basis of $\mathcal{H} \subset L^2(\mathcal{X})$ (with respect to the scalar product induced by $K$). Therefore, if we assume that the labels $y^\mu = f_0(\boldsymbol{x}^\mu)$ are generated from a ground truth target function (not necessarily part of $\mathcal{H}$), we can expand both $f$ and $f_0$ the feature basis:

$$f(\boldsymbol{x}) = \sum_{i=1}^\infty w_i \varphi_i(\boldsymbol{x}), \qquad\qquad f_0(\boldsymbol{x}) = \sum_{i=1}^\infty \theta_{0i} \varphi_i(\boldsymbol{x}) \tag{75}$$

Note that $f \in \mathcal{H}$ implies that for this sum to make sense $w_i$ needs to decay fast enough with respect to $\sqrt{w_i}$, but in general we can have $f_0 \notin \mathcal{H}$ meaning that $\theta_0^i$ decays slower than $\sqrt{\omega_i}$ but still fast enough such that $f_0 \in L^2(\mathcal{X})$. If the number of features is finite ($\omega_i = 0$ for $i \geq d$) or if we introduce a cut-off $d \gg n, |\mathcal{X}|$, the representation in the feature basis in eq. (75) allow us to rewrite Kernel regression problem in eq. (72) simply as ridge regression in feature space:

$$\min_{\boldsymbol{w} \in \mathbb{R}^d} \left[ \frac{1}{2} \sum_{\mu=1}^n \left( \boldsymbol{\theta}_0^\top \boldsymbol{\varphi}(\boldsymbol{x}) - \boldsymbol{w}^\top \boldsymbol{\varphi}(\boldsymbol{x}) \right)^2 + \frac{\lambda}{2} ||\boldsymbol{w}||_2^2 \right] . \tag{76}$$

Letting $\boldsymbol{v} = \boldsymbol{\varphi}(\boldsymbol{x}) \in \mathbb{R}^d$, this formulation is equivalent to our model with $p = d$ and covariance matrices given by:

$$\Psi = \Phi = \Omega = \mathrm{diag}(\omega_\mathrm{i}). \tag{77}$$

Indeed, inserting this expression equation (59):

$$\begin{cases} \hat{V} = \hat{m} = \frac{\alpha}{1+V} \\ \hat{q} = \alpha \frac{\rho + q - 2m}{(1+V)^2} \end{cases} , \qquad \begin{cases} V = \frac{1}{d} \sum_{i=1}^d \frac{\omega_i}{\lambda + \hat{V}\omega_i} \\ q = \frac{1}{d} \sum_{i=1}^d \frac{\hat{q}\omega_i^2 + \theta_{0i}^2 \omega_i^3 \hat{m}^2}{(\lambda + \hat{V}\omega_i)^2} \\ m = \frac{\hat{m}}{d} \sum_{i=1}^d \frac{\omega_i^2 \theta_{0i}^2}{\lambda + \hat{V}\omega_i} \end{cases} . \tag{78}$$

and making a change of variables $\hat{q} \leftarrow \hat{q}\frac{d^2}{n}$, $\hat{m} \leftarrow \hat{m}\frac{d}{n}$, $\hat{V} \leftarrow \hat{q}\frac{d}{n}$, $\rho \leftarrow d\rho$, $m \leftarrow dm$, $q \leftarrow dq$, $\lambda \leftarrow d\lambda$ we recover exactly the self-consistent equations of [32] for the performance of kernel ridge regression directly from our equations. Moreover, our model allow to generalise this discussion to more involved kernel tasks such as kernel logistic regression and support vector machines.

# B Rigorous proof of the Main result

This section presents the core technical result of this paper in its full generality, along with the required assumptions and its complete proof. For technical reasons, variables different than the ones appearing in the replica calculation are introduced. The proof is nonetheless presented in a self-contained way and the relation with the replica variables are given in appendix C, eq.(280). We start by reminding the formulation of the problem. Consider the matrices $U \in \mathbb{R}^{n \times p}$ of concatenated vectors $\mathbf{u}$ used by the teacher and $\mathcal{V} \in \mathbb{R}^{n \times d}$ the corresponding one for the student. The estimator may now be defined using potentially non-separable functions:

$$\hat{\mathbf{w}} = \arg\min_{\mathbf{w} \in \mathbb{R}^d} \left[ g\left( \frac{1}{\sqrt{d}} \mathcal{V} \mathbf{w}, \mathbf{y} \right) + r(\mathbf{w}) \right], \tag{79}$$

where the function $g : \mathbb{R}^n \to \mathbb{R}$. The training and generalization errors are reminded as:

$$\mathcal{E}_{\text{train}}(\mathbf{w}) \equiv \frac{1}{n} \mathbb{E} \left[ g\left( \frac{1}{\sqrt{d}} \mathcal{V} \mathbf{w}, \mathbf{y} \right) + r(\mathbf{w}) \right] \tag{80}$$

$$\mathcal{E}_{\text{gen}}(\mathbf{w}) \equiv \mathbb{E} \left[ \hat{g}(\hat{f}(\mathbf{v}_{\text{new}}^\top \mathbf{w}), y_{\text{new}}) \right] \equiv \mathbb{E} \left[ \hat{g}\left( \hat{f}(\mathbf{v}_{\text{new}}^\top \mathbf{w}), \boldsymbol{f}_0(\mathbf{u}_{\text{new}}^\top \boldsymbol{\theta}_0) \right) \right]. \tag{81}$$

Intuitively, the variables $\mathbf{u}_{\text{new}}^\top \boldsymbol{\theta}_0$ and $\mathbf{v}_{\text{new}}^\top \mathbf{w}$ will play a key role in the analysis. Given an instance of $\boldsymbol{\theta}_0$ and $\mathbf{w}$, the tuple $\left( \frac{1}{\sqrt{p}} \mathbf{u}_{\text{new}}^\top \boldsymbol{\theta}_0, \frac{1}{\sqrt{d}} \mathbf{v}_{\text{new}}^\top \mathbf{w} \right)$ is a bivariate Gaussian with covariance:

$$\begin{bmatrix} \frac{1}{p} \boldsymbol{\theta}_0^\top \Psi \boldsymbol{\theta}_0 & \frac{1}{\sqrt{dp}} (\Phi^\top \boldsymbol{\theta}_0)^\top \mathbf{w} \\ \frac{1}{\sqrt{dp}} (\Phi^\top \boldsymbol{\theta}_0)^\top \mathbf{w} & \frac{1}{d} \mathbf{w}^\top \Omega \mathbf{w} \end{bmatrix}. \tag{82}$$

We thus define the following overlaps, that will play a fundamental role in the analysis:

$$\rho = \frac{1}{p} \boldsymbol{\theta}_0^\top \Psi \boldsymbol{\theta}_0, \quad m = \frac{1}{\sqrt{dp}} (\Phi^\top \boldsymbol{\theta}_0)^\top \mathbf{w}, \quad q = \frac{1}{d} \mathbf{w}^\top \Omega \mathbf{w}, \quad \chi = \frac{1}{d} \boldsymbol{\theta}_0^\top \Phi \Omega^{-1} \Phi^\top \boldsymbol{\theta}_0. \tag{83}$$

Note that here, we will not introduce the spectral decomposition 7 as it will not simplify the expressions as in the $l_2$ case. The representations are mathematically equivalent nonetheless. Our main result is that the distribution of the estimator $\hat{\mathbf{w}}$ can be exactly computed in the weak sense from the solution to six scalar fixed point equations with a unique solution.

## B.1 Necessary assumptions

We start with a list of the necessary assumptions for the most generic version of the result to hold. We also briefly discuss how they are relevant in a supervised machine learning context.

**(A1)** The vector $\boldsymbol{\theta}_0$ is pulled from any given distribution $p_{\boldsymbol{\theta}_0} \in \mathbb{R}^p$ (this includes deterministic vectors with bounded norm), and is independent of the matrices U and $\mathcal{V}$. Additionally, the signal is non-vanishing and has finite squared norm, i.e. the following holds almost surely:

$$\lim_{p \to \infty} 0 < \mathbb{E} \left[ \frac{\boldsymbol{\theta}_0^\top \boldsymbol{\theta}_0}{p} \right] < +\infty \tag{84}$$

**(A2)** The covariance matrices verify:

$$(\Psi, \Omega) \in \mathbb{S}_p^{++} \times \mathbb{S}_d^{++}, \quad \Omega - \Phi^\top \Psi^{-1} \Phi \succeq 0 \tag{85}$$

The spectral distributions of the matrices $\Phi, \Psi$ and $\Omega$ converge to distributions such that the overlaps defined by equation (83) are well-defined. Additionally, the maximum singular values of the covariance matrices are bounded with high probability when $n, p, d \to \infty$.

**(A3)** The functions $r$ and $g$ are proper, lower semi-continuous, convex functions. Additionally, we assume that the cost function $r + g$ is coercive, i.e.:

$$\lim_{\|\mathbf{w}\|_2 \to +\infty} (r + g)(\mathbf{w}) = +\infty \tag{86}$$

and that the following scaling condition holds : for all $n, d \in \mathbb{N}, \mathbf{z} \in \mathbb{R}^n$ and any constant $c > 0$, there exist finite, positive constants $C_1, C_2, C_3$, such that, for any standard normal random vectors $\mathbf{h} \in \mathbb{R}^d$ and $\mathbf{g} \in \mathbb{R}^n$:

$$\|\mathbf{z}\|_2 \leqslant c\sqrt{n} \implies \sup_{\mathbf{x} \in \partial g(\mathbf{z})} \|\mathbf{x}\|_2 \leqslant C_1 \sqrt{n}, \quad \frac{1}{d} \mathbb{E} [r(\mathbf{h})] < +\infty, \quad \frac{1}{n} \mathbb{E} [g(\mathbf{g})] < +\infty \tag{87}$$

**(A4)** The random elements of the function $f_0$ are independent of the matrices $U$ and $\mathcal{V}$. Additionally the following limit exists and is finite

$$\lim_{n\to\infty} \mathbb{E}\left[\frac{1}{n}f_0(U\boldsymbol{\theta}_0)^\top f_0(U\boldsymbol{\theta}_0)\right] < +\infty$$

**(A5)** When we send the dimensions $n, p, d$ to infinity, they grow with finite ratios $\alpha = n/d$, $\gamma = p/d$.

**(A6)** **Additional assumptions for linear finite sample size rates** : the teacher vector $\boldsymbol{\theta}_0$ has sub-Gaussian one dimensional marginals. The functions $r, g, \phi_1, \phi_2$ are pseudo-Lipschitz of finite order. The eigenvalues of the covariance matrices are bounded with probability one.

**(A7)** **Additional assumptions for exponential finite sample size rates**: all of the above, and the loss function $g$ is separable and pseudo-Lipschitz of order 2, the regularisation is either a ridge or a Lipschitz function, the functions $\phi_1, \phi_2$ are respectively separable, pseudo-Lipschitz of order 2, and a square or Lipschitz function.

The first assumption (A1) ensures that the teacher distribution is non-vanishing. The positive definiteness in (A2) means the covariance matrices of the blocks U and V are well-specified. Note that the cross-correlation matrix $\Phi$ can have singular values equal to zero. The assumption about the limiting spectral distribution is essentially a summability condition which is immediately verified if the limiting spectral distributions have compact support, a common case. The scaling assumptions from (A3) are natural as they imply that non-diverging inputs result in non-diverging outputs in the functions $f$ and $g$, as well as the sub-differentials. Similar scaling assumptions are encountered in proofs such as [27]. They also allow to show Gaussian concentration of Moreau envelopes, as we will see in Lemma 5. The *coercivity* assumption is verified in most common machine learning setups : any convex loss with ridge regularisation, or any convex loss that is bounded below with a coercive regularisation (LASSO, elastic-net,...), see Corollary 11.15 from [68]. Assumption (A4) is a classical assumption of teacher-student setups, where any correlation between the teacher and the student is modeled by the covariance matrices and not by the label generating function $f_0$. The summability condition ensures generalization error is well-defined for squared performance measures. Finally, (A5) is the typical *high-dimensional* limit used in statistical physics of learning, random matrix theory and a large recent body of work in high-dimensional statistical learning.

## B.2 Main theorem

First, let's define quantities and a scalar optimization problem that will be used to state the asymptotic behaviour of (2-3):

**Definition 1.** *(Scalar potentials/replica free energy) Define the following functions of the scalar variables $\tau_1 > 0, \tau_2 > 0, \kappa \geqslant 0, \eta \geqslant 0, \nu, m$:*

$$\mathcal{L}_g(\tau_1, \kappa, m, \eta) = \frac{1}{n}\mathbb{E}\left[\mathcal{M}_{\frac{\tau_1}{\kappa}g(\cdot,\mathbf{y})}\left(\frac{m}{\sqrt{\rho}}\mathbf{s} + \eta\mathbf{h}\right)\right], \tag{88}$$

$$\mathcal{L}_r(\tau_2, \eta, \nu, \kappa) = \frac{1}{d}\mathbb{E}\left[\mathcal{M}_{\frac{\eta}{\tau_2}r(\Omega^{-1/2}\cdot)}\left(\frac{\eta}{\tau_2}(\nu\mathbf{t} + \kappa\mathbf{g})\right)\right],$$

*where $\mathbf{s}, \mathbf{h} \sim \mathcal{N}(0, I_n)$ and $\mathbf{g} \sim \mathcal{N}(0, I_d)$ are random vectors independent of the other quantities, $\mathbf{t} = \Omega^{-1/2}\Phi^\top\boldsymbol{\theta}_0$, $\mathbf{y} = \boldsymbol{f}_0\left(\sqrt{\rho}\mathbf{s}\right)$, and $\mathcal{M}$ denotes the Moreau envelope of a target function.*

*From these quantities define the following potential:*

$$\mathcal{E}(\tau_1, \tau_2, \kappa, \eta, \nu, m) = \frac{\kappa\tau_1}{2} - \frac{\eta\tau_2}{2} + m\nu\sqrt{\gamma} - \frac{\tau_2}{2\eta}\frac{m^2}{\rho}$$

$$- \frac{\eta}{2\tau_2}(\nu^2\chi + \kappa^2) + \alpha\mathcal{L}_g(\tau_1, \kappa, m, \eta) + \mathcal{L}_r(\tau_2, \eta, \nu, \kappa). \tag{89}$$

*Under Assumption (B.1), the previously defined quantities all admit finite limits when $n, p, d \to \infty$.*

*Proof*: This follows directly from Lemma 5.

The next lemma characterizes important properties of the "potential" function $\mathcal{E}(\tau_1, \tau_2, \kappa, \eta, \nu, m)$:

**Lemma 1.** *(Geometry and minimizers of $\mathcal{E}$) The function $\mathcal{E}(\tau_1, \tau_2, \kappa, \eta, \nu, m)$ is jointly convex in $(m, \eta, \tau_1)$ and jointly concave in $(\nu, \kappa, \tau_2)$, and the optimization problem*

$$\min_{m,\eta,\tau_1} \max_{\kappa,\nu,\tau_2} \mathcal{E}(\tau_1, \tau_2, \kappa, \eta, \nu, m) \tag{90}$$

*has a unique solution $(\tau_1^*, \tau_2^*, \kappa^*, \eta^*, \nu^*, m^*)$ on $dom(\mathcal{E})$.*

*Proof*: see Appendix B.5. The optimality condition of problem (90) yields the set of self-consistent fixed point equations given in Lemma 12 of Appendix B. Finally, define the following variables:

$$\mathbf{w}^* = \Omega^{-1/2} \text{prox}_{\frac{\eta^*}{\tau_2^*} r(\Omega^{-1/2}.)} \left( \frac{\eta^*}{\tau_2^*}(\nu^* \mathbf{t} + \kappa^* \mathbf{g}) \right), \quad \mathbf{z}^* = \text{prox}_{\frac{\tau_1^*}{\kappa^*} g(\cdot, \mathbf{y})} \left( \frac{m^*}{\sqrt{\rho}} \mathbf{s} + \eta^* \mathbf{h} \right). \tag{91}$$

where prox denotes the proximal operator. With these definitions, we can now state our main result:

**Theorem 4.** *(Training loss and generalisation error) Under Assumption (B.1), there exist constants $C, c, c' > 0$ such that, for any optimal solution $\hat{\mathbf{w}}$ to (3), the training loss and generalisation error defined by equation verify, for any $0 < \epsilon < c'$:*

$$\mathbb{P}\left( |\mathcal{E}_{\text{train}}(\hat{\mathbf{w}}) - \mathcal{E}_{\text{train}}^*| \geqslant \epsilon \right) \leqslant \frac{C}{\epsilon} e^{-cn\epsilon^2}, \tag{92}$$

$$\mathbb{P}\left( \left| \mathcal{E}_{\text{gen}}(\hat{\mathbf{w}}) - \mathbb{E}_{\omega,\xi}\left[ \hat{g}(f_0(\omega), \hat{f}(\xi)) \right] \right| \geqslant \epsilon \right) \leqslant \frac{C}{\epsilon} e^{-cn\epsilon^2},$$

*where $\mathcal{E}_{train}^*$ is defined as follows:*

$$\mathcal{E}_{\text{train}}^* = \frac{1}{n} \mathbb{E}\left[ g\left(\mathbf{z}^*, \mathbf{y}\right) \right] + \frac{1}{\alpha d} \mathbb{E}\left[ r\left(\mathbf{w}^*\right) \right], \tag{93}$$

*and the random variables $(\omega, \xi)$ are jointly Gaussian with covariance*

$$(\omega, \xi) \sim \mathcal{N}\left( 0, \begin{bmatrix} \rho & m^* \\ m^* & q^* \end{bmatrix} \right), \quad q^* = (\eta^*)^2 + \frac{(m^*)^2}{\rho}. \tag{94}$$

*Proof*: see Appendix B.6. Note that the regularisation may be removed to evaluate the training loss. A more generic result, aiming directly at the estimator $\hat{\mathbf{w}}$, can also be stated:

**Theorem 5.** *Under Assumption (B.1), for any optimal solution $\hat{\mathbf{w}}$ to (3), denote $\hat{\mathbf{z}} = \frac{1}{\sqrt{d}}\mathcal{V}\hat{\mathbf{w}}$. Then, there exist constants $C, c, c' > 0$ such that, for any Lipschitz function $\phi_1 : \mathbb{R}^d \to \mathbb{R}$, and separable, pseudo-Lipschitz function $\phi_2 : \mathbb{R}^n \to \mathbb{R}$ and any $0 < \epsilon < c'$:*

$$\mathbb{P}\left( \left| \phi_1(\frac{\hat{\mathbf{w}}}{\sqrt{d}}) - \mathbb{E}\left[ \phi_1\left( \frac{\mathbf{w}^*}{\sqrt{d}} \right) \right] \right| \geqslant \epsilon \right) \leqslant \frac{C}{\epsilon^2} e^{-cn\epsilon^4}, \tag{95}$$

$$\mathbb{P}\left( \left| \phi_2(\frac{\hat{\mathbf{z}}}{\sqrt{n}}) - \mathbb{E}\left[ \phi_2\left( \frac{\mathbf{z}^*}{\sqrt{n}} \right) \right] \right| \geqslant \epsilon \right) \leqslant \frac{C}{\epsilon^2} e^{-cn\epsilon^4}. \tag{96}$$

*Proof*: see Appendix B.6. Concentration still holds for a larger class of functions $\phi_{1,2}$, but exponential rates are lost. This is discussed in Appendix B.1.

### B.3 Theoretical toolbox

Here we remind a few known results that are used throughout the proof. We also provide proofs of useful, straightforward consequences of theses results that do not appear explicitly in the literature for completeness.

#### B.3.1 A Gaussian comparison theorem

We start with the Convex Gaussian Min-max Theorem, as presented in [27], which is a tight version of an inequality initially derived in [20].

**Theorem 6.** *(CGMT) Let $\mathbf{G} \in \mathbb{R}^{m \times n}$ be an i.i.d. standard normal matrix and $\mathbf{g} \in \mathbb{R}^m$, $\mathbf{h} \in \mathbb{R}^n$ two i.i.d. standard normal vectors independent of one another. Let $\mathcal{S}_\mathbf{w}, \mathcal{S}_\mathbf{u}$ be two compact sets such that*

$\mathcal{S}_{\mathbf{w}} \subset \mathbb{R}^n$ and $\mathcal{S}_{\mathbf{u}} \subset \mathbb{R}^m$. *Consider the two following optimization problems for any continuous $\psi$ on* $\mathcal{S}_{\mathbf{w}} \times \mathcal{S}_{\mathbf{u}}$ :

$$\mathbf{C}(\mathbf{G}) := \min_{\mathbf{w} \in \mathcal{S}_{\mathbf{w}}} \max_{\mathbf{u} \in \mathcal{S}_{\mathbf{u}}} \mathbf{u}^{\top} \mathbf{G} \mathbf{w} + \psi(\mathbf{w}, \mathbf{u}), \tag{97}$$

$$\mathcal{C}(\mathbf{g}, \mathbf{h}) := \min_{\mathbf{w} \in \mathcal{S}_{\mathbf{w}}} \max_{\mathbf{u} \in \mathcal{S}_{\mathbf{u}}} \|\mathbf{w}\|_2 \mathbf{g}^{\top} \mathbf{u} + \|\mathbf{u}\|_2 \mathbf{h}^{\top} \mathbf{w} + \psi(\mathbf{w}, \mathbf{u}) \tag{98}$$

*then the following holds:*

1. *For all $c \in \mathbb{R}$:*
$$\mathbb{P}(\mathbf{C}(\mathbf{G}) < c) \leqslant 2\mathbb{P}(\mathcal{C}(\mathbf{g}, \mathbf{h}) \leqslant c)$$

2. *Further assume that $\mathcal{S}_{\mathbf{w}}, \mathcal{S}_{\mathbf{u}}$ are convex sets and $\psi$ is convex-concave on $\mathcal{S}_{\mathbf{w}} \times \mathcal{S}_{\mathbf{u}}$. Then, for all $c \in \mathbb{R}$,*
$$\mathbb{P}(\mathbf{C}(\mathbf{G}) > c) \leqslant 2\mathbb{P}(\mathcal{C}(\mathbf{g}, \mathbf{h}) \geqslant c)$$

*In particular, for all $\mu \in \mathbb{R}, t > 0, \mathbb{P}(|\mathbf{C}(\mathbf{G}) - \mu| > t) \leqslant 2\mathbb{P}(|\mathcal{C}(\mathbf{g}, \mathbf{h}) - \mu| \geqslant t)$.*

Following [27], we will say that any reformulation of a target problem matching the form of (97) is an acceptable primary optimization problem (PO), and the corresponding form (98) is an acceptable auxiliary problem (AO). The main idea of this approach is to study the asymptotic properties of the (PO) by studying the simpler (AO).

### B.3.2 Proximal operators and Moreau envelopes : differentials and useful functions

Here we remind the definition and some important properties of Moreau envelopes and proximal operators, key elements of convex analysis. Other properties will be used throughout the proof but at less crucial stages, thus we don't remind them explicitly. Our main reference for these properties will be [68].
Consider a closed, proper function $f$ such that dom(f)$\subset \mathbb{R}^n$. Its Moreau envelope and proximal operator are respectively defined by :

$$\mathcal{M}_{\tau f}(\mathbf{x}) = \min_{\mathbf{z} \in \text{dom}(f)} \{f(\mathbf{z}) + \frac{1}{2\tau} \|\mathbf{x} - \mathbf{z}\|_2^2\}, \quad \text{prox}_{\tau f}(\mathbf{x}) = \arg\min_{\mathbf{z} \in \text{dom}(f)} \{f(\mathbf{z}) + \frac{1}{2\tau} \|\mathbf{x} - \mathbf{z}\|_2^2\} \tag{99}$$

As reminded in [27], the Moreau envelope is jointly convex in $(\tau, \mathbf{x})$ and differentiable almost everywhere, with gradients:

$$\nabla_{\mathbf{x}} \mathcal{M}_{\tau f}(\mathbf{x}) = \frac{1}{\tau}(\mathbf{x} - \text{prox}_{\tau f}(\mathbf{x})) \tag{100}$$

$$\frac{\partial}{\partial \tau} \mathcal{M}_{\tau f}(\mathbf{x}) = -\frac{1}{2\tau^2} \left\| \mathbf{x} - \text{prox}_{\tau f}(\mathbf{x}) \right\|_2^2 \tag{101}$$

We remind that $\text{prox}_{\tau f}(\mathbf{x})$ is the unique point which solves the strongly convex optimization problem defining the Moreau envelope, i.e.:

$$\mathcal{M}_{\tau f}(\mathbf{x}) = f(\text{prox}_{\tau f}(\mathbf{x})) + \frac{1}{2\tau} \left\| \mathbf{x} - \text{prox}_{\tau f}(\mathbf{x}) \right\|_2^2 \tag{102}$$

We also remind the definition of order k pseudo-Lipschitz function.

**Definition 2.** *Pseudo-Lipschitz function For $k \in \mathbb{N}^*$ and any $n, m \in \mathbb{N}^*$, a function $\phi : \mathbb{R}^n \to \mathbb{R}^m$ is called a* pseudo-Lipschitz *of order k if there exists a constant $L(k)$ such that for any $\mathbf{x}, \mathbf{y} \in \mathbb{R}^n$,*

$$\|\phi(\mathbf{x}) - \phi(\mathbf{y})\|_2 \leqslant L(k) \left(1 + (\|\mathbf{x}\|_2)^{k-1} + (\|\mathbf{y}\|_2)^{k-1}\right) \|\mathbf{x} - \mathbf{y}\|_2 \tag{103}$$

We now give some further properties that will be helpful throughout the proof.

**Lemma 2.** *(Moreau envelope of pseudo-Lipschitz function) Consider a proper, lower-semicontinuous, convex, pseudo-Lipschitz function $f : \mathbb{R}^n \to \mathbb{R}$ of order k. Then its Moreau envelope is also pseudo-Lipschitz of order k.*

*Proof of Lemma 2*: For any $\mathbf{x}, \mathbf{y}$ in $\mathrm{dom}(f)$, we have, using the pseudo-Lipschitz property:

$$\left| f(\mathrm{prox}_{\tau f}(\mathbf{x})) - f(\mathrm{prox}_{\tau f}(\mathbf{y})) \right| \leqslant L(k) \left( 1 + \left( \left\| \mathrm{prox}_{\tau f}(\mathbf{x}) \right\|_2 \right)^{k-1} + \left( \left\| \mathrm{prox}_{\tau f}(\mathbf{y}) \right\|_2 \right)^{k-1} \right)$$
$$\left\| \mathrm{prox}_{\tau f}(\mathbf{x}) - \mathrm{prox}_{\tau f}(\mathbf{y}) \right\|_2$$
$$\leqslant L(k) \left( 1 + (\|\mathbf{x}\|_2)^{k-1} + (\|\mathbf{y}\|_2)^{k-1} \right) \|\mathbf{x} - \mathbf{y}\|_2 \qquad (104)$$

where the second line follows immediately with the same constant $L(k)$ owing to the firm-nonexpansiveness of the proximal operator. Furthermore

$$\left\| \mathbf{x} - \mathrm{prox}_{\tau f}(\mathbf{x}) \right\|_2^2 - \left\| \mathbf{y} - \mathrm{prox}_{\tau f}(\mathbf{y}) \right\|_2^2 =$$
$$\tau \left| \partial f(\mathrm{prox}_{\tau f}(\mathbf{x})) + \partial f(\mathrm{prox}_{\tau f}(\mathbf{y})) \right| \left| (\mathbf{x} - \mathrm{prox}_{\tau f}(\mathbf{x}) - \mathbf{y} + \mathrm{prox}_{\tau f}(\mathbf{y})) \right|$$
$$\leqslant \tau \left\| \partial f(\mathrm{prox}_{\tau f}(\mathbf{x})) + \partial f(\mathrm{prox}_{\tau f}(\mathbf{y})) \right\|_2 \left\| (\mathbf{x} - \mathrm{prox}_{\tau f}(\mathbf{x}) - \mathbf{y} + \mathrm{prox}_{\tau f}(\mathbf{y})) \right\|_2 \qquad (105)$$

due to the pseudo-Lipschitz property, one has

$$\partial f(\mathrm{prox}_{\tau f}(\mathbf{x})) \leqslant L(k) \left( 1 + 2 \left\| \mathrm{prox}_{\tau f}(\mathbf{x}) \right\|_2^{k-1} \right) \qquad (106)$$

This, along with the firm-nonexpansiveness of $\mathrm{Id} - \mathrm{prox}$, concludes the proof. $\qquad\square$

**Lemma 3.** *(Useful functions) For any $\mathbf{x} \in \mathbb{R}^n, \tau > 0, \theta \in \mathbb{R}$ and any proper, convex lower semi-continuous function $f$, define the following functions:*

$$h_1 : \mathbb{R} \to \mathbb{R}$$
$$\theta \mapsto \mathbf{x}^T \mathit{prox}_{\tau f(.)}(\theta \mathbf{x}) \qquad (107)$$
$$h_2 : \mathbb{R} \to \mathbb{R}$$
$$\tau \mapsto \frac{1}{2\tau^2} \left\| \mathbf{x} - \mathit{prox}_{\tau f(.)}(\mathbf{x}) \right\|_2^2 \qquad (108)$$
$$h_3 : \mathbb{R} \to \mathbb{R}$$
$$\tau \mapsto \left\| \mathit{prox}_{\frac{f}{\tau}(.)} \left( \frac{\mathbf{x}}{\tau} \right) \right\|_2^2 \qquad (109)$$
$$h_4 : \mathbb{R} \to \mathbb{R}$$
$$\tau \mapsto \left\| \mathbf{x} - \mathit{prox}_{\tau f}(\mathbf{x}) \right\|_2^2 \qquad (110)$$

*$h_1$ is nondecreasing, and $h_2, h_3, h_4$ are nonincreasing.*

*Proof of Lemma 3*: For any $\theta, \tilde{\theta} \in \mathbb{R}$:

$$(\theta - \tilde{\theta})(h_1(\theta) - h_1(\tilde{\theta})) = (\theta \mathbf{x} - \tilde{\theta} \mathbf{x})^\top \left( \mathrm{prox}_{\tau f(.)}(\theta \mathbf{x}) - \mathrm{prox}_{\tau f(.)}(\tilde{\theta} \mathbf{x}) \right)$$
$$\geqslant \left\| \mathrm{prox}_{\tau f(.)}(\theta \mathbf{x}) - \mathrm{prox}_{\tau f(.)}(\tilde{\theta} \mathbf{x}) \right\|_2^2$$
$$\geqslant 0 \qquad (111)$$

where the inequality comes from the firm non-expansiveness of the proximal operator. Thus $h_1$ is nondecreasing.

Since the Moreau envelope $\mathcal{M}_{\tau f}(\mathbf{x})$ is convex in $\tau$, we have, for any $\tau, \tilde{\tau}$ in $\mathbb{R}_{++}$

$$(\tau - \tilde{\tau}) \left( \frac{\partial}{\partial \tau} \mathcal{M}_{\tau f}(\mathbf{x}) - \frac{\partial}{\partial \tilde{\tau}} \mathcal{M}_{\tilde{\tau} f}(\mathbf{x}) \right) \geqslant 0, \qquad \Longleftrightarrow \qquad (\tau - \tilde{\tau}) \left( h_2(\tilde{\tau}) - h_2(\tau) \right) \geqslant 0 \qquad (112)$$

which implies that $h_2$ is non-increasing.
Using the Moreau decomposition, see e.g. [68], we have:

$$h_2(\tau) = \frac{1}{2\tau^2} \left\| \mathbf{x} - \left( \mathbf{x} - \tau \mathrm{prox}_{\frac{f^*}{\tau}} \left( \frac{\mathbf{x}}{\tau} \right) \right) \right\|_2^2 = \left\| \mathrm{prox}_{\frac{f^*}{\tau}} \left( \frac{\mathbf{x}}{\tau} \right) \right\|_2^2 \qquad (113)$$

which is a nonincreasing function of $\tau$. Since $f$ is convex, we can restart this short process with the conjugate of $f$ to obtain the desired result. Thus $h_3$ is nonincreasing and $(\tau - \tilde{\tau})(h_3(\tau) - h_3(\tilde{\tau})) \leqslant 0$. Moving to $h_4$, proving that it is nonincreasing is equivalent to proving that the following function is increasing

$$h_5(\tau) = \text{prox}_{\tau f}(\mathbf{x})^\top \left(2\mathbf{x} - \text{prox}_{\tau f}(\mathbf{x})\right) \tag{114}$$

using the Moreau decomposition again

$$h_5(\tau) = \left(\mathbf{x} - \tau \text{prox}_{\frac{f^*}{\tau}}\left(\frac{\mathbf{x}}{\tau}\right)\right)^\top \left(\mathbf{x} + \tau \text{prox}_{\frac{f^*}{\tau}}\left(\frac{\mathbf{x}}{\tau}\right)\right) \tag{115}$$

then, for any $\tau, \tilde{\tau}$ in $\mathbb{R}_{++}$:

$$(\tau - \tilde{\tau})(h_5(\tau) - h_5(\tilde{\tau})) = (\tau - \tilde{\tau}) \left(\tilde{\tau}^2 \left\|\text{prox}_{\frac{f^*}{\tilde{\tau}}}\left(\frac{\mathbf{x}}{\tilde{\tau}}\right)\right\|_2^2 - \tau^2 \left\|\text{prox}_{\frac{f^*}{\tau}}\left(\frac{\mathbf{x}}{\tau}\right)\right\|_2^2\right) \tag{116}$$

separating the cases $\tau \leqslant \tilde{\tau}$ and $\tau \geqslant \tilde{\tau}$, and using the result on $h_3$ then gives the desired result. $\square$
The following inequality is similar to one that appeared in one-dimensional form in [27].

**Lemma 4.** *(A useful inequality) For any proper, lower semi-continuous convex function $f$, any $\mathbf{x}, \tilde{\mathbf{x}}$ in $dom(f)$, and any $\gamma, \tilde{\gamma} \in \mathbb{R}_{++}$, the following holds:*

$$\left(prox_{\tilde{\gamma}f}(\tilde{\mathbf{x}}) - prox_{\gamma f}(\mathbf{x})\right)^\top \left(\frac{\tilde{\mathbf{x}}}{\tilde{\gamma}} - \frac{\mathbf{x}}{\gamma} - \frac{1}{2}\left(\frac{1}{\tilde{\gamma}} - \frac{1}{\gamma}\right)\left(prox_{\tilde{\gamma}f}(\tilde{\mathbf{x}}) + prox_{\gamma f}(\mathbf{x})\right)\right)$$

$$\geqslant \left(\frac{1}{2\tilde{\gamma}} + \frac{1}{2\gamma}\right) \left\|\left(prox_{\tilde{\gamma}f}(\tilde{\mathbf{x}}) - prox_{\gamma f}(\mathbf{x})\right)\right\|_2^2 \tag{117}$$

*Proof of Lemma 4* : the subdifferential of a proper convex function is a monotone operator, thus:

$$\left(\text{prox}_{\tilde{\gamma}f}(\tilde{\mathbf{x}}) - \text{prox}_{\gamma f}(\mathbf{x})\right)^\top \left(\partial f(\text{prox}_{\tilde{\gamma}f}(\tilde{\mathbf{x}})) - \partial f(\text{prox}_{\gamma f}(\mathbf{x}))\right) \geqslant 0 \tag{118}$$

additionally, $\text{prox}_{\gamma f}(\mathbf{x}) = (\text{Id} + \gamma \partial \text{f})^{-1}(\mathbf{x})$, hence:

$$\partial f(\text{prox}_{\tilde{\gamma}f}(\tilde{\mathbf{x}})) - \partial f(\text{prox}_{\gamma f}(\mathbf{x})) = \left(\frac{\tilde{\mathbf{x}}}{\tilde{\gamma}} - \frac{\mathbf{x}}{\gamma} - \frac{1}{\tilde{\gamma}}\text{prox}_{\tilde{\gamma}}(\tilde{\mathbf{x}}) + \frac{1}{\gamma}\text{prox}_{\gamma f}(\mathbf{x})\right)$$

$$= \frac{\tilde{\mathbf{x}}}{\tilde{\gamma}} - \frac{\mathbf{x}}{\gamma} - \frac{1}{\tilde{\gamma}}\text{prox}_{\tilde{\gamma}}(\tilde{\mathbf{x}}) + \frac{1}{\gamma}\text{prox}_{\gamma f}(\mathbf{x}) - \frac{1}{2}\left(\frac{1}{\tilde{\gamma}} - \frac{1}{\gamma}\right)\left(\text{prox}_{\tilde{\gamma}f}(\tilde{\mathbf{x}}) + \text{prox}_{\gamma f}(\mathbf{x})\right)$$

$$+ \frac{1}{2}\left(\frac{1}{\tilde{\gamma}} - \frac{1}{\gamma}\right)\left(\text{prox}_{\tilde{\gamma}f}(\tilde{\mathbf{x}}) + \text{prox}_{\gamma f}(\mathbf{x})\right)$$

$$= \left(\frac{\tilde{\mathbf{x}}}{\tilde{\gamma}} - \frac{\mathbf{x}}{\gamma} - \frac{1}{2}\left(\frac{1}{\tilde{\gamma}} - \frac{1}{\gamma}\right)\left(\text{prox}_{\tilde{\gamma}f}(\tilde{\mathbf{x}}) + \text{prox}_{\gamma f}(\mathbf{x})\right)\right) - \left(\frac{1}{2\tilde{\gamma}} + \frac{1}{2\gamma}\right)\left(\text{prox}_{\tilde{\gamma}f}(\tilde{\mathbf{x}}) - \text{prox}_{\gamma f}(\mathbf{x})\right) \tag{119}$$

which gives the desired inequality. $\square$

### B.3.3 Useful concentration of measure elements

We begin by reminding the Gaussian-Poincaré inequality, see e.g. [69].

**Proposition 1.** *(Gaussian Poincaré inequality)*
*Let $\mathbf{g} \in \mathbb{R}^n$ be a $\mathcal{N}(0, I_n)$ random vector. Then for any continuous, weakly differentiable $\varphi$, there exists a constant $c$ such that:*

$$Var[\varphi(\mathbf{g})] \leqslant c\, \mathbb{E}\left[\|\nabla \varphi(\mathbf{g})\|_2^2\right] \tag{120}$$

We now use this previous result to show Gaussian concentration of Moreau envelopes of appropriately scaled convex functions.

**Lemma 5.** *(Gaussian concentration of Moreau envelopes)*
*Consider a proper, convex function $f : \mathbb{R}^n \to \mathbb{R}$ verifying the scaling conditions of Assumptions B.1 and let $\mathbf{g} \in \mathbb{R}^n$ be a standard normal random vector. Then, for any parameter $\tau > 0$ and any $\epsilon > 0$, there exists a constant $c$ such that the following holds:*

$$\mathbb{P}\left(\left|\frac{1}{n}\mathcal{M}_{\tau f(.)}(\mathbf{g}) - \mathbb{E}\left[\frac{1}{n}\mathcal{M}_{\tau f(.)}(\mathbf{g})\right]\right| \geqslant \epsilon\right) \leqslant \frac{c}{n\tau^2\epsilon^2} \tag{121}$$

*Proof of Lemma 5*:

We start by showing that the Moreau envelope of a proper, convex function $f : \mathbb{R}^n \to \mathbb{R}$ verifying the scaling conditions of Assumptions B.1 is integrable with respect to the Gaussian measure. Using the convexity of the optimization problem defining the Moreau envelope, and the fact that $f$ is proper, there exists $\mathbf{z}_0 \in \mathbb{R}^n$ and a finite constant $\mathcal{K}$ such that :

$$\frac{1}{n}\mathcal{M}_{\tau f(.)}(\mathbf{g}) \leqslant \frac{1}{n}f(\mathbf{z}_0) + \frac{1}{2n\tau}\|\mathbf{z}_0 - \mathbf{g}\|_2^2$$
$$\leqslant \mathcal{K} + \frac{1}{2n\tau}\|\mathbf{z}_0 - \mathbf{g}\|_2^2 \tag{122}$$

where the second line is integrable under a multivariate Gaussian measure. Then, using Proposition 1, we get:

$$\mathrm{Var}\left[\frac{1}{n}\mathcal{M}_{\tau f(.)}(\mathbf{g})\right] \leqslant \frac{c}{n^2}\mathbb{E}\left[\left\|\nabla_{\mathbf{z}}\mathcal{M}_{\tau f(.)}(\mathbf{g})\right\|_2^2\right] \tag{123}$$

$$= \frac{c}{n^2}\mathbb{E}\left[\left\|\frac{1}{\tau}\left(\mathbf{z} - \mathrm{prox}_{\tau f}(\mathbf{g})\right)\right\|_2^2\right] \tag{124}$$

Using Proposition 12.27 and Corollary 4.3 from [68], $\mathbf{g} \to \mathbf{z} - \mathrm{prox}_{\tau f}(\mathbf{g})$ is firmly non-expansive and:

$$\left\|\mathbf{g} - \mathrm{prox}_{\tau f}(\mathbf{g})\right\|_2^2 \leqslant \langle \mathbf{g}|\mathbf{g} - \mathrm{prox}_{\tau f}(\mathbf{g})\rangle \quad \text{which implies} \tag{125}$$

$$\left\|\mathbf{g} - \mathrm{prox}_{\tau f}(\mathbf{g})\right\|_2^2 \leqslant \|\mathbf{g}\|_2^2 \quad \text{using the Cauchy-Schwarz inequality} \tag{126}$$

then

$$\mathrm{Var}\left[\frac{1}{n}\mathcal{M}_{\tau f(.)}(\mathbf{g})\right] \leqslant \frac{c}{n^2\tau^2}\mathbb{E}\left[\|\mathbf{g}\|_2^2\right] = \frac{c}{n\tau^2} \tag{127}$$

Chebyshev's inequality then gives, for any $\epsilon > 0$:

$$\mathbb{P}\left(\left|\frac{1}{n}\mathcal{M}_{\tau f(.)}(\mathbf{g}) - \mathbb{E}\left[\frac{1}{n}\mathcal{M}_{\tau f(.)}(\mathbf{g})\right]\right| \geqslant \epsilon\right) \leqslant \frac{c}{n\tau^2\epsilon^2} \tag{128}$$

$\square$

Gaussian concentration of pseudo-Lipschitz functions of finite order can also be proven using the Gaussian Poincaré inequality to yield a bound similar to the one obtained for Moreau envelopes. We thus give the result without proof:

**Lemma 6.** *(Concentration of pseudo-Lipschitz functions) Consider a pseudo-Lipschitz function of finite order $k$, $f : \mathbb{R}^n \to \mathbb{R}$. Then for any vector $\mathbf{g} \sim \mathcal{N}(0, I_n)$ and any $\epsilon > 0$, there exists a constant $C(k) > 0$ such that*

$$\mathbb{P}\left(\left|f(\frac{\mathbf{g}}{\sqrt{n}}) - \mathbb{E}\left[f(\frac{\mathbf{g}}{\sqrt{n}})\right]\right| \geqslant \epsilon\right) \leqslant \frac{L^2(k)C(k)}{n\epsilon^2} \tag{129}$$

We now cite an exponential concentration lemma for separable, pseudo-Lipschitz functions of order 2, taken from [70].

**Lemma 7.** *(Lemma B.5 from [70]) Consider a separable, pseudo-Lipschitz function of order 2, $f : \mathbb{R}^n \to \mathbb{R}$. Then for any vector $\mathbf{g} \sim \mathcal{N}(0, I_n)$ and any $\epsilon > 0$, there exists constants $C, c, c' > 0$ such that*

$$\mathbb{P}\left(\left|\frac{1}{n}f(\mathbf{g}) - \mathbb{E}\left[\frac{1}{n}f(\mathbf{g})\right]\right| \geqslant c'\epsilon\right) \leqslant Ce^{-cn\epsilon^2} \tag{130}$$

*where it is understood that $f(\mathbf{g}) = \sum_{i=1}^n f(g_i)$.*

## B.4 Determining a candidate primary problem, auxiliary problem and its solution.

We start with a reformulation of the problem (2-3) in order to obtain an acceptable primary problem in the framework of Theorem 6. Partitioning the Gaussian distribution, we can rewrite the matrices U and $\mathcal{V}$ in the following way, introducing the standard normal vector:

$$\begin{bmatrix} \mathbf{a} \\ \mathbf{b} \end{bmatrix} \in \mathbb{R}^{p+d} \sim \mathcal{N}(0, \mathrm{I}_{p+d}) \tag{131}$$

We can then rewrite the vectors $\mathbf{u}, \mathbf{v}$ and matrices $U, \mathcal{V}$ as:

$$\mathbf{u} = \Psi^{1/2}\mathbf{a}, \quad U = A\Psi^{1/2} \tag{132}$$

$$\mathbf{v} = \Phi^\top \Psi^{-1/2}\mathbf{a} + \left(\Omega - \Phi^\top \Psi^{-1}\Phi\right)^{1/2}\mathbf{b}, \quad \mathcal{V} = A\Psi^{-1/2}\Phi + B\left(\Omega - \Phi^\top \Psi^{-1}\Phi\right)^{1/2} \tag{133}$$

where the matrices $A$ and $B$ have independent standard normal entries and are independent of $\boldsymbol{\theta}_0$. The learning problem then becomes equivalent to :

$$\text{Generate labels according to :} \quad \mathbf{y} = f_0\left(\frac{1}{\sqrt{p}}A\Psi^{1/2}\boldsymbol{\theta}_0\right) \tag{134}$$

$$\text{Learn according to :} \quad \underset{\mathbf{w}}{\arg\min}\, g\left(\frac{1}{\sqrt{d}}\left(A\Psi^{-1/2}\Phi + B\left(\Omega - \Phi^\top \Psi^{-1}\Phi\right)^{1/2}\right)\mathbf{w}, \mathbf{y}\right) + r(\mathbf{w}) \tag{135}$$

We are then interested in the optimal cost of the following problem

$$\min_{\mathbf{w}} \frac{1}{d}\left[g\left(\frac{1}{\sqrt{d}}\left(A\Psi^{-1/2}\Phi + B\left(\Omega - \Phi^\top \Psi^{-1}\Phi\right)^{1/2}\right)\mathbf{w}, \mathbf{y}\right) + r(\mathbf{w})\right] \tag{136}$$

Introducing the auxiliary variable $\mathbf{z}$:

$$\min_{\mathbf{w}} g\left(\frac{1}{\sqrt{d}}\left(A\Psi^{-1/2}\Phi + B\left(\Omega - \Phi^\top \Psi^{-1}\Phi\right)^{1/2}\right)\mathbf{w}, \mathbf{y}\right) + r(\mathbf{w}) \tag{137}$$

$$\iff \min_{\mathbf{w},\mathbf{z}} g(\mathbf{z}, \mathbf{y}) + r(\mathbf{w})$$

$$\text{s.t. } \mathbf{z} = \frac{1}{\sqrt{d}}\left(A\Psi^{-1/2}\Phi + B\left(\Omega - \Phi^\top \Psi^{-1}\Phi\right)^{1/2}\right)\mathbf{w} \tag{138}$$

Introducing the corresponding Lagrange multiplier $\boldsymbol{\lambda} \in \mathbb{R}^n$ and using strong duality, the problem is equivalent to :

$$\min_{\mathbf{w},\mathbf{z}} \max_{\boldsymbol{\lambda}} \boldsymbol{\lambda}^\top \frac{1}{\sqrt{d}}\left(A\Psi^{-1/2}\Phi + B\left(\Omega - \Phi^\top \Psi^{-1}\Phi\right)^{1/2}\right)\mathbf{w} - \boldsymbol{\lambda}^\top\mathbf{z} + g(\mathbf{z}, \mathbf{y}) + r(\mathbf{w}) \tag{139}$$

In the remainder of the proof, the preceding cost function will be denoted

$$\mathbf{C}(\mathbf{w}, \mathbf{z}) = \max_{\boldsymbol{\lambda}} \boldsymbol{\lambda}^\top \frac{1}{\sqrt{d}}\left(A\Psi^{-1/2}\Phi + B\left(\Omega - \Phi^\top \Psi^{-1}\Phi\right)^{1/2}\right)\mathbf{w} - \boldsymbol{\lambda}^\top\mathbf{z} + g(\mathbf{z}, \mathbf{y}) + r(\mathbf{w}) \tag{140}$$

such that the problem reads $\min_{\mathbf{w},\mathbf{z}} \mathbf{C}(\mathbf{w}, \mathbf{z})$. Theorem 6 requires working with compact feasibility sets. Adopting similar approaches to the ones from [27, 35], the next lemma shows that the optimization problem (139) can be equivalently recast as one over compact sets.

**Lemma 8.** *(Compactness of feasibility set) Let $\mathbf{w}^*, \mathbf{z}^*, \boldsymbol{\lambda}^*$ be optimal in (139). Then there exists positive constants $C_{\mathbf{w}}, C_{\mathbf{z}}$ and $C_{\boldsymbol{\lambda}}$ such that*

$$\mathbb{P}\left(\|\mathbf{w}^*\|_2 \leqslant C_{\mathbf{w}}\sqrt{d}\right) \xrightarrow[d\to\infty]{P} 1, \ \mathbb{P}\left(\|\mathbf{z}^*\|_2 \leqslant C_{\mathbf{z}}\sqrt{n}\right) \xrightarrow[n\to\infty]{P} 1, \ \mathbb{P}\left(\|\boldsymbol{\lambda}^*\|_2 \leqslant C_{\boldsymbol{\lambda}}\sqrt{n}\right) \xrightarrow[n\to\infty]{P} 1 \tag{141}$$

*Proof of Lemma 8*: consider the initial minimisation problem:

$$\hat{\mathbf{w}} = \underset{\mathbf{w}\in\mathbb{R}^d}{\arg\min}\, g\left(\frac{1}{\sqrt{d}}\mathcal{V}\mathbf{w}, \mathbf{y}\right) + r(\mathbf{w}) \tag{142}$$

From assumption (A3), the cost function $g + r$ is coercive, proper and lower semi-continuous. Since it is proper, there exists $\mathbf{w}_0 \in \mathbb{R}^d$ such that $g\left(\frac{1}{\sqrt{d}}\mathcal{V}\mathbf{w}, \mathbf{y}\right) + r(\mathbf{w}) \in \mathbb{R}$. The coercivity implies that there exists $\eta \in ]0, +\infty[$ such that, for every $\mathbf{w} \in \mathbb{R}^d$ satisfying $\|\mathbf{w} - \mathbf{w}_0\| \geqslant \eta$, $g\left(\frac{1}{\sqrt{d}}\mathcal{V}\mathbf{w}, \mathbf{y}\right) + r(\mathbf{w}) \geqslant g\left(\frac{1}{\sqrt{d}}\mathcal{V}\mathbf{w}_0, \mathbf{y}\right) + r(\mathbf{w}_0)$. Let $S = \{\mathbf{w} \in \mathbb{R}^d | \|\mathbf{w} - \mathbf{w}_0\| \leqslant \eta\}$. Then $S \cap \mathbb{R}^d \neq \emptyset$ and $S$ is compact. Then, there exists $\mathbf{w}^* \in S$ such that $g\left(\frac{1}{\sqrt{d}}\mathcal{V}\mathbf{w}^*, \mathbf{y}\right) + r(\mathbf{w}^*) = \inf_{\mathbf{w} \in S} g\left(\frac{1}{\sqrt{d}}\mathcal{V}\mathbf{w}, \mathbf{y}\right) + r(\mathbf{w}) \leqslant g\left(\frac{1}{\sqrt{d}}\mathcal{V}\mathbf{w}_0, \mathbf{y}\right) + f(\mathbf{w}_0)$. Thus $g\left(\frac{1}{\sqrt{d}}\mathcal{V}\mathbf{w}^*, \mathbf{y}\right) + r(\mathbf{w}^*) \in \inf_{\mathbf{w} \in \mathbb{R}^d} g\left(\frac{1}{\sqrt{d}}\mathcal{V}\mathbf{w}, \mathbf{y}\right) + r(\mathbf{w})$ and the set of minimisers is bounded. Closure is immediately checked by considering a sequence of minimisers converging to $\mathbf{w}^*$.

We conclude that the set of minimisers of problem (142) is a non-empy compact set. Then there exists a constant $C_{\mathbf{w}}$ independent of the dimension $d$, such that:

$$\|\mathbf{w}\|_2 \leqslant C_{\mathbf{w}}\sqrt{d} \tag{143}$$

Now consider the equivalent formulation of problem (142):

$$\min_{\mathbf{w}, \mathbf{z}} \max_{\boldsymbol{\lambda}} \boldsymbol{\lambda}^\top \frac{1}{\sqrt{d}}\mathcal{V}\mathbf{w} - \boldsymbol{\lambda}^\top \mathbf{z} + g(\mathbf{z}, \mathbf{y}) + r(\mathbf{w}) \tag{144}$$

Its optimality condition reads :

$$\nabla_{\boldsymbol{\lambda}} : \frac{1}{\sqrt{d}}\mathcal{V}\mathbf{w} = \mathbf{z}, \qquad \nabla_{\mathbf{z}} : \boldsymbol{\lambda} \in \partial g(\mathbf{z}, \mathbf{y}), \qquad \nabla_{\mathbf{w}} : \frac{1}{\sqrt{d}}\mathcal{V}^\top \boldsymbol{\lambda} \in \partial r(\mathbf{w}) \tag{145}$$

The optimality condition in $\boldsymbol{\lambda}$ gives:

$$\|\mathbf{z}\|_2 \leqslant \left\|\frac{1}{\sqrt{d}}\mathcal{V}\right\|_{op} \|\mathbf{w}\|_2$$

$$\leqslant \left\|\frac{1}{\sqrt{d}}\left(A\Psi^{-1/2}\Phi + B\left(\Omega - \Phi^\top\Psi^{-1}\Phi\right)^{1/2}\right)\right\|_{op} \|\mathbf{w}\|_2$$

$$\leqslant \left[\left\|\Psi^{-1/2}\Phi\right\|_{op}\left\|\frac{1}{\sqrt{d}}A\right\|_{op} + \left\|\left(\Omega - \Phi^\top\Psi^{-1}\Phi\right)^{1/2}\right\|_{op}\left\|\frac{1}{\sqrt{d}}B\right\|_{op}\right] \|\mathbf{w}\|_2 \tag{146}$$

According to assumption (A2), the operator norms of the matrices involving the covariance matrices are bounded with high probability and using known results on random matrices, see e.g. [71], the operator norms of $\frac{1}{\sqrt{d}}A$ and $\frac{1}{\sqrt{d}}B$ are bounded by finite constants with high probability when the dimensions go to infinity. Thus there exists a constant $C_{\mathbf{z}}$ also independent of d such that:

$$\mathbb{P}\left(\|\mathbf{z}\|_2 \leqslant C_{\mathbf{z}}\sqrt{n}\right) \xrightarrow[n \to \infty]{P} 1 \tag{147}$$

Finally, the scaling condition from assumption (A3) directly shows that there exists a constant $C_{\boldsymbol{\lambda}}$ such that

$$\mathbb{P}\left(\|\boldsymbol{\lambda}\|_2 \leqslant C_{\boldsymbol{\lambda}}\sqrt{n}\right) \xrightarrow[n \to \infty]{P} 1 \tag{148}$$

This concludes the proof of Lemma 8. $\qquad\square$

Defining the sets $\mathcal{S}_{\mathbf{w}} = \{\mathbf{w} \in \mathbb{R}^d | \|\mathbf{w}\|_2 \leqslant C_{\mathbf{w}}\sqrt{d}\}, \mathcal{S}_{\mathbf{z}} = \{\mathbf{z} \in \mathbb{R}^n | \|\mathbf{z}\|_2 \leqslant C_{\mathbf{z}}\sqrt{n}\}$ and $\mathcal{S}_{\boldsymbol{\lambda}} = \{\boldsymbol{\lambda} \in \mathbb{R}^n | \|\boldsymbol{\lambda}\|_2 \leqslant C_{\boldsymbol{\lambda}}\sqrt{n}\}$, the optimization problem can now be reduced to:

$$\min_{\mathbf{w} \in \mathcal{S}_{\mathbf{w}}, \mathbf{z} \in \mathcal{S}_{\mathbf{z}}} \max_{\boldsymbol{\lambda} \in \mathcal{S}_{\boldsymbol{\lambda}}} \boldsymbol{\lambda}^\top \frac{1}{\sqrt{d}}\left(A\Psi^{-1/2}\Phi + B\left(\Omega - \Phi^\top\Psi^{-1}\Phi\right)^{1/2}\right)\mathbf{w} - \boldsymbol{\lambda}^\top \mathbf{z} + g(\mathbf{z}, \mathbf{y}) + r(\mathbf{w}) \tag{149}$$

The rest of this section can then be summarized by the following lemma, the proof of which shows how to find an acceptable (PO) for problem (149), the corresponding (AO) and how to reduce the (AO) to a scalar optimization problem. At this point we will assume the teacher vector $\boldsymbol{\theta}_0$ is deterministic, and relax this assumption in paragraph B.7. For this reason we do not add it to the initial list of assumptions in section B.1.

**Lemma 9.** *(Scalar equivalent problem) In the framework of Theorem 6, acceptable (AO)s of problem (149) can be reduced to the following scalar optimization problems*

$$\text{For } \boldsymbol{\theta}_0 \notin Ker(\Phi^\top) : \max_{\kappa, \nu, \tau_2} \min_{m, \eta, \tau_1} \mathcal{E}_n(\tau_1, \tau_2, \kappa, \eta, \nu, m) \tag{150}$$

$$\text{For } \boldsymbol{\theta}_0 \in Ker(\Phi^\top) : \max_{\kappa, \tau_2} \min_{\eta, \tau_1} \mathcal{E}_n^0(\tau_1, \tau_2, \kappa, \eta) \tag{151}$$

*where*

$$\mathcal{E}_n(\tau_1, \tau_2, \kappa, \eta, \nu, m) = \frac{\kappa\tau_1}{2} - \frac{\eta\tau_2}{2} + m\nu\sqrt{\gamma} - \frac{\tau_2}{2\eta}\frac{m^2}{\rho}$$
$$- \frac{\eta}{2\tau_2 d}(\nu\mathbf{v} + \kappa\Omega^{1/2}\mathbf{g})^\top \Omega^{-1}(\nu\mathbf{v} + \kappa\Omega^{1/2}\mathbf{g}) - \kappa\mathbf{g}^\top \left(\Sigma^{1/2} - \Omega^{1/2}\right)\frac{m\sqrt{\gamma}}{\|\tilde{\mathbf{v}}\|_2^2}\mathbf{v}$$
$$+ \frac{1}{d}\mathcal{M}_{\frac{\tau_1}{\kappa}g(.,\mathbf{y})}\left(\frac{m}{\sqrt{\rho}}\mathbf{s} + \eta\mathbf{h}\right) + \frac{1}{d}\mathcal{M}_{\frac{\eta}{\tau_2}r(\Omega^{-1/2}.)}\left(\frac{\eta}{\tau_2}\left(\nu\Omega^{-1/2}\tilde{\mathbf{v}} + \kappa\mathbf{g}\right)\right), \tag{152}$$

$$\mathcal{E}_n^0(\tau_1, \tau_2, \kappa, \nu) = -\frac{\eta\tau_2}{2} + \frac{\kappa\tau_1}{2} + \frac{1}{d}\mathcal{M}_{\frac{\tau_1}{\kappa}g(.,\mathbf{y})}(\eta\mathbf{h}) + \frac{1}{d}\mathcal{M}_{\frac{\eta}{\tau_2}f(\Omega^{-1/2}.)}(\frac{\eta}{\tau_2}\kappa\mathbf{g}) - \frac{\eta}{2\tau_2 d}\kappa^2\mathbf{g}^\top\mathbf{g} \tag{153}$$

*and*

$$\Sigma = \Omega - \frac{\tilde{\mathbf{v}}\tilde{\mathbf{v}}^T}{\rho p} \quad \tilde{\mathbf{v}} = \Phi^T\boldsymbol{\theta}_0 \quad \rho = \frac{1}{p}\boldsymbol{\theta}_0^\top\Psi\boldsymbol{\theta}_0 \tag{154}$$

*Proof of Lemma 9*: We need to find an i.i.d. Gaussian matrix independent from the rest of the problem in order to use Theorem 6. We thus decompose the mixing matrix A by taking conditional expectations w.r.t. $\mathbf{y}$, which amounts to conditioning on a linear subset of the Gaussian space generated by A. Dropping the feasibility sets for confort of notation in the following lines:

$$\min_{\mathbf{w},\mathbf{z}} \max_{\boldsymbol{\lambda}} \boldsymbol{\lambda}^\top \frac{1}{\sqrt{d}}\left(\left(\mathbb{E}\left[A|\mathbf{y}\right] + A - \mathbb{E}\left[A|\mathbf{y}\right]\right)\Psi^{-1/2}\Phi + B\left(\Omega - \Phi^\top\Psi^{-1}\Phi\right)^{1/2}\right)\mathbf{w}$$
$$- \boldsymbol{\lambda}^\top\mathbf{z} + g(\mathbf{z},\mathbf{y}) + r(\mathbf{w}) \tag{155}$$

$$\Longleftrightarrow \min_{\mathbf{w},\mathbf{z}} \max_{\boldsymbol{\lambda}} \boldsymbol{\lambda}^\top \frac{1}{\sqrt{d}}\left(\left(\mathbb{E}\left[A|A\Psi^{1/2}\boldsymbol{\theta}_0\right] + A - \mathbb{E}\left[A|A\Psi^{1/2}\boldsymbol{\theta}_0\right]\right)\Psi^{-1/2}\Phi\right.$$
$$\left. + B\left(\Omega - \Phi^\top\Psi^{-1}\Phi\right)^{1/2}\right)\mathbf{w} - \boldsymbol{\lambda}^\top\mathbf{z} + g(\mathbf{z},\mathbf{y}) + r(\mathbf{w}) \tag{156}$$

Conditioning in Gaussian spaces amounts to doing orthogonal projections. Denoting $\tilde{\boldsymbol{\theta}}_0 = \Psi^{1/2}\boldsymbol{\theta}_0$ and $\tilde{A}$ a copy of A independent of $\mathbf{y}$, the minimisation problem then becomes:

$$\min_{\mathbf{w},\mathbf{z}} \max_{\boldsymbol{\lambda}} \boldsymbol{\lambda}^\top \frac{1}{\sqrt{d}}\left(\left(A\mathbf{P}_{\tilde{\boldsymbol{\theta}}_0} + \tilde{A}\mathbf{P}_{\tilde{\boldsymbol{\theta}}_0}^\perp\right)\Psi^{-1/2}\Phi + B\left(\Omega - \Phi^\top\Psi^{-1}\Phi\right)^{1/2}\right)\mathbf{w} - \boldsymbol{\lambda}^\top\mathbf{z} + g(\mathbf{z},\mathbf{y}) + r(\mathbf{w})$$

$$\tag{157}$$

$$\Longleftrightarrow \min_{\mathbf{w},\mathbf{z}} \max_{\boldsymbol{\lambda}} \boldsymbol{\lambda}^\top \frac{1}{\sqrt{d}}A\mathbf{P}_{\tilde{\boldsymbol{\theta}}_0}\Psi^{-1/2}\Phi\mathbf{w} + \boldsymbol{\lambda}^\top \frac{1}{\sqrt{d}}\tilde{A}\mathbf{P}_{\tilde{\boldsymbol{\theta}}_0}^\perp\Psi^{-1/2}\Phi\mathbf{w} + \boldsymbol{\lambda}^\top \frac{1}{\sqrt{d}}B\left(\Omega - \Phi^\top\Psi^{-1}\Phi\right)^{1/2}\mathbf{w}$$
$$- \boldsymbol{\lambda}^\top\mathbf{z} + g(\mathbf{z},\mathbf{y}) + r(\mathbf{w}) \tag{158}$$

$$\Longleftrightarrow \min_{\mathbf{w},\mathbf{z}} \max_{\boldsymbol{\lambda}} \boldsymbol{\lambda}^\top \frac{1}{\sqrt{d}}\mathbf{s}\frac{\tilde{\boldsymbol{\theta}}_0^\top}{\left\|\tilde{\boldsymbol{\theta}}_0\right\|_2}\Psi^{-1/2}\Phi\mathbf{w} + \boldsymbol{\lambda}^\top \frac{1}{\sqrt{d}}\tilde{A}\mathbf{P}_{\tilde{\boldsymbol{\theta}}_0}^\perp\Psi^{-1/2}\Phi\mathbf{w} + \boldsymbol{\lambda}^\top \frac{1}{\sqrt{d}}B\left(\Omega - \Phi^\top\Psi^{-1}\Phi\right)^{1/2}\mathbf{w}$$
$$- \boldsymbol{\lambda}^\top\mathbf{z} + g(\mathbf{z},\mathbf{y}) + r(\mathbf{w}) \tag{159}$$

where we used $\mathbf{P}_{\tilde{\boldsymbol{\theta}}_0} = \frac{\tilde{\boldsymbol{\theta}}_0\tilde{\boldsymbol{\theta}}_0^\top}{\|\tilde{\boldsymbol{\theta}}_0\|_2^2}$ and $\mathbf{s} = A\frac{\tilde{\boldsymbol{\theta}}_0}{\|\tilde{\boldsymbol{\theta}}_0\|_2}$. Knowing that $\tilde{A}, B$ are independent standard Gaussian matrices, and independent from $\mathbf{A}, \mathbf{y}, f_0$, we can rewrite the problem as :

$$\min_{\mathbf{w},\mathbf{z}} \max_{\boldsymbol{\lambda}} \boldsymbol{\lambda}^\top \frac{1}{\sqrt{d}}\mathbf{s}\frac{\boldsymbol{\theta}_0^\top}{\left\|\Psi^{1/2}\boldsymbol{\theta}_0\right\|}\Phi\mathbf{w} + \boldsymbol{\lambda}^\top \frac{1}{\sqrt{d}}Z\Sigma^{1/2}\mathbf{w} - \boldsymbol{\lambda}^\top\mathbf{z} + g(\mathbf{z},\mathbf{y}) + r(\mathbf{w}) \tag{160}$$

where $\Sigma = \Phi^\top \Psi^{-1/2} \mathbf{P}_{\tilde{\theta}_0}^\perp \Psi^{-1/2} \Phi + \Omega - \Phi^\top \Psi^{-1} \Phi = \Omega - \Phi^\top \Psi^{-1/2} \mathbf{P}_{\tilde{\theta}_0} \Psi^{-1/2} \Phi$, and $Z$ is a standard Gaussian matrix independent of $\mathbf{A}, \mathbf{y}, f_0$. Recall $\rho = \frac{1}{p} \theta_0^\top \Psi \theta_0$ from the main text. Replacing with the expression of $\tilde{\theta}_0$ and letting $\tilde{\mathbf{v}} = \Phi^\top \theta_0$, we have

$$\Sigma = \Omega - \phi^\top \Psi^{-1/2} \tilde{\theta}_0 \tilde{\theta}_0^\top \Psi^{-1/2} \Phi \frac{1}{\left\| \tilde{\theta}_0 \right\|_2^2} = \Omega - \frac{\phi^\top \theta_0 \theta_0^\top \Phi}{\theta_0^\top \Psi \theta_0} \tag{161}$$

$$= \Omega - \frac{\tilde{\mathbf{v}} \tilde{\mathbf{v}}^\top}{p\rho} \tag{162}$$

The problem then becomes

$$\min_{\mathbf{w},\mathbf{z}} \max_{\boldsymbol{\lambda}} \boldsymbol{\lambda}^\top \frac{1}{\sqrt{dp}} \mathbf{s} \frac{\tilde{\mathbf{v}}^\top}{\sqrt{\rho}} \mathbf{w} + \boldsymbol{\lambda}^\top \frac{1}{\sqrt{d}} Z \Sigma^{1/2} \mathbf{w} - \boldsymbol{\lambda}^\top \mathbf{z} + g(\mathbf{z},\mathbf{y}) + r(\mathbf{w}) \tag{163}$$

Two cases must now be considered, $\theta_0 \notin \mathrm{Ker}(\phi^\top)$ and $\theta_0 \in \mathrm{Ker}(\phi^\top)$. Another possible case is $\Phi = 0_{p \times d}$, however it leads to the same steps as the case $\theta_0 \in \mathrm{Ker}(\Phi^\top)$.

**Case 1: $\theta_0 \notin \mathbf{Ker}(\Phi^\top)$**

It is tempting to invert the matrix $\Sigma^{1/2}$ to make the change of variable $\mathbf{w}_\perp = \Sigma^{1/2} \mathbf{w}$ and continue the calculation. However there is no guarantee that $\Sigma$ is invertible : it is only semi-positive definite. Taking identities everywhere gives for examples $\mathbf{P}_{\tilde{\theta}_0}^\perp$ which is non-invertible. We thus introduce an additional variable:

$$\min_{\mathbf{w},\mathbf{z},\mathbf{p}} \max_{\boldsymbol{\lambda},\boldsymbol{\mu}} \boldsymbol{\lambda}^\top \frac{1}{\sqrt{dp}} \mathbf{s} \frac{\tilde{\mathbf{v}}^\top}{\sqrt{\rho}} \mathbf{w} + \boldsymbol{\lambda}^\top \frac{1}{\sqrt{d}} Z \mathbf{p} - \boldsymbol{\lambda}^\top \mathbf{z} + g(\mathbf{z},\mathbf{y}) + r(\mathbf{w}) + \boldsymbol{\mu}^\top \left( \Sigma^{1/2} \mathbf{w} - \mathbf{p} \right) \tag{164}$$

Here the minimisation on $f$ and $g$ is linked by the bilinear form $\boldsymbol{\lambda}^\top \mathbf{s} \tilde{\mathbf{v}}^\top \mathbf{w}$. We wish to separate them in order for the Moreau envelopes to appear later on in simple fashion. To do so, we introduce the orthogonal decomposition of $\mathbf{w}$ on the direction of $\tilde{\mathbf{v}}$:

$$\mathbf{w} = \left( \mathbf{P}_{\tilde{\mathbf{v}}} + \mathbf{P}_{\tilde{\mathbf{v}}}^\perp \right) \mathbf{w} = \frac{\tilde{\mathbf{v}}^\top \mathbf{w}}{\|\tilde{\mathbf{v}}\|_2^2} \tilde{\mathbf{v}} + \mathbf{P}_{\tilde{\mathbf{v}}}^\perp \mathbf{w}$$

$$= \frac{\tilde{\mathbf{v}}^\top \mathbf{w}}{\|\tilde{\mathbf{v}}\|_2^2} \tilde{\mathbf{v}} + \mathbf{w}_\perp \quad \text{where } \mathbf{w}_\perp \perp \tilde{\mathbf{v}}$$

$$= \frac{m\sqrt{dp}}{\|\tilde{\mathbf{v}}\|_2^2} \tilde{\mathbf{v}} + \mathbf{w}_\perp \quad \text{where } m = \frac{1}{\sqrt{dp}} \tilde{\mathbf{v}}^\top \mathbf{w} \tag{165}$$

where the parameter $m$ corresponds to the one defined in (83). This gives the following, after introducing the scalar Lagrange multiplier $\nu \in \mathbb{R}$ to enforce the constraint $\mathbf{w}_\perp \perp \tilde{\mathbf{v}}$. Note that several methods can be used to express the orthogonality constraint, as in e.g. [35], but the one chosen here allows to complete the proof and match the replica prediction. Reintroducing the normalization, we then have the equivalent form for (136):

$$\min_{m,\mathbf{w}_\perp,\mathbf{z},\mathbf{p}} \max_{\boldsymbol{\lambda},\boldsymbol{\mu},\nu} \frac{1}{d} \left[ \boldsymbol{\lambda}^\top \frac{m}{\sqrt{\rho}} \mathbf{s} + \boldsymbol{\lambda}^\top \frac{1}{\sqrt{d}} Z \mathbf{m} - \boldsymbol{\lambda}^\top \mathbf{z} + g(\mathbf{z},\mathbf{y}) + r \left( \frac{m\sqrt{dp}}{\|\tilde{\mathbf{v}}\|_2^2} \tilde{\mathbf{v}} + \mathbf{w}_\perp \right) \right.$$

$$\left. + \boldsymbol{\mu}^\top \left( \Sigma^{1/2} \left( \frac{m\sqrt{dp}}{\|\tilde{\mathbf{v}}\|_2^2} \tilde{\mathbf{v}} + \mathbf{w}_\perp \right) - \mathbf{p} \right) - \nu \tilde{\mathbf{v}}^\top \mathbf{w}_\perp \right] \tag{166}$$

A follow-up of the previous equations shows that the feasibility set now reads :

$$\mathcal{S}_{m,\mathbf{w}_\perp,\mathbf{z},\mathbf{p},\boldsymbol{\lambda},\boldsymbol{\mu},\nu} = \left\{ m \in \mathbb{R}, \mathbf{w}_\perp \in \mathbb{R}^{d-1}, \mathbf{z} \in \mathbb{R}^n, \mathbf{p} \in \mathbb{R}^d, \boldsymbol{\lambda} \in \mathbb{R}^n, \boldsymbol{\mu} \in \mathbb{R}^d, \nu \in \mathbb{R} \right|$$

$$\left. \sqrt{m^2 + \frac{\|\mathbf{w}_\perp\|_2^2}{d}} \leqslant C_{\mathbf{w}}, \|\mathbf{z}\|_2 \leqslant C_{\mathbf{z}} \sqrt{n}, \|\mathbf{p}\|_2 \leqslant \sigma_{max}(\Sigma^{1/2}) C_{\mathbf{w}} \sqrt{d}, \|\boldsymbol{\lambda}\|_2 \leqslant C_{\boldsymbol{\lambda}} \sqrt{n} \right\}$$

$$\tag{167}$$

where the boundedness of $\|\mathbf{p}\|_2$ follows immediately from the assumptions on the covariance matrices and Lemma 8. We denote $\mathcal{S}_{\mathbf{p}} = \{\mathbf{p} \in \mathbb{R}^d | \|\mathbf{p}\|_2 \leqslant C_{\mathbf{p}}\}$ for some constant $C_{\mathbf{p}} \geqslant \sigma_{max}(\Sigma^{1/2})C_{\mathbf{w}}$.

The set $\mathcal{S}_{\mathbf{p}} \times \mathcal{S}_{\boldsymbol{\lambda}}$ is compact and the matrix $Z$ is independent of all other random quantities of the problem, thus problem (166) is an acceptable (PO). We can now write the auxiliary optimization problem (AO) corresponding to the primary one (166), dropping the feasibility sets again for convenience:

$$
\min_{m,\mathbf{w}_\perp,\mathbf{z},\mathbf{p}} \max_{\boldsymbol{\lambda},\boldsymbol{\mu},\nu} \frac{1}{d}\left[\boldsymbol{\lambda}^\top \frac{m}{\sqrt{\rho}}\mathbf{s} + \frac{1}{\sqrt{d}}\|\boldsymbol{\lambda}\|_2 \mathbf{g}^\top \mathbf{p} + \frac{1}{\sqrt{d}}\|\mathbf{p}\|_2 \mathbf{h}^\top \boldsymbol{\lambda} - \boldsymbol{\lambda}^\top \mathbf{z} + g(\mathbf{z},\mathbf{y}) + r\left(\frac{m\sqrt{dp}}{\|\tilde{\mathbf{v}}\|_2^2}\tilde{\mathbf{v}} + \mathbf{w}_\perp\right)\right.
$$
$$
\left. + \boldsymbol{\mu}^\top\left(\Sigma^{1/2}\left(\frac{m\sqrt{dp}}{\|\tilde{\mathbf{v}}\|_2^2}\tilde{\mathbf{v}} + \mathbf{w}_\perp\right) - \mathbf{p}\right) - \nu\tilde{\mathbf{v}}^\top \mathbf{w}_\perp\right] \tag{168}
$$

We now turn to the simplification of this problem.

The variable $\boldsymbol{\lambda}$ only appears in linear terms, we can thus directly optimize over its direction, introducing the positive scalar variable $\kappa = \|\boldsymbol{\lambda}\|_2/\sqrt{d}$:

$$
\min_{m,\mathbf{w}_\perp,\mathbf{z},\mathbf{p}} \max_{\kappa,\boldsymbol{\mu},\nu} \frac{1}{d}\left[\kappa\mathbf{g}^\top\mathbf{p} + \kappa\left\|\frac{m}{\sqrt{\rho}}\sqrt{d}\mathbf{s} + \|\mathbf{p}\|_2\mathbf{h} - \sqrt{d}\mathbf{z}\right\|_2 + g(\mathbf{z},\mathbf{y}) + r\left(\frac{m\sqrt{dp}}{\|\tilde{\mathbf{v}}\|_2^2}\tilde{\mathbf{v}} + \mathbf{w}_\perp\right)\right.
$$
$$
\left. + \boldsymbol{\mu}^\top\left(\Sigma^{1/2}\left(\frac{m\sqrt{dp}}{\|\tilde{\mathbf{v}}\|_2^2}\tilde{\mathbf{v}} + \mathbf{w}_\perp\right) - \mathbf{p}\right) - \nu\tilde{\mathbf{v}}^\top\mathbf{w}_\perp\right] \tag{169}
$$

The previous expression may not be convex-concave because of the term $\|\mathbf{p}\|_2\mathbf{h}$. However, it was shown in [27] that the order of the min and max can still be inverted in this case, because of the convexity of the original problem. As the proof would be very similar, we do not reproduce it. Inverting the max-min order and performing the linear optimization on $\mathbf{p}$ with $\eta = \|\mathbf{p}\|_2/\sqrt{d}$:

$$
\max_{\kappa,\boldsymbol{\mu},\nu} \min_{m,\mathbf{w}_\perp,\mathbf{z},\eta} \left\{-\frac{\eta}{\sqrt{d}}\|\boldsymbol{\mu} + \kappa\mathbf{g}\|_2 + \frac{\kappa}{\sqrt{d}}\left\|\frac{m}{\sqrt{\rho}}\mathbf{s} + \eta\mathbf{h} - \mathbf{z}\right\|_2 + \right.
$$
$$
\left. + \frac{1}{d}\left[g(\mathbf{z},\mathbf{y}) + r\left(\frac{m\sqrt{dp}}{\|\tilde{\mathbf{v}}\|_2^2}\tilde{\mathbf{v}} + \mathbf{w}_\perp\right) + \boldsymbol{\mu}^\top\Sigma^{1/2}\left(\frac{m\sqrt{dp}}{\|\tilde{\mathbf{v}}\|_2^2}\tilde{\mathbf{v}} + \mathbf{w}_\perp\right) - \nu\tilde{\mathbf{v}}^\top\mathbf{w}_\perp\right]\right\} \tag{170}
$$

using the following representation of the norm, as in [27], for any vector $t$, $\|t\|_2 = \min_{\tau>0} \frac{\tau}{2} + \frac{\|t\|_2^2}{2\tau}$:

$$
\max_{\kappa,\boldsymbol{\mu},\nu,\tau_2} \min_{m,\mathbf{w}_\perp,\mathbf{z},\eta,\tau_1} \left\{\frac{\kappa\tau_1}{2} - \frac{\eta\tau_2}{2} - \frac{\eta}{2\tau_2 d}\|\boldsymbol{\mu} + \kappa\mathbf{g}\|_2^2 + \frac{\kappa}{2\tau_1 d}\left\|\frac{m}{\sqrt{\rho}}\mathbf{s} + \eta\mathbf{h} - \mathbf{z}\right\|_2^2\right.
$$
$$
\left. + \frac{1}{d}\left[g(\mathbf{z},\mathbf{y}) + r\left(\frac{m\sqrt{dp}}{\|\tilde{\mathbf{v}}\|_2^2}\tilde{\mathbf{v}} + \mathbf{w}_\perp\right) + \boldsymbol{\mu}^\top\Sigma^{1/2}\left(\frac{m\sqrt{dp}}{\|\tilde{\mathbf{v}}\|_2^2}\tilde{\mathbf{v}} + \mathbf{w}_\perp\right) - \nu\tilde{\mathbf{v}}^\top\mathbf{w}_\perp\right]\right\} \tag{171}
$$

performing the minimisation over $\mathbf{z}$ and recognizing the Moreau envelope of $g(.,\mathbf{y})$:

$$
\max_{\kappa,\boldsymbol{\mu},\nu,\tau_2} \min_{m,\mathbf{w}_\perp,\eta,\tau_1} \left\{\frac{\kappa\tau_1}{2} - \frac{\eta\tau_2}{2} + \frac{1}{d}\mathcal{M}_{\frac{\tau_1}{\kappa}g(.,\mathbf{y})}\left(\frac{m}{\sqrt{\rho}}\mathbf{s} + \beta\mathbf{h}\right) - \frac{\eta}{2\tau_2 d}\|\boldsymbol{\mu} + \kappa\mathbf{g}\|_2^2\right.
$$
$$
\left. + \frac{1}{d}\left[r\left(\frac{m\sqrt{dp}}{\|\tilde{\mathbf{v}}\|_2^2}\tilde{\mathbf{v}} + \mathbf{w}_\perp\right) + \boldsymbol{\mu}^\top\Sigma^{1/2}\left(\frac{m\sqrt{dp}}{\|\tilde{\mathbf{v}}\|_2^2}\tilde{\mathbf{v}} + \mathbf{w}_\perp\right) - \nu\tilde{\mathbf{v}}^\top\mathbf{w}_\perp\right]\right\} \tag{172}
$$

At this point we have a convex-concave problem. Inverting the min-max order, $\boldsymbol{\mu}$ appears in a well defined strictly convex least-square problem.

$$
\max_{\kappa,\nu,\tau_2} \min_{m,\mathbf{w}_\perp,\eta,\tau_1} \frac{\kappa\tau_1}{2} - \frac{\eta\tau_2}{2} + \frac{1}{d}\mathcal{M}_{\frac{\tau_1}{\kappa}g(.,\mathbf{y})}\left(\frac{m}{\sqrt{\rho}}\mathbf{s} + \eta\mathbf{h}\right) - \frac{\nu}{d}\tilde{\mathbf{v}}^\top\mathbf{w}_\perp + \frac{1}{d}r\left(\frac{m\sqrt{dp}}{\|\tilde{\mathbf{v}}\|_2^2}\tilde{\mathbf{v}} + \mathbf{w}_\perp\right)
$$
$$
+ \frac{1}{d}\max_{\boldsymbol{\mu}}\left\{-\frac{\eta}{2\tau_2}\|\boldsymbol{\mu} + \kappa\mathbf{g}\|_2^2 + \boldsymbol{\mu}^\top\Sigma^{1/2}\left(\frac{m\sqrt{dp}}{\|\tilde{\mathbf{v}}\|_2^2}\tilde{\mathbf{v}} + \mathbf{w}_\perp\right)\right\} \tag{173}
$$

Solving it:

$$\max_{\boldsymbol{\mu}} \left\{ -\frac{\eta}{2\tau_2} \|\boldsymbol{\mu} + \kappa \mathbf{g}\|_2^2 + \boldsymbol{\mu}^\top \Sigma^{1/2} \left( \frac{m\sqrt{dp}}{\|\tilde{\mathbf{v}}\|_2^2} \tilde{\mathbf{v}} + \mathbf{w}_\perp \right) \right\}$$

$$\boldsymbol{\mu}^* = \frac{\tau_2}{\eta} \Sigma^{1/2} \left( \frac{m\sqrt{dp}}{\|\tilde{\mathbf{v}}\|_2^2} \tilde{\mathbf{v}} + \mathbf{w}_\perp \right) - \kappa \mathbf{g}$$

with optimal cost $\dfrac{\tau_2}{2\eta} \left\| \Sigma^{1/2} \left( \dfrac{m\sqrt{dp}}{\|\tilde{\mathbf{v}}\|_2^2} \tilde{\mathbf{v}} + \mathbf{w}_\perp \right) \right\|_2^2 - \kappa \mathbf{g}^\top \Sigma^{1/2} \left( \dfrac{m\sqrt{dp}}{\|\tilde{\mathbf{v}}\|_2^2} \tilde{\mathbf{v}} + \mathbf{w}_\perp \right)$ (174)

remembering that $\Sigma = \Omega - \tilde{\mathbf{v}}\tilde{\mathbf{v}}^\top/(p\rho)$ and $\mathbf{w}_\perp \perp \tilde{\mathbf{v}}$, the optimal cost of this least-square problem simplifies to:

$$c^* = \frac{\tau_2}{2\eta} \left( \left\| \Omega^{1/2} \left( \frac{m\sqrt{dp}}{\|\tilde{\mathbf{v}}\|_2^2} \tilde{\mathbf{v}} + \mathbf{w}_\perp \right) \right\|_2^2 - \frac{m^2}{\rho} d \right) - \kappa \mathbf{g}^\top \Sigma^{1/2} \left( \frac{m\sqrt{dp}}{\|\tilde{\mathbf{v}}\|_2^2} \tilde{\mathbf{v}} + \mathbf{w}_\perp \right)$$ (175)

The (AO) then reads :

$$\max_{\kappa,\nu,\tau_2,\eta,\tau_1} \min_{m,\mathbf{w}_\perp,\eta,\tau_1} \left\{ \frac{\kappa\tau_1}{2} - \frac{\eta\tau_2}{2} + \frac{1}{d}\mathcal{M}_{\frac{\tau_1}{\kappa} g(.,\mathbf{y})} \left( \frac{m}{\sqrt{\rho}}\mathbf{s} + \eta\mathbf{h} \right) - \frac{\tau_2}{2\eta}\frac{m^2}{\rho} - \frac{\nu}{d}\tilde{\mathbf{v}}^\top \mathbf{w}_\perp \right.$$

$$\left. + \frac{1}{d}r\left( \frac{m\sqrt{dp}}{\|\tilde{\mathbf{v}}\|_2^2}\tilde{\mathbf{v}} + \mathbf{w}_\perp \right) + \frac{\tau_2}{2\eta d}\left\| \Omega^{1/2}\left( \frac{m\sqrt{dp}}{\|\tilde{\mathbf{v}}\|_2^2}\tilde{\mathbf{v}} + \mathbf{w}_\perp \right) \right\|_2^2 - \frac{\kappa}{d}\mathbf{g}^\top \Sigma^{1/2}\left( \frac{m\sqrt{dp}}{\|\tilde{\mathbf{v}}\|_2^2}\tilde{\mathbf{v}} + \mathbf{w}_\perp \right) \right\}$$ (176)

We now need to solve in $\mathbf{w}_\perp$. To do so, we can replace $r$ with its convex conjugate and solve the least-square problem in $\mathbf{w}_\perp$. This will lead to a Moreau envelope of $r^*$ in the introduced dual variable, which can be linked to the Moreau envelope of $r$ by Moreau decomposition. Intuitively, it is natural to think that the corresponding primal variable will be $\frac{m\sqrt{dp}}{\|\tilde{\mathbf{v}}\|_2^2}\tilde{\mathbf{v}} + \mathbf{w}_\perp = \mathbf{w}$ for any feasible $m, \mathbf{w}_\perp$. However, we would like to have an explicit follow-up of the variables we optimize on, as we had for the Moreau envelpe of $g$ which is defined with $\mathbf{z}$, so we prefer to introduce a slack variable $\mathbf{w}' = \frac{m\sqrt{dp}}{\|\tilde{\mathbf{v}}\|_2^2}\tilde{\mathbf{v}} + \mathbf{w}_\perp$ with corresponding dual parameter $\boldsymbol{\eta}$ to show that the (AO) can be reformulated in terms of the original variable $\mathbf{w}$. Note that the feasibility set on $\mathbf{w}'$ is almost surely compact.

$$\max_{\kappa,\nu,\tau_2,\boldsymbol{\eta}} \min_{m,\mathbf{w}_\perp,\mathbf{w}',\eta,\tau_1} \frac{\kappa\tau_1}{2} - \frac{\eta\tau_2}{2} + \frac{1}{d}\mathcal{M}_{\frac{\tau_1}{\kappa} g(.,\mathbf{y})} \left( \frac{m}{\sqrt{\rho}}\mathbf{s} + \eta\mathbf{h} \right) + \frac{1}{d}r(\mathbf{w}') - \frac{1}{d}\boldsymbol{\eta}^T\mathbf{w}' - \frac{\tau_2}{2\eta}\frac{m^2}{\rho}$$

$$- \frac{\nu}{d}\tilde{\mathbf{v}}^\top \mathbf{w}_\perp + \frac{\tau_2}{2\eta d}\left\| \Omega^{1/2}\left( \frac{m\sqrt{dp}}{\|\tilde{\mathbf{v}}\|_2^2}\tilde{\mathbf{v}} + \mathbf{w}_\perp \right) \right\|_2^2 - \frac{\kappa}{d}\mathbf{g}^\top \Sigma^{1/2}\left( \frac{m\sqrt{dp}}{\|\tilde{\mathbf{v}}\|_2^2}\tilde{\mathbf{v}} + \mathbf{w}_\perp \right) + \frac{1}{d}\boldsymbol{\eta}^\top\left( \frac{m\sqrt{dp}}{\|\tilde{\mathbf{v}}\|_2^2}\tilde{\mathbf{v}} + \mathbf{w}_\perp \right)$$ (177)

Isolating the terms depending on $\mathbf{w}_\perp$, we get a strictly convex least-square problem, remembering that $\Omega \in \mathcal{S}_d^{++}$:

$$\max_{\kappa,\nu,\tau_2,\boldsymbol{\eta}} \min_{m,\mathbf{w}_\perp,\mathbf{w}',\eta,\tau_1} \frac{\kappa\tau_1}{2} - \frac{\eta\tau_2}{2} + \frac{1}{d}\mathcal{M}_{\frac{\tau_1}{\kappa} g(.,\mathbf{y})} \left( \frac{m}{\sqrt{\rho}}\mathbf{s} + \eta\mathbf{h} \right) + \frac{1}{d}r(\mathbf{w}') - \frac{1}{d}\boldsymbol{\eta}^T\mathbf{w}' - \frac{\tau_2}{2\eta}\frac{m^2}{\rho} + \boldsymbol{\eta}^\top \frac{m\sqrt{\kappa_2}}{\|\tilde{\mathbf{v}}\|_2^2}\tilde{\mathbf{v}}$$

$$- \kappa\mathbf{g}^\top \Sigma^{1/2}\frac{m\sqrt{\gamma}}{\|\tilde{\mathbf{v}}\|_2^2}\tilde{\mathbf{v}} - \frac{\nu}{d}\tilde{\mathbf{v}}^\top\mathbf{w}_\perp + \frac{\tau_2}{2\eta d}\left\| \Omega^{1/2}\left( \frac{m\sqrt{dp}}{\|\tilde{\mathbf{v}}\|_2^2}\tilde{\mathbf{v}} + \mathbf{w}_\perp \right) \right\|_2^2 - \frac{\kappa}{d}\mathbf{g}^\top\Sigma^{1/2}\mathbf{w}_\perp + \frac{1}{d}\boldsymbol{\eta}^\top\mathbf{w}_\perp$$ (178)

$$\max_{\kappa,\nu,\tau_2,\boldsymbol{\eta}} \min_{m,\mathbf{w}',\eta,\tau_1} \frac{\kappa\tau_1}{2} - \frac{\eta\tau_2}{2} + \frac{1}{d}\mathcal{M}_{\frac{\tau_1}{\kappa} g(.,\mathbf{y})} \left( \frac{m}{\sqrt{\rho}}\mathbf{s} + \eta\mathbf{h} \right) + \frac{1}{d}r(\mathbf{w}') - \frac{1}{d}\boldsymbol{\eta}^T\mathbf{w}' - \frac{\tau_2}{2\eta}\frac{m^2}{\rho} + \boldsymbol{\eta}^\top \frac{m\sqrt{\kappa_2}}{\|\tilde{\mathbf{v}}\|_2^2}\tilde{\mathbf{v}}$$

$$- \kappa\mathbf{g}^\top \Sigma^{1/2}\frac{m\sqrt{\gamma}}{\|\tilde{\mathbf{v}}\|_2^2}\tilde{\mathbf{v}} + \frac{1}{d}\left[ \min_{\mathbf{w}_\perp} \frac{\tau_2}{2\eta}\left\| \Omega^{1/2}\left( \frac{m\sqrt{dp}}{\|\tilde{\mathbf{v}}\|_2^2}\tilde{\mathbf{v}} + \mathbf{w}_\perp \right) \right\|_2^2 - \mathbf{w}_\perp^\top\left( \kappa\Sigma^{1/2}\mathbf{g} - \boldsymbol{\eta} + \nu\tilde{\mathbf{v}} \right) \right]$$ (179)

The quantity $\mathbf{g}^\top \Sigma^{1/2} \mathbf{w}_\perp$ is a Gaussian random variable with variance $\left\| \Sigma^{1/2} \mathbf{w}_\perp \right\|_2^2 = \mathbf{w}_\perp^\top (\Omega - \tilde{\mathbf{v}} \tilde{\mathbf{v}}^\top / (p\rho)) \mathbf{w}_\perp = \mathbf{w}_\perp \Omega \mathbf{w}_\perp = \left\| \Omega^{1/2} \mathbf{w}_\perp \right\|_2^2$ using the expression of $\Sigma$ and the orthogonality of $\mathbf{w}_\perp$ with respect to $\tilde{\mathbf{v}}$. We can thus change $\Sigma^{1/2}$ for $\Omega^{1/2}$ in front of $\mathbf{w}_\perp$ combined with $\mathbf{g}$. The least-square problem, its solution and optimal cost then read:

$$\min_{\mathbf{w}_\perp} \frac{\tau_2}{2\eta} \left\| \Omega^{1/2} \left( \frac{m\sqrt{dp}}{\|\tilde{\mathbf{v}}\|_2^2} \tilde{\mathbf{v}} + \mathbf{w}_\perp \right) \right\|_2^2 - \mathbf{w}_\perp^\top \left( \kappa \Omega^{1/2} \mathbf{g} - \boldsymbol{\eta} + \nu \tilde{\mathbf{v}} \right) \tag{180}$$

$$\mathbf{w}_\perp^* = \frac{\eta}{\tau_2} \Omega^{-1} \left( \kappa \Omega^{1/2} \mathbf{g} - \boldsymbol{\eta} + \nu \mathbf{v} \right) - \frac{m\sqrt{dp}}{\|\tilde{\mathbf{v}}\|_2^2} \tilde{\mathbf{v}} \tag{181}$$

with optimal cost $-\dfrac{\eta}{2\tau_2} \left( \kappa \Omega^{1/2} \mathbf{g} - \boldsymbol{\eta} + \nu \tilde{\mathbf{v}} \right)^\top \Omega^{-1} \left( \kappa \Omega^{1/2} \mathbf{g} - \boldsymbol{\eta} + \nu \tilde{\mathbf{v}} \right) + \dfrac{m\sqrt{dp}}{\|\tilde{\mathbf{v}}\|_2^2} \tilde{\mathbf{v}}^\top \left( \kappa \Omega^{1/2} \mathbf{g} - \boldsymbol{\eta} + \nu \tilde{\mathbf{v}} \right)$

$$\tag{182}$$

replacing in the (AO) and simplifying :

$$\iff \max_{\kappa, \nu, \tau_2, \boldsymbol{\eta}} \min_{m, \mathbf{w}', \eta, \tau_1} \frac{\kappa \tau_1}{2} - \frac{\eta \tau_2}{2} + \frac{1}{d} \mathcal{M}_{\frac{\tau_1}{\kappa} g(\cdot, \mathbf{y})} \left( \frac{m}{\sqrt{\rho}} \mathbf{s} + \eta \mathbf{h} \right) + \frac{1}{d} r(\mathbf{w}') - \frac{1}{d} \boldsymbol{\eta}^T \mathbf{w}' - \frac{\tau_2}{2\eta} \frac{m^2}{\rho}$$

$$- \kappa \mathbf{g}^\top \left( \Sigma^{1/2} - \Omega^{1/2} \right) \frac{m\sqrt{\gamma}}{\|\tilde{\mathbf{v}}\|_2^2} \tilde{\mathbf{v}} - \frac{\eta}{2\tau_2 d} \left( \kappa \Omega^{1/2} \mathbf{g} - \boldsymbol{\eta} + \nu \tilde{\mathbf{v}} \right)^\top \Omega^{-1} \left( \kappa \Omega^{1/2} \mathbf{g} - \boldsymbol{\eta} + \nu \tilde{\mathbf{v}} \right) + m\nu \sqrt{\gamma}$$

$$\tag{183}$$

Another strictly convex least-square problem appears on $\boldsymbol{\eta}$, the solution and optimal value of which read

$$\boldsymbol{\eta}^* = -\frac{\tau_2}{\eta} \Omega \mathbf{w}' + (\kappa \Omega^{1/2} \mathbf{g} + \nu \tilde{\mathbf{v}}) \tag{184}$$

with optimal cost $\dfrac{\tau_2}{2\eta d} \mathbf{w}'^\top \Omega \mathbf{w}' - \mathbf{w}'^\top (\kappa \Omega^{1/2} \mathbf{g} + \nu \tilde{\mathbf{v}})$ $\tag{185}$

At this point we have expressed feasible solutions of $\boldsymbol{\eta}, \mathbf{w}_\perp$ as functions of the remaining variables. For any feasible solution in those variables, $\mathbf{w}$ and $\mathbf{w}'$ are the same. Replacing in the (AO) and a completion of squares leads to

$$\max_{\kappa, \nu, \tau_2} \min_{m, \eta, \tau_1} \frac{\kappa \tau_1}{2} - \frac{\eta \tau_2}{2} + m\nu \sqrt{\gamma} - \frac{\tau_2}{2\eta} \frac{m^2}{\rho} - \frac{\eta}{2\tau_2 d} (\nu \tilde{\mathbf{v}} + \kappa \Omega^{1/2} \mathbf{g})^\top \Omega^{-1} (\nu \tilde{\mathbf{v}} + \kappa \Omega^{1/2} \mathbf{g})$$

$$- \kappa \mathbf{g}^\top \left( \Sigma^{1/2} - \Omega^{1/2} \right) \frac{m\sqrt{\gamma}}{\|\tilde{\mathbf{v}}\|_2^2} \tilde{\mathbf{v}} + \min_{\mathbf{w}'} \left\{ r(\mathbf{w}') + \frac{\tau_2}{2\eta} \left\| \Omega^{1/2} \mathbf{w}' - \frac{\eta}{\tau_2} (\nu \Omega^{-1/2} \tilde{\mathbf{v}} + \kappa \mathbf{g}) \right\|_2^2 \right\}$$

$$\tag{186}$$

Recognizing the Moreau envelope of $f$ and introducing the variable $\tilde{\mathbf{w}} = \Omega^{1/2} \mathbf{w}' = \Omega^{1/2} \mathbf{w}$, it follows:

$$\max_{\kappa, \nu, \tau_2} \min_{m, \eta, \tau_1} \frac{\kappa \tau_1}{2} - \frac{\eta \tau_2}{2} + m\nu \sqrt{\gamma} - \frac{\tau_2}{2\eta} \frac{m^2}{\rho} - \frac{\eta}{2\tau_2 d} (\nu \tilde{\mathbf{v}} + \kappa \Omega^{1/2} \mathbf{g})^\top \Omega^{-1} (\nu \tilde{\mathbf{v}} + \kappa \Omega^{1/2} \mathbf{g})$$

$$- \kappa \mathbf{g}^\top \left( \Sigma^{1/2} - \Omega^{1/2} \right) \frac{m\sqrt{\gamma}}{\|\tilde{\mathbf{v}}\|_2^2} \tilde{\mathbf{v}} + \frac{1}{d} \mathcal{M}_{\frac{\tau_1}{\kappa} g(\cdot, \mathbf{y})} \left( \frac{m}{\sqrt{\rho}} \mathbf{s} + \eta \mathbf{h} \right) + \frac{1}{d} \mathcal{M}_{\frac{\eta}{\tau_2} r(\Omega^{-1/2} \cdot)} \left( \frac{\eta}{\tau_2} \left( \nu \Omega^{-1/2} \tilde{\mathbf{v}} + \kappa \mathbf{g} \right) \right)$$

$$\tag{187}$$

where the Moreau envelopes of $f$ and $g$ are respectively defined w.r.t. the variables $\mathbf{w}''$ and $\mathbf{z}$. At this point we have reduced the initial high-dimensional minimisation problem (168) to a scalar problem over six parameters. Another follow-up of the feasibility set shows that there exist positive constants $C_m, C_\kappa, C_\eta$ independent of $n, p, d$ such that $0 \leqslant \kappa \leqslant C_\kappa, 0 \leqslant \eta \leqslant C_\eta$ and $0 \leqslant m \leqslant C_m$.

**Case 2: $\boldsymbol{\theta}_0 \in \mathbf{Ker}(\Phi^\top)$** In this case, the min-max problem (163) becomes:

$$\min_{\mathbf{w}, \mathbf{z}} \max_{\boldsymbol{\lambda}} \boldsymbol{\lambda}^\top \frac{1}{\sqrt{d}} Z \Omega^{1/2} \mathbf{w} - \boldsymbol{\lambda}^\top \mathbf{z} + g(\mathbf{z}, \mathbf{y}) + f(\mathbf{w}) \tag{188}$$

Since $\Omega$ is positive definite, we can define $\tilde{\mathbf{w}} = \Omega^{1/2}\mathbf{w}$ and write the equivalent problem:

$$\min_{\tilde{\mathbf{w}},\mathbf{z}} \max_{\boldsymbol{\lambda}} \boldsymbol{\lambda}^\top \frac{1}{\sqrt{d}} Z\tilde{\mathbf{w}} - \boldsymbol{\lambda}^\top \mathbf{z} + g(\mathbf{z},\mathbf{y}) + f(\Omega^{-1/2}\tilde{\mathbf{w}}) \tag{189}$$

where the compactness of the feasibility set is preserved almost surely from the almost sure boundedness of the eigenvalues of $\Omega$. We can thus write the corresponding auxiliary optimization problem, reintroducing the normalization by d:

$$\min_{\tilde{\mathbf{w}},\mathbf{z}} \max_{\boldsymbol{\lambda}} \frac{1}{d}\left[\|\boldsymbol{\lambda}\|_2 \frac{1}{\sqrt{d}}\mathbf{g}^\top\tilde{\mathbf{w}} + \|\tilde{\mathbf{w}}\|_2 \frac{1}{\sqrt{d}}\mathbf{h}^\top\boldsymbol{\lambda} - \boldsymbol{\lambda}^\top\mathbf{z} + g(\mathbf{z},\mathbf{y}) + f(\Omega^{-1/2}\tilde{\mathbf{w}})\right] \tag{190}$$

introducing the convex conjugate of $f$ with dual parameter $\boldsymbol{\eta}$:

$$\min_{\tilde{\mathbf{w}},\mathbf{z}} \max_{\boldsymbol{\lambda},\boldsymbol{\eta}} \frac{1}{d}\left[\|\boldsymbol{\lambda}\|_2 \frac{1}{\sqrt{d}}\mathbf{g}^\top\tilde{\mathbf{w}} + \|\mathbf{w}_\perp\|_2 \frac{1}{\sqrt{d}}\mathbf{h}^\top\boldsymbol{\lambda} - \boldsymbol{\lambda}^\top\mathbf{z} + g(\mathbf{z},\mathbf{y}) + \boldsymbol{\eta}^\top\Omega^{-1/2}\tilde{\mathbf{w}} - f^*(\boldsymbol{\eta})\right] \tag{191}$$

We then define the scalar quantities $\kappa = \frac{\|\boldsymbol{\lambda}\|_2}{\sqrt{d}}$ and $\eta = \frac{\|\tilde{\mathbf{w}}\|_2}{\sqrt{d}}$ and perform the linear optimization on $\boldsymbol{\lambda}, \tilde{\mathbf{w}}$, giving the equivalent:

$$\min_{\mathbf{z},\eta\geqslant 0} \max_{\boldsymbol{\eta},\kappa\geqslant 0} -\frac{\eta}{\sqrt{d}}\left\|\kappa\mathbf{g} - \Omega^{-1/2}\boldsymbol{\eta}\right\|_2 + \frac{\kappa}{\sqrt{d}}\|\eta\mathbf{h} - \mathbf{z}\|_2 + \frac{1}{d}g(\mathbf{z},\mathbf{y}) - \frac{1}{d}f^*(\boldsymbol{\eta}) \tag{192}$$

Using the square root trick with parameters $\tau_1, \tau_2$:

$$\min_{\tau_1>0,\mathbf{z},\eta\geqslant 0} \max_{\tau_2>0,\boldsymbol{\eta},\kappa\geqslant 0} -\frac{\eta\tau_2}{2} - \frac{\eta}{2\tau_2 d}\left\|\kappa\mathbf{g} - \Omega^{-1/2}\boldsymbol{\eta}\right\|_2^2 + \frac{\kappa\tau_1}{2} + \frac{\kappa}{2\tau_1 d}\|\eta\mathbf{h} - \mathbf{z}\|_2^2 + \frac{1}{d}g(\mathbf{z},\mathbf{y}) - \frac{1}{d}f^*(\boldsymbol{\eta}) \tag{193}$$

performing the optimizations on $\mathbf{z}, \boldsymbol{\eta}$ and recognizing the Moreau envelopes, the problem becomes:

$$\min_{\tau_1>0,\eta\geqslant 0} \max_{\tau_2>0,\kappa\geqslant 0} -\frac{\eta\tau_2}{2} + \frac{\kappa\tau_1}{2} + \frac{1}{d}\mathcal{M}_{\frac{\tau_1}{\kappa}g(.,\mathbf{y})}(\eta\mathbf{h}) - \frac{1}{d}\mathcal{M}_{\frac{\tau_2}{\eta}f^*(\Omega^{1/2}.)}(\kappa\mathbf{g}) \tag{194}$$

$$\iff \min_{\tau_1>0,\eta\geqslant 0} \max_{\tau_2>0,\kappa\geqslant 0} -\frac{\eta\tau_2}{2} + \frac{\kappa\tau_1}{2} + \frac{1}{d}\mathcal{M}_{\frac{\tau_1}{\kappa}g(.,\mathbf{y})}(\eta\mathbf{h}) + \frac{1}{d}\mathcal{M}_{\frac{\eta}{\tau_2}f(\Omega^{-1/2}.)}\left(\frac{\eta}{\tau_2}\kappa\mathbf{g}\right) - \frac{\eta}{2\tau_2 d}\kappa^2\mathbf{g}^\top\mathbf{g} \tag{195}$$

This concludes the proof of Lemma 9. $\qquad\qquad\square$

## B.5   Study of the scalar equivalent problem : geometry and asymptotics.

Here we study the geometry, solutions and asymptotics of the scalar optimization problem (187). We will focus on the case $\boldsymbol{\theta}_0 \notin \text{Ker}(\Phi^\top)$ as the other case simply shows that no learning is performed (see the remark at the end of this section). The following lemma characterizes the continuity and geometry of the cost function $\mathcal{E}_n$.

**Lemma 10.**   *(Geometry of $\mathcal{E}_n$) Recall the function:*

$$\mathcal{E}_n(\tau_1,\tau_2,\kappa,\eta,\nu,m) = \frac{\kappa\tau_1}{2} - \frac{\eta\tau_2}{2} + m\nu\sqrt{\gamma} - \frac{\tau_2}{2\eta}\frac{m^2}{\rho} - \frac{\eta}{2\tau_2 d}(\nu\tilde{\mathbf{v}} + \kappa\Omega^{1/2}\mathbf{g})^\top\Omega^{-1}(\nu\tilde{\mathbf{v}} + \kappa\Omega^{1/2}\mathbf{g})$$

$$- \kappa\mathbf{g}^\top\left(\Sigma^{1/2} - \Omega^{1/2}\right)\frac{m\sqrt{\gamma}}{\|\tilde{\mathbf{v}}\|_2^2}\tilde{\mathbf{v}} + \frac{1}{d}\mathcal{M}_{\frac{\tau_1}{\kappa}g(.,\mathbf{y})}\left(\frac{m}{\sqrt{\rho}}\mathbf{s} + \eta\mathbf{h}\right) + \frac{1}{d}\mathcal{M}_{\frac{\eta}{\tau_2}r(\Omega^{-1/2}.)}\left(\frac{\eta}{\tau_2}\left(\nu\Omega^{-1/2}\tilde{\mathbf{v}} + \kappa\mathbf{g}\right)\right) \tag{196}$$

*Then $\mathcal{E}_n(\tau_1,\tau_2,\kappa,\eta,\nu,m)$ is continuous on its domain, jointly convex in $(m,\eta,\tau_1)$ and jointly concave in $(\kappa,\nu,\tau_2)$.*

*Proof of Lemma 10 :* $\mathcal{E}_n(\tau_1,\tau_2,\kappa,\eta,\nu,m)$ is a linear combination of linear and quadratic terms with Moreau envelopes, which are all continuous on their domain. Remembering the formulation

$$\mathcal{E}_n(\tau_1,\tau_2,\kappa,\eta,\nu,m) = \frac{\kappa\tau_1}{2} - \frac{\eta\tau_2}{2} + m\nu\sqrt{\gamma} - \frac{\tau_2}{2\eta}\frac{m^2}{\rho} - \kappa\mathbf{g}^\top\left(\Sigma^{1/2} - \Omega^{1/2}\right)\frac{m\sqrt{\gamma}}{\|\tilde{\mathbf{v}}\|_2^2}\tilde{\mathbf{v}}$$

$$+ \frac{1}{d}\mathcal{M}_{\frac{\tau_1}{\kappa}g(.,\mathbf{y})}\left(\frac{m}{\sqrt{\rho}}\mathbf{s} + \eta\mathbf{h}\right) - \frac{1}{d}\mathcal{M}_{\frac{\tau_2}{\eta}f^*(\Omega^{1/2}.)}\left(\Omega^{-1/2}\left(\nu\tilde{\mathbf{v}} + \kappa\Omega^{1/2}\mathbf{g}\right)\right) \tag{197}$$

and using the properties of Moreau envelopes, $\mathcal{M}_{\frac{\tau_1}{\kappa} g(.,\mathbf{y})}\left(\frac{m}{\sqrt{\rho}}\mathbf{s} + \eta\mathbf{h}\right)$ is jointly convex in $(\kappa, \tau_1, m, \eta)$ as a composition of convex functions of those arguments. The same applies for $\mathcal{M}_{\frac{\tau_2}{\eta} f^*(\Omega^{1/2}.)}\left(\Omega^{-1/2}\left(\nu\tilde{\mathbf{v}} + \kappa\Omega^{1/2}\mathbf{g}\right)\right)$, jointly convex in $(\tau_2, \eta, \nu, \kappa)$, and its opposite is jointly concave in those parameters. The remaining terms being linear in $\tau_1, \tau_2, \nu$, we conclude that $\mathcal{E}_n(\tau_1, \tau_2, \kappa, \eta, \nu, m)$ is jointly concave in $(\nu, \tau_2)$ and convex in $\tau_1$ whatever the values of $(\kappa, \eta, m)$. Going back to equation (171), we can write

$$\mathcal{E}_n(\tau_1, \tau_2, \kappa, \eta, \nu, m) = \max_{\boldsymbol{\mu}} \min_{\mathbf{z},\mathbf{w}_\perp} \frac{\kappa\tau_1}{2} - \frac{\eta\tau_2}{2} - \frac{\eta}{2\tau_2 d}\|\boldsymbol{\mu} + \kappa\mathbf{g}\|_2^2 + \frac{\kappa}{2\tau_1 d}\left\|\frac{m}{\sqrt{\rho}}\mathbf{s} + \eta\mathbf{h} - \mathbf{z}\right\|_2^2$$
$$+ \frac{1}{d}\left[g(\mathbf{z},\mathbf{y}) + f\left(\frac{m\sqrt{dp}}{\|\tilde{\mathbf{v}}\|_2^2}\tilde{\mathbf{v}} + \mathbf{w}_\perp\right) + \boldsymbol{\mu}^\top \Sigma^{1/2}\left(\frac{m\sqrt{dp}}{\|\tilde{\mathbf{v}}\|_2^2}\tilde{\mathbf{v}} + \mathbf{w}_\perp\right) - \nu\tilde{\mathbf{v}}^\top\mathbf{w}_\perp\right]$$
(198)

The squared term in $m, \eta, \mathbf{z}$ can be written as

$$\frac{\kappa}{2\tau_1 d}\left\|\frac{m}{\sqrt{\rho}}\mathbf{s} + \eta\mathbf{h} - \mathbf{z}\right\|_2^2 = \tau_1 \frac{\kappa}{2d}\left\|\frac{m}{\tau_1\sqrt{\rho}}\mathbf{s} + \frac{\eta}{\tau_1}\mathbf{h} - \frac{\mathbf{z}}{\tau_1}\right\|_2^2 \tag{199}$$

which is the perspective function with parameter $\tau_1$ of a function jointly convex in $(\mathbf{z}, m, \eta)$. Thus it is jointly convex in $(\tau_1, \mathbf{z}, m, \eta)$. Furthermore, the term $f\left(\frac{m\sqrt{dp}}{\|\tilde{\mathbf{v}}\|_2^2}\tilde{\mathbf{v}} + \mathbf{w}_\perp\right)$ is a composition of a convex function with a linear one, thus it is jointly convex in $(m, \mathbf{w}_\perp)$. The remaining terms in $\tau_1, \eta, m$ are linear. Since minimisation on convex sets preserves convexity, minimizing with respect to $\mathbf{z}, \mathbf{w}_\perp$ will lead to a jointly convex function in $(\tau_1, \eta, m)$. Similarly, the term $-\frac{\eta}{2\tau_2 d}\|\boldsymbol{\mu} + \kappa\mathbf{g}\|_2^2$ is jointly concave in $\tau_2, \kappa, \boldsymbol{\mu}$, and maximizing over $\boldsymbol{\mu}$ will result in a jointly concave function in $(\tau_2, \nu, \kappa)$. We conclude that $\mathcal{E}_n(\tau_1, \tau_2, \kappa, \eta, \nu, m)$ is jointly convex in $(\tau_1, m, \eta)$ and jointly concave in $(\kappa, \nu, \tau_2)$. $\qquad\square$

The next lemma then characterizes the infinite dimensional limit of the scalar optimization problem (187), along with the consistency of its optimal value.

**Lemma 11.** *(Asymptotics of $\mathcal{E}_n$) Recall the following quantities:*

$$\mathcal{L}_g(\tau_1, \kappa, m, \eta) = \frac{1}{n}\mathbb{E}\left[\mathcal{M}_{\frac{\tau_1}{\kappa} g(.,\mathbf{y})}\left(\frac{m}{\sqrt{\rho}}\mathbf{s} + \eta\mathbf{h}\right)\right] \text{ where } \mathbf{y} = f_0(\sqrt{\rho_p}\mathbf{s}), \ \mathbf{s} \sim \mathcal{N}(0, \mathbb{I}_n) \quad (200)$$

$$\mathcal{L}_r(\tau_2, \eta, \nu, \kappa) = \frac{1}{d}\mathbb{E}\left[\mathcal{M}_{\frac{\eta}{\tau_2} r(\Omega^{-1/2}.)}\left(\frac{\eta}{\tau_2}\left(\nu\Omega^{-1/2}\tilde{\mathbf{v}} + \kappa\mathbf{g}\right)\right)\right] \text{ where } \tilde{\mathbf{v}} = \Phi^\top\boldsymbol{\theta}_0 \quad (201)$$

$$\chi = \frac{1}{d}\boldsymbol{\theta}_0^\top \Phi\Omega^{-1}\Phi^\top\boldsymbol{\theta}_0 \tag{202}$$

$$\rho = \frac{1}{p}\boldsymbol{\theta}_0^\top\Psi\boldsymbol{\theta}_0 \tag{203}$$

*and the potential:*

$$\mathcal{E}(\tau_1, \tau_2, \kappa, \eta, \nu, m) = \frac{\kappa\tau_1}{2} - \frac{\eta\tau_2}{2} + m\nu\sqrt{\gamma} - \frac{\tau_2}{2\eta}\frac{m^2}{\rho} - \frac{\eta}{2\tau_2}(\nu^2\chi + \kappa^2) + \alpha\mathcal{L}_g(\tau_1, \kappa, m, \eta) + \mathcal{L}_r(\tau_2, \eta, \nu, \kappa)$$
(204)

*Then:*
$$\max_{\kappa,\nu,\tau_2} \min_{m,\eta,\tau_1} \mathcal{E}_n(\tau_1, \tau_2, \kappa, \eta, \nu, m) \xrightarrow[n,p,d\to\infty]{P} \max_{\kappa,\nu,\tau_2} \min_{m,\eta,\tau_1} \mathcal{E}(\tau_1, \tau_2, \kappa, \eta, \nu, m) \tag{205}$$

*and $\mathcal{E}(\tau_1, \tau_2, \kappa, \eta, \nu, m)$ is continuously differentiable on its domain, jointly convex in $(m, \eta, \tau_1)$ and jointly concave in $(\kappa, \nu, \tau_2)$.*

*Proof of Lemma 11*: The strong law of large numbers, see e.g. [72] gives $\frac{1}{d}\mathbf{g}^\top\mathbf{g} \xrightarrow[d\to\infty]{a.s.} 1$. Additionally, using assumption (A2) on the summability of $\boldsymbol{\theta}_0$ and (A3) on the boundedness of the spectrum of the covariance matrices, the quantity $\chi = \lim_{d\to\infty} \frac{1}{d}\boldsymbol{\theta}_0^\top \Phi\Omega^{-1}\Phi^\top\boldsymbol{\theta}_0$ exists and is finite. Since $\boldsymbol{\theta}_0 \notin \mathrm{Ker}(\Phi^\top)$ and using the non-vanishing signal hypothesis, the quantity $\rho_{\tilde{\mathbf{v}}} = \lim_{d\to\infty} \frac{1}{d}\tilde{\mathbf{v}}^\top\tilde{\mathbf{v}}$

exists, is finite and strictly positive. Then $\kappa \mathbf{g}^\top \left( \Sigma^{1/2} - \Omega^{1/2} \right) \frac{m\sqrt{\gamma}}{\|\tilde{\mathbf{v}}\|_2^2} \mathbf{v}$ is a centered Gaussian random variable with variance verifying:

$$\mathrm{Var}\left[ \kappa \mathbf{g}^\top \left( \Sigma^{1/2} - \Omega^{1/2} \right) \frac{m\sqrt{\gamma}}{\|\tilde{\mathbf{v}}\|_2^2} \tilde{\mathbf{v}} \right] \leqslant \kappa^2 \sigma_{max}^2 \left( \Sigma^{1/2} - \Omega^{1/2} \right) \frac{m^2 \gamma}{\|\tilde{\mathbf{v}}\|_2^2}$$

$$= \kappa^2 \sigma_{max}^2 \left( \Sigma^{1/2} - \Omega^{1/2} \right) \frac{m^2 \gamma}{d \rho_{\tilde{\mathbf{v}}}} \tag{206}$$

Using lemma 8, $\kappa$ and $m$ are finitely bounded independently of the dimension $d$. $\gamma, \sigma_{max} \left( \Sigma^{1/2} - \Omega^{1/2} \right)$ are finite. Thus there exists a finite constant $C$ such that the standard deviation of $\kappa \mathbf{g}^\top \left( \Sigma^{1/2} - \Omega^{1/2} \right) \frac{m\sqrt{\gamma}}{\|\tilde{\mathbf{v}}\|_2^2} \tilde{\mathbf{v}}$ is smaller than $\sqrt{C}/\sqrt{d}$. Then, for any $\epsilon > 0$:

$$\mathbb{P}\left( \left| \kappa \mathbf{g}^\top \left( \Sigma^{1/2} - \Omega^{1/2} \right) \frac{m\sqrt{\gamma}}{\|\tilde{\mathbf{v}}\|_2^2} \tilde{\mathbf{v}} \right| \geqslant \epsilon \right) \leqslant \mathbb{P}\left( |\mathcal{N}(0,1)| \geqslant \epsilon \sqrt{d}/\sqrt{C} \right)$$

$$\leqslant \frac{\sqrt{C}}{\epsilon \sqrt{d}} \frac{1}{\sqrt{2\pi}} \exp\left( -\frac{1}{2} \frac{\epsilon^2 d}{C} \right) \tag{207}$$

using the Gaussian tail. The Borel-Cantelli lemma and summability of this tail gives

$$\kappa \mathbf{g}^\top \left( \Sigma^{1/2} - \Omega^{1/2} \right) \frac{m\sqrt{\gamma}}{\|\tilde{\mathbf{v}}\|_2^2} \tilde{\mathbf{v}} \xrightarrow[d\to\infty]{a.s.} 0 \tag{208}$$

Concentration of the Moreau envelopes of both $f$ and $g$ follows directly from lemma 5.
We thus have the pointwise convergence:

$$\mathcal{E}_n(\tau_1, \tau_2, \kappa, \eta, \nu, m) \xrightarrow[n,p,d\to\infty]{P} \mathcal{E}(\tau_1, \tau_2, \kappa, \eta, \nu, m) \tag{209}$$

Since pointwise convergence preserves convexity, $\mathcal{E}(\tau_1, \tau_2, \kappa, \eta, \nu, m)$ is jointly convex in $(m, \eta, \tau_1)$ and jointly concave in $(\kappa, \nu, \tau_2)$.
Now recall the expression of $\mathcal{E}$

$$\mathcal{E}(\tau_1, \tau_2, \kappa, \eta, \nu, m) = \frac{\kappa \tau_1}{2} - \frac{\eta \tau_2}{2} + m\nu\sqrt{\gamma} - \frac{\tau_2}{2\eta} \frac{m^2}{\rho} - \frac{\eta}{2\tau_2}(\nu^2 \chi + \kappa^2) + \alpha \mathcal{L}_g(\tau_1, \kappa, m, \eta) + \mathcal{L}_f(\tau_2, \eta, \nu, \kappa) \tag{210}$$

The feasibility sets of $\kappa, \eta, m$ are compact from Lemma 8 and the subsequent follow-up of the feasibility sets. Then, using Proposition 12.32 from [68], for fixed $(\tau_2, \kappa, \eta, \nu, m)$, we have:

$$\lim_{\tau_1 \to +\infty} \frac{1}{d} \mathcal{M}_{\frac{\tau_1}{\kappa} g(.,\mathbf{y})} \left( \frac{m}{\sqrt{\rho}} \mathbf{s} + \eta \mathbf{h} \right) = \frac{1}{d} \inf_{\mathbf{z} \in \mathbb{R}^n} g(\mathbf{z}, \mathbf{y}) \tag{211}$$

which is a finite quantity since $g(., \mathbf{y})$ is a proper, convex function verifying the scaling assumptions B.1. Then, since $\kappa > 0$, we have:

$$\lim_{\tau_1 \to +\infty} \mathcal{E}_n(\tau_1, \tau_2, \kappa, \eta, \nu, m) = +\infty \tag{212}$$

Similarly, for fixed $(\tau_1, \kappa, \eta, \nu, m)$ and noting that composing $f$ with the positive definite matrix $\Omega^{-1/2}$ does not change its convexity, or it being proper and lower semi-continuous, we get:

$$\lim_{\tau_2 \to +\infty} \frac{1}{d} \mathcal{M}_{\frac{\eta}{\tau_2} f(\Omega^{-1/2}.)} \left( \frac{\eta}{\tau_2} \left( \nu \Omega^{-1/2} \tilde{\mathbf{v}} + \kappa \mathbf{g} \right) \right) = \frac{1}{d} f(0_d) \tag{213}$$

which is also a bounded quantity from the scaling assumptions made on $f$. Since $\beta > 0$, we then have:

$$\lim_{\tau_2 \to +\infty} \mathcal{E}_n(\tau_1, \tau_2, \kappa, \eta, \nu, m) = -\infty \tag{214}$$

Finally, the limit $\lim_{\nu \to +\infty} \mathcal{E}_n(\tau_1, \tau_2, \kappa, \eta, \nu, m)$ needs to be checked for both $+\infty$ and $-\infty$ since there is no restriction on the sign of $\nu$. From the definition of the Moreau envelope, we can write:

$$\frac{1}{d} \mathcal{M}_{\frac{\eta}{\tau_2} f(\Omega^{-1/2}.)} \left( \frac{\eta}{\tau_2} \left( \nu \Omega^{-1/2} \tilde{\mathbf{v}} + \kappa \mathbf{g} \right) \right) \leqslant \frac{1}{d} f(0_d) + \frac{\tau_2}{2\eta} \left\| \frac{\eta}{d\tau_2} \left( \nu \Omega^{-1/2} \tilde{\mathbf{v}} + \kappa \mathbf{g} \right) \right\|_2^2 \tag{215}$$

Thus, for any fixed $(\tau_1, \tau_2, m, \kappa, \eta)$:

$$\mathcal{E}_n(\tau_1, \tau_2, \kappa, \eta, \nu, m) \leqslant \frac{\kappa\tau_1}{2} - \frac{\eta\tau_2}{2} + m\nu\sqrt{\gamma} - \kappa\mathbf{g}^\top \left(\Sigma^{1/2} - \Omega^{1/2}\right) \frac{m\sqrt{\gamma}}{\|\tilde{\mathbf{v}}\|_2^2}\mathbf{v} + \frac{1}{d}\mathcal{M}_{\frac{\tau_1}{\kappa}g(.,\mathbf{y})}\left(\frac{m}{\sqrt{\rho}}\mathbf{s} + \eta\mathbf{h}\right)$$
$$+ \frac{1}{d}f(0_d)$$
(216)

which immediately gives $\lim_{\nu \to -\infty} \mathcal{E}_n = -\infty$. Turning to the other limit, remembering that $\mathcal{E}_n$ is continuously differentiable on its domain, we have:

$$\frac{\partial \mathcal{E}_n}{\partial \nu}(\tau_1, \tau_2, \kappa, \eta, \nu, m) = m\sqrt{\gamma} - \frac{1}{d}\tilde{\mathbf{v}}^\top \Omega^{-1/2}\text{prox}_{\frac{\eta}{\tau_2}f(\Omega^{-1/2}.)}\left(\frac{\eta}{\tau_2}\left(\nu\Omega^{-1/2}\tilde{\mathbf{v}} + \kappa\mathbf{g}\right)\right) \quad (217)$$

Thus $\lim_{\nu \to +\infty} \frac{\partial \mathcal{E}_n(\tau_1, \tau_2, \kappa, \eta, \nu, m)}{\partial \nu} \to -\infty$. Since $\mathcal{E}_n$ is continuously differentiable in $\nu$ on $[0, +\infty[$, and from the short argument led above, we have shown

$$\lim_{|\nu| \to +\infty} \mathcal{E}_n(\tau_1, \tau_2, \kappa, \eta, \nu, m) = -\infty \quad (218)$$

Using similar arguments as in the proof of Lemma 8, we can now reduce the feasibility set of $\tau_1, \tau_2, \nu$ to a compact one. Then, using the fact that convergence of convex functions on compact sets implies uniform convergence [73], we obtain

$$\max_{\kappa,\nu,\tau_2} \min_{m,\eta,\tau_1} \mathcal{E}_n(\tau_1, \tau_2, \kappa, \eta, \nu, m) \xrightarrow[n,p,d \to +\infty]{P} \max_{\kappa,\nu,\tau_2} \min_{m,\eta,\tau_1} \mathcal{E}(\tau_1, \tau_2, \kappa, \eta, \nu, m) \quad (219)$$

which is the desired result. □

At this point, it is necessary to characterize the set of solutions of the asymptotic minimisation problem (90). We start with the explicit form of the optimality condition associated to any solution.

**Lemma 12.** *(Fixed point equations) The zero-gradient condition of the optimization problem (90) prescribes the following set of fixed point equations for any feasible solution:*

$$\partial_\kappa : \tau_1 = \frac{1}{d}\mathbb{E}\left[\mathbf{g}^\top \text{prox}_{\frac{\eta}{\tau_2}f(\Omega^{-1/2}.)}\left(\frac{\eta}{\tau_2}\left(\nu\Omega^{-1/2}\tilde{\mathbf{v}} + \kappa\mathbf{g}\right)\right)\right] \quad (220)$$

$$\partial_\nu : m\sqrt{\gamma} = \frac{1}{d}\mathbb{E}\left[\tilde{\mathbf{v}}^\top \Omega^{-1/2}\text{prox}_{\frac{\eta}{\tau_2}f(\Omega^{-1/2}.)}\left(\frac{\eta}{\tau_2}\left(\nu\Omega^{-1/2}\tilde{\mathbf{v}} + \kappa\mathbf{g}\right)\right)\right] \quad (221)$$

$$\partial_\eta : \tau_2 = \alpha\frac{\kappa}{\tau_1}\eta - \frac{\kappa\alpha}{\tau_1 n}\mathbb{E}\left[\mathbf{h}^\top \text{prox}_{\frac{\tau_1}{\kappa}g(.,\mathbf{y})}\left(\frac{m}{\sqrt{\rho}}\mathbf{s} + \eta\mathbf{h}\right)\right] \quad (222)$$

$$\partial_{\tau_2} : \frac{1}{2d}\frac{\tau_2}{\eta}\mathbb{E}\left[\left\|\frac{\eta}{\tau_2}(\nu\Omega^{-1/2}\tilde{\mathbf{v}} + \kappa\mathbf{g}) - \text{prox}_{\frac{\eta}{\tau_2}f(\Omega^{-1/2}.)}\left(\frac{\eta}{\tau_2}\left(\nu\Omega^{-1/2}\tilde{\mathbf{v}} + \kappa\mathbf{g}\right)\right)\right\|_2^2\right] =$$

$$\frac{\eta}{2\tau_2}(\nu^2\chi + \kappa^2) - m\nu\sqrt{\gamma} - \kappa\tau_1 + \frac{\eta\tau_2}{2} + \frac{\tau_2}{2\eta}\frac{m^2}{\rho} \quad (223)$$

$$\partial_m : \nu\sqrt{\gamma} = \alpha\frac{\kappa}{n\tau_1}\mathbb{E}\left[(\frac{m}{\eta\rho}\mathbf{h} - \frac{\mathbf{s}}{\sqrt{\rho}})^\top \text{prox}_{\frac{\tau_1}{\kappa}g(.,\mathbf{y})}\left(\frac{m}{\sqrt{\rho}}\mathbf{s} + \eta\mathbf{h}\right)\right] \quad (224)$$

$$\partial_{\tau_1} : \frac{\tau_1^2}{2} = \frac{1}{2}\alpha\frac{1}{n}\mathbb{E}\left[\left\|\frac{m}{\sqrt{\rho}}\mathbf{s} + \eta\mathbf{h} - \text{prox}_{\frac{\tau_1}{\kappa}g(.,y)}\left(\frac{m}{\sqrt{\rho}}\mathbf{s} + \eta\mathbf{h}\right)\right\|_2^2\right] \quad (225)$$

*This set of equations can be converted to the replica notations using the table (280).*

*Proof of Lemma 12*: Using arguments similar to the ones in the proof of Lemma 5, Moreau envelopes and their derivatives verify the necessary conditions of the dominated convergence theorem. Additionally, uniform convergence of the sequence of derivatives can be verified in a straightforward manner as all involved functions are firmly non-expansive and integrated w.r.t. Gaussian measures. We can therefore invert the limits and derivatives, and invert expectations and derivatives. We can now write explicitly the optimality condition for the scalar problem (204), using the expressions for

derivatives of Moreau envelopes from Appendix B.3. Some algebra and replacing with prescriptions obtained from each partial derivative leads to the set of equations above. $\qquad\square$

**Remark** : Here we see that the potential function (204) can be further studied using the fixed point equations (12) and the relation (102). For any optimal $(\tau_1, \tau_2, \kappa, \eta, \nu, m)$, it holds that

$$\mathcal{E}(\tau_1, \tau_2, \kappa, \eta, \nu, m)$$
$$= \alpha \frac{1}{n} \mathbb{E}\left[g\left(\text{prox}_{\frac{\tau_1}{\kappa} g(.,y)}\left(\frac{m}{\sqrt{\rho}}\mathbf{s} + \eta \mathbf{h}\right), \mathbf{y}\right)\right] + \frac{1}{d}\mathbb{E}\left[f\left(\Omega^{-1/2}\text{prox}_{\frac{\eta}{\tau_2} f(\Omega^{-1/2}.)}\left(\frac{\eta}{\tau_2}\left(\nu\Omega^{-1/2}\tilde{\mathbf{v}} + \kappa\mathbf{g}\right)\right)\right)\right]$$
(226)

Finally, we give a strict-convexity and strict-concavity property of the asymptotic potential $\mathcal{E}$ which will be helpful to prove Lemma 1.

**Lemma 13.** *(Strict convexity and strict concavity near minimisers) Consider the asymptotic potential function $\mathcal{E}(\tau_1, \tau_2, \kappa, \eta, \nu, m)$. Then for any fixed $(\eta, m, \tau_1)$ in their feasibility sets, the function*

$$\tau_2, \kappa, \nu \to \mathcal{E}(\tau_1, \tau_2, \kappa, \eta, \nu, m) \tag{227}$$

*is jointly strictly concave in $(\tau_2, \kappa, \nu)$.*
*Additionally, consider the set $\mathcal{S}_{\partial_{\nu, \tau_2}}$ defined by:*

$$\mathcal{S}_{\partial_{\nu, \tau_2}} = \left\{ \tau_1, \tau_2, \kappa, \eta, \nu, m \mid m\sqrt{\gamma} = \frac{1}{d}\mathbb{E}\left[\tilde{\mathbf{v}}^T \Omega^{-1/2} prox_{\frac{\eta}{\tau_2} f(\Omega^{-1/2}.)}\left(\frac{\eta}{\tau_2}\left(\nu\Omega^{-1/2}\tilde{\mathbf{v}} + \kappa\mathbf{g}\right)\right)\right], \right.$$
$$\left. \frac{1}{2d}\frac{1}{\eta}\mathbb{E}\left[\left\|prox_{\frac{\eta}{\tau_2} f(\Omega^{-1/2}.)}\left(\frac{\eta}{\tau_2}\left(\nu\Omega^{-1/2}\tilde{\mathbf{v}} + \kappa\mathbf{g}\right)\right)\right\|_2^2\right] = \frac{\eta}{2} + \frac{1}{2\eta}\frac{m^2}{\rho}\right\}$$
(228)

*then for any fixed $\tau_2, \kappa, \nu$ in $\mathcal{S}_{\partial_{\nu, \tau_2}}$, the function $(\eta, m, \tau_1) \to \mathcal{E}(\tau_1, \tau_2, \kappa, \eta, \nu, m)$ is jointly strictly convex in $(\eta, m, \tau_1)$ on $\mathcal{S}_{\partial_{\nu, \tau_2}}$*

*Proof of Lemma 13*: We will use the following first order characterization of strictly convex functions: $f$ is strictly convex $\iff \langle \mathbf{x} - \mathbf{y} | \nabla f(\mathbf{x}) - \nabla f(\mathbf{y}) \rangle > 0 \; \forall \mathbf{x} \neq \mathbf{y} \in \text{dom}(f)$. To simplify notations, we will write, for any fixed $(m, \eta, \tau_1)$

$$(\nabla_{\kappa, \nu, \tau_2}\mathcal{E}) = ((\partial_\kappa \mathcal{E}, \partial_\nu \mathcal{E}, \partial_{\tau_2}\mathcal{E})(\tau_1, \tau_2, \kappa, \eta, \nu, m))_i \tag{229}$$

as the i-th component of the gradient of $\mathcal{E}(\tau_1, \tau_2, \kappa, \eta, \nu, m)$ with respect to $(\kappa, \nu, \tau_2)$ for any fixed $(m, \eta, \tau_1)$ in the feasibility set. Then for any distinct triplets $(\kappa, \nu, \tau_2), (\tilde{\kappa}, \tilde{\nu}, \tilde{\tau}_2)$ and fixed $(\eta, m, \tau_1)$ in the feasibility set, determining the partial derivatives of $\mathcal{E}$ in similar fashion as is implied in the proof of Lemma 12, we have:

$$((\kappa, \nu, \tau_2) - (\tilde{\kappa}, \tilde{\nu}, \tilde{\tau}_2))^\top (\nabla\mathcal{E}_{\kappa, \nu, \tau_2} - \nabla\mathcal{E}_{\tilde{\kappa}, \tilde{\nu}, \tilde{\tau}_2})$$
$$= (\kappa - \tilde{\kappa})\alpha\frac{1}{2\tau_1}\frac{1}{n}\left(\mathbb{E}\left[\left\|\mathbf{r}_1 - \text{prox}_{\frac{\tau_1}{\kappa} g(.,\mathbf{y})}(\mathbf{r}_1)\right\|_2^2 - \left\|\mathbf{r}_1 - \text{prox}_{\frac{\tau_1}{\tilde{\kappa}} g(.,\mathbf{y})}(\mathbf{r}_1)\right\|_2^2\right]\right)$$
$$+ \left(\text{prox}_{\frac{\eta}{\tau_2} f(\Omega^{-1/2}.)}\left(\frac{\eta}{\tau_2}\mathbf{r}_2\right) - \text{prox}_{\frac{\eta}{\tilde{\tau}_2} f(\Omega^{-1/2}.)}\left(\frac{\eta}{\tilde{\tau}_2}\tilde{\mathbf{r}}_2\right)\right)^\top \left(\tilde{\mathbf{r}}_2 - \mathbf{r}_2\right.$$
$$\left. + \frac{\tau_2 - \tilde{\tau}_2}{2\eta d}\left(\text{prox}_{\frac{\eta}{\tau_2} f(\Omega^{-1/2}.)}\left(\frac{\eta}{\tau_2}\mathbf{r}_2\right) + \text{prox}_{\frac{\eta}{\tilde{\tau}_2} f(\Omega^{-1/2}.)}\left(\frac{\eta}{\tilde{\tau}_2}\tilde{\mathbf{r}}_2\right)\right)\right)$$
$$\leqslant (\kappa - \tilde{\kappa})\alpha\frac{1}{2\tau_1}\frac{1}{n}\left(\mathbb{E}\left[\left\|\mathbf{r}_1 - \text{prox}_{\frac{\tau_1}{\kappa} g(.,y)}(\mathbf{r}_1)\right\|_2^2 - \left\|\mathbf{r}_1 - \text{prox}_{\frac{\tau_1}{\tilde{\kappa}} g(.,y)}(\mathbf{r}_1)\right\|_2^2\right]\right)$$
$$+ \left(\frac{(\tau_2 + \tilde{\tau}_2)}{2\eta d}\mathbb{E}\left[-\left\|\text{prox}_{\frac{\eta}{\tau_2} f(\Omega^{-1/2}.)}\left(\frac{\eta}{\tau_2}\mathbf{r}_2\right) - \text{prox}_{\frac{\eta}{\tilde{\tau}_2} f(\Omega^{-1/2}.)}\left(\frac{\eta}{\tilde{\tau}_2}\tilde{\mathbf{r}}_2\right)\right\|_2^2\right]\right)$$
(230)

where the last line follows from the inequality in Lemma 4, and we defined the shorthands, $\mathbf{r}_1 = \frac{m}{\sqrt{\rho}}\mathbf{s} + \eta\mathbf{h}$, $\mathbf{r}_2 = \nu\Omega^{-1/2}\tilde{\mathbf{v}} + \kappa\mathbf{g}$, $\tilde{\mathbf{r}}_2 = \tilde{\nu}\Omega^{-1/2}\mathbf{v} + \tilde{\kappa}\mathbf{g}$. Using Lemma 3, the first term of the r.h.s of the last inequality is also negative as an increment of a nonincreasing function. Thus, both

expectations are taken on negative functions. If those functions are not zero almost everywhere with respect to the Lebesgue measure, then the result will be strictly negative. Moreover, the functional taking each operator $T$ to its resolvent $(\mathrm{Id} + T)^{-1}$ is a bijection on the set of non-trivial, maximally monotone operators, see e.g. [68] Proposition 23.21 and the subsequent discussion. The subdifferential of a proper, closed, convex function being maximally monotone, for two different parameters the corresponding proximal operator cannot be equal almost everywhere. The previously studied increment $((\kappa, \nu, \tau_2) - (\tilde{\kappa}, \tilde{\nu}, \tilde{\tau}_2))^\top (\nabla \mathcal{E}_{\kappa,\nu,\tau_2} - \nabla \mathcal{E}_{\tilde{\kappa},\tilde{\nu},\tilde{\tau}_2})$ is therefore strictly negative, giving the desired strict concavity in $(\kappa, \nu, \tau_2)$. Restricting ourselves to the set $\mathcal{S}_{\partial\nu,\tau_2}$, the increment in $(m, \eta, \tau_1)$ can be written similarly. Note that $Id - \mathrm{prox}$ will appear in the expressions instead of prox. The appropriate terms can then be brought to the form of the inequality from Lemma 4 using Moreau's decomposition. Using the definitions of the set $\mathcal{S}_{\partial\nu,\tau_2}$ and the increments from Lemma 3, a similar argument as the previous one can be carried out. The lemma is proved. $\square$

What is now left to do is link the properties of the scalar optimization problem (90) to the original learning problem (3) using the tight inequalities from Theorem 6.

**Remark**: in the case $\boldsymbol{\theta}_0 \in \mathrm{Ker}(\Phi^T)$, the cost function $\mathcal{E}_n^0$ will uniformly converge to the following potential:

$$-\frac{\eta\tau_2}{2} + \frac{\kappa\tau_1}{2} - \frac{\eta}{2\tau_2}\kappa^2 + \frac{\alpha}{n}\mathbb{E}\left[\mathcal{M}_{\frac{\tau_1}{\kappa}g(.,\mathbf{y})}(\eta\mathbf{h})\right] + \frac{1}{d}\mathbb{E}\left[\mathcal{M}_{\frac{\eta}{\tau_2}f(\Omega^{-1/2}.)}(\frac{\eta}{\tau_2}\kappa\mathbf{g})\right] \tag{231}$$

As we will see in the next section, this will lead to estimators solely based on noise.

## B.6   Back to the original problem : proof of Theorem 4 and 5

We begin this part by considering that the "necessary assumptions for exponential rates" from the set of assumptions B.1 are verified. In the end we will discuss how relaxing these assumptions modifies the convergence speed. We closely follow the analysis introduced in [51] and further developed in [29]. The main difference resides in checking the concentration properties of generic Moreau envelopes depending on the regularity of the target function instead of specific instances such as the LASSO. Since the dimensions $n, p, d$ are linked by multiplicative constants, we can express the rates with any of the three. Recall the original reformulation of the problem defining the student.

$$\max_{\boldsymbol{\lambda}} \min_{\mathbf{w},\mathbf{z}} g(\mathbf{z}, \mathbf{y}) + f(\mathbf{w}) + \boldsymbol{\lambda}^\top \left(\frac{1}{\sqrt{d}}\left(A\Psi^{-1/2}\Phi + B\left(\Omega - \Phi^\top\Psi^{-1}\Phi\right)^{1/2}\right)\mathbf{w} - \mathbf{z}\right) \tag{232}$$

Introducing the variable $\tilde{\mathbf{w}} = \Omega^{1/2}\mathbf{w}$ it can be equivalently written, since $\Omega$ is almost surely invertible and the problem is convex concave with a closed convex feasibility set on $\tilde{\mathbf{w}}, \mathbf{z}$.

$$\min_{\tilde{\mathbf{w}},\mathbf{z}} \max_{\boldsymbol{\lambda}} g(\mathbf{z}, \mathbf{y}) + f(\Omega^{-1/2}\tilde{\mathbf{w}}) + \boldsymbol{\lambda}^\top \left(\frac{1}{\sqrt{d}}\left(A\Psi^{-1/2}\Phi + B\left(\Omega - \Phi^\top\Psi^{-1}\Phi\right)^{1/2}\right)\Omega^{-1/2}\tilde{\mathbf{w}} - \mathbf{z}\right)$$
$$\tag{233}$$

Recall the equivalent scalar auxiliary problem at finite dimension $\mathcal{E}_n$ and its asymptotic counterpart $\mathcal{E}$ both defined on the same variables as the original problem $\tilde{\mathbf{w}}, \mathbf{z}$ through the Moreau envelopes of $g$ and $r$:

$$\mathcal{E}(\tau_1, \tau_2, \kappa, \eta, \nu, m) = \frac{\kappa\tau_1}{2} - \frac{\eta\tau_2}{2} + m\nu\sqrt{\gamma} - \frac{\tau_2}{2\eta}\frac{m^2}{\rho} - \frac{\eta}{2\tau_2}(\nu^2\chi + \kappa^2) + \alpha\mathcal{L}_g(\tau_1, \kappa, m, \eta) + \mathcal{L}_f(\tau_2, \eta, \nu, \kappa)$$
$$\tag{234}$$

$$\mathcal{E}_n(\tau_1, \tau_2, \kappa, \eta, \nu, m) = \frac{\kappa\tau_1}{2} - \frac{\eta\tau_2}{2} + m\nu\sqrt{\gamma} - \frac{\tau_2}{2\eta}\frac{m^2}{\rho} - \frac{\eta}{2\tau_2 d}(\nu\tilde{\mathbf{v}} + \kappa\Omega^{1/2}\mathbf{g})^\top\Omega^{-1}(\nu\mathbf{v} + \kappa\Omega^{1/2}\mathbf{g})$$

$$- \kappa\mathbf{g}^\top\left(\Sigma^{1/2} - \Omega^{1/2}\right)\frac{m\sqrt{\gamma}}{\|\tilde{\mathbf{v}}\|_2^2}\mathbf{v} + \frac{1}{d}\mathcal{M}_{\frac{\tau_1}{\kappa}g(.,\mathbf{y})}\left(\frac{m}{\sqrt{\rho}}\mathbf{s} + \eta\mathbf{h}\right) + \frac{1}{d}\mathcal{M}_{\frac{\eta}{\tau_2}r(\Omega^{-1/2}.)}\left(\frac{\eta}{\tau_2}\left(\nu\Omega^{-1/2}\tilde{\mathbf{v}} + \kappa\mathbf{g}\right)\right)$$
$$\tag{235}$$

Recall the variables:

$$\tilde{\mathbf{w}}^* = \mathrm{prox}_{\frac{\eta^*}{\tau_2^*}f(\Omega^{-1/2}.)}(\frac{\eta^*}{\tau_2^*}(\nu^*\mathbf{t} + \kappa^*\mathbf{g})), \qquad \mathbf{z}^* = \mathrm{prox}_{\frac{\tau_1^*}{\kappa^*}g(.,\mathbf{y})}\left(\frac{m^*}{\sqrt{\rho}}\mathbf{s} + \eta^*\mathbf{h}\right) \tag{236}$$

Denote $(\tau_1^*, \tau_2^*, \kappa^*, \eta^*, \nu^*, m^*)$ the unique solution to the optimization problem (90) and $\mathcal{E}^*$ the corresponding optimal cost. $\mathcal{E}^*$ defines a strongly convex optimization problem (due to the Moreau envelopes) on $\tilde{\mathbf{w}}, \mathbf{z}$ whose solution is given by Eq.(236). Similarly, denote $(\tau_{1,n}^*, \tau_{2,n}^*, \kappa_n^*, \eta_n^*, \nu_n^*, m_n^*)$ any solution to the optimization problem on $\mathcal{E}_n$ and $\mathcal{E}_n^*$ the corresponding optimal value. Finally, we write $E_n(\tilde{\mathbf{w}}, \mathbf{z})$ the cost function of the optimization problem on $\tilde{\mathbf{w}}, \mathbf{z}$ defined by $\mathcal{E}_n^*$ for any optimal solution $(\tau_{1,n}^*, \tau_{2,n}^*, \kappa_n^*, \eta_n^*, \nu_n^*, m_n^*)$, such that:

$$\mathcal{E}_n^* = \min_{\tilde{\mathbf{w}}, \mathbf{z}} E_n(\tilde{\mathbf{w}}, \mathbf{z}) \tag{237}$$

By the definition of Moreau envelopes, we have that $E_n(\tilde{\mathbf{w}}, \mathbf{z})$ is $\frac{\kappa_n^*}{2d\tau_{1,n}^*}$ strongly convex in $\mathbf{z}$ and $\frac{\tau_{2,n}^*}{2d\eta_n^*}$ strongly convex in $\tilde{\mathbf{w}}$. The following lemma ensures that these strong convexity constants are non-zero for any finite $n$.

**Lemma 14.** *Consider the finite size scalar optimization problem*

$$\max_{\kappa, \nu, \tau_2} \min_{m, \eta, \tau_1} \mathcal{E}_n(\tau_1, \tau_2, \kappa, \eta, \nu, m) \tag{238}$$

*where the feasibility set of $(\tau_1, \tau_2, \kappa, \eta, \nu, m)$ is compact and $\tau_1 > 0, \tau_2 > 0$. Then any optimal values $\kappa^*, \tau_2^*$ verify:*

$$\kappa^* \neq 0 \quad \tau_2^* \nrightarrow 0 \tag{239}$$

*Proof of Lemma 14*: from the analysis carried out in the proof of Lemma 11, the feasibility set of the optimization problem is compact. Suppose $\kappa^* = 0$. Then the value of $m$ minimizing the cost function is $-\infty$, which contradicts the compactness of the feasibility set. A similar argument holds for $\tau_2$. $\square$

The next lemma characterizes the speed of convergence of the optimal value of the finite dimensional scalar optimization problem to its asymptotic counterpart, which has a unique solution in $\tau_1, \tau_2, \kappa, \eta, \nu, m$. The intuition is that, using the strong convexity of the auxiliary problems, we can show that the solution in $\tilde{\mathbf{w}}, \tilde{\mathbf{z}}$ to the finite size problem $\mathcal{E}_n^*$ converges to the solution $\tilde{\mathbf{w}}^*, \tilde{\mathbf{z}}^*$ of the asymptotic problem $\mathcal{E}^*$, with convergence rates governed by those of the finite size cost towards its asymptotic counterpart.

**Lemma 15.** *For any $\epsilon > 0$, there exist constants $C, c, \gamma$ such that:*

$$\mathbb{P}\left(|\mathcal{E}_n^* - \mathcal{E}^*| \geqslant \gamma\epsilon\right) \leqslant \frac{C}{\epsilon} \exp^{-cn\epsilon^2} \tag{240}$$

*which is equivalent to*

$$\mathbb{P}\left(\left|\min_{\tilde{\mathbf{w}}, \mathbf{z}} E_n(\tilde{\mathbf{w}}, \mathbf{z}) - \mathcal{E}^*\right| \geqslant \gamma\epsilon\right) \leqslant \frac{C}{\epsilon} \exp^{-cn\epsilon^2} \tag{241}$$

*Proof of Lemma 15*: for any fixed $(\tau_1, \tau_2, \kappa, \nu, \eta, m)$, we can determine the rates of convergence of all the random quantities in $\mathcal{E}_n$. The linear terms involving $\frac{1}{d}\mathbf{g}^T\mathbf{v}$ are sub-Gaussian with sub-Gaussian norm bounded by $C/d$ for some constant $C > 0$. Thus we can find constants, $C, c > 0$ such that, for any $\epsilon > 0$:

$$\mathbb{P}\left(\left|\frac{1}{d}\mathbf{g}^T\tilde{\mathbf{v}}\right| \geqslant \epsilon\right) \leqslant Ce^{-cn\epsilon^2} \tag{242}$$

The term involving $\mathbf{v}^T\Omega\mathbf{v}$ is deterministic in this setting. We will see in section B.7 how a random $\boldsymbol{\theta}_0$ affects the convergence rates. The term involving $\frac{1}{d}\mathbf{g}^T\mathbf{g}$ is a weighted sum of sub-exponential random variables, the tail of which can be determined using Bernstein's inequality, see e.g. [74] Corollary 2.8.3, which gives a sub-Gaussian tail for small deviations and a sub-exponential tail for large deviations. Parametrizing the deviation $\epsilon$ with a scalar variable $c'$, we thus get the following bound : for any $\epsilon > 0$, there exists constants $C, c, c' > 0$ such that:

$$\mathbb{P}\left(\left|\frac{1}{d}\mathbf{g}^T\mathbf{g} - 1\right| \geqslant c'\epsilon\right) \leqslant Ce^{-cn\epsilon^2} \tag{243}$$

Since, in this case, we assume that the eigenvalues of the covariance matrices are bounded with probability one, multiplications by these matrices do not change these two previous rates. The

remaining convergence rates that need to be determined are those of the Moreau envelopes. By assumption, the function $g$ is separable, and pseudo-Lipschitz of order two. Moreover, the argument $\frac{m}{\sqrt{\rho}}\mathbf{s} + \eta\mathbf{h}$ is an i.i.d. Gaussian random vector with finite variance. The Moreau envelope $\frac{1}{d}\mathcal{M}_{\frac{\eta}{\tau_2}r(\Omega^{-1/2}.)}\left(\frac{\eta}{\tau_2}\left(\nu\Omega^{-1/2}\tilde{\mathbf{v}} + \kappa\mathbf{g}\right)\right)$ is therefore a sum of pseudo-Lipschitz functions of order 2 of scalar Gaussian random variables. Using the concentration Lemma 7, we can find constants $C, c, \gamma > 0$ such that, for any $\epsilon > 0$, the following holds:

$$\mathbb{P}\left(\left|\alpha\frac{1}{n}\mathcal{M}_{\frac{\tau_1}{\kappa}g(.,\mathbf{y})}\left(\frac{m}{\sqrt{\rho}}\mathbf{s} + \eta\mathbf{h}\right) - \mathbb{E}\left[\alpha\frac{1}{n}\mathcal{M}_{\frac{\tau_1}{\kappa}g(.,\mathbf{y})}\left(\frac{m}{\sqrt{\rho}}\mathbf{s} + \eta\mathbf{h}\right)\right]\right| \geqslant \gamma\epsilon\right) \leqslant Ce^{-cn\epsilon^2}$$

(244)

For the second Moreau envelope, the argument $\frac{\eta}{\tau_2}\left(\nu\Omega^{-1/2}\tilde{\mathbf{v}} + \kappa\mathbf{g}\right)$ is not separable. If the regularization is a square, it is the concentration will reduce to that of the terms $\frac{1}{d}\mathbf{g}^T\mathbf{v}$ and $\frac{1}{d}\mathbf{g}^T\mathbf{g}$. If the regularization is a Lipschitz function, then the Moreau envelope is also Lipschitz from Lemma 2. Furthermore, since the eigenvalues of the covariance matrix $\Omega$ are bounded with probability one, the composition with the deterministic term $\nu\Omega^{1/2}\mathbf{v}$ does not change the Lipschitz property. Gaussian concentration of Lipschitz functions then gives an exponential decay indepedent of the magnitude of the deviation. Taking the loosest bound, which is the one obtained with the square penalty, we obtain that, for any $\epsilon > 0$, there exist constants $C, c, \gamma > 0$ such that the event

$$\left\{\left|\frac{1}{d}\mathcal{M}_{\frac{\eta}{\tau_2}r(\Omega^{-1/2}.)}\left(\frac{\eta}{\tau_2}\left(\nu\Omega^{-1/2}\tilde{\mathbf{v}} + \kappa\mathbf{g}\right)\right) - \mathbb{E}\left[\frac{1}{d}\mathcal{M}_{\frac{\eta}{\tau_2}r(\Omega^{-1/2}.)}\left(\frac{\eta}{\tau_2}\left(\nu\Omega^{-1/2}\tilde{\mathbf{v}} + \kappa\mathbf{g}\right)\right)\right]\right| \geqslant \gamma\epsilon\right\}$$

(245)

has probability at most $Ce^{-cn\epsilon^2}$. Combining these bounds gives the exponential rate for the convergence of $\mathcal{E}_n$ to $\mathcal{E}$ for any fixed $(\tau_1, \tau_2, \kappa, \nu, \eta, m)$. An $\varepsilon$-net argument can then be used to obtain the bound on the minmax values. $\qquad\square$

The next lemma shows that the function $E_n$ evaluated at $\tilde{\mathbf{w}}^*, \mathbf{z}^*$ is close to the optimal value $\mathcal{E}^*$.

**Lemma 16.** *For any $\epsilon > 0$, there exist constants $C, c, \gamma$ such that:*

$$\mathbb{P}\left(|E_n(\tilde{\mathbf{w}}^*, \mathbf{z}^*) - \mathcal{E}^*| \geqslant \gamma\epsilon\right) \leqslant Ce^{-cn\epsilon^2}$$

(246)

*Proof of Lemma 16*: this Lemma can be proved in similar fashion to [51] Theorem B.1. using the strong convexity in $\tilde{\mathbf{w}}$ and $\mathbf{z}$ of $E_n(\tilde{\mathbf{w}}, \mathbf{z})$ along with Gordon's Lemma. We leave the detail of this part to a longer version of this paper.

**Lemma 17.** *For any $\epsilon > 0$, there exists constants $\gamma, c, C > 0$ such that the event*

$$\exists(\tilde{\mathbf{w}}, \mathbf{z}) \in \mathbb{R}^{n+d},\ \frac{1}{d}\min(\frac{\kappa_n^*}{2\tau_{1,n}^*}, \frac{\tau_{2,n}^*}{2\eta_n^*})\|(\tilde{\mathbf{w}}, \mathbf{z}) - (\tilde{\mathbf{w}}^*, \mathbf{z}^*)\|_2^2 > \epsilon\ \text{and}\ \min_{\tilde{\mathbf{w}}, \mathbf{z}} E_n(\tilde{\mathbf{w}}, \mathbf{z}) \leqslant E_n(\tilde{\mathbf{w}}^*, \mathbf{z}^*) + \gamma\epsilon$$

(247)

*has probability at most $\frac{C}{\epsilon}e^{-cn\epsilon^2}$.*

This lemma can be proven using the same arguments as in [51] Appendix B, Theorem B.1. Intuitively, if two values of a strongly convex function are arbitrarily close, then the corresponding points are arbitrarily close. Note that we are normalizing the norm of a vector of size $(n + d)$ with $d$, which are proportional. This shows that any solution outside the ball centered around $\tilde{\mathbf{w}}^*, \mathbf{z}^*$ is sub-optimal. Now define the set:

$$D_{\tilde{\mathbf{w}}, \mathbf{z}, \epsilon} = \left\{\tilde{\mathbf{w}} \in \mathbb{R}^d, \mathbf{z} \in \mathbb{R}^n : \left|\phi_1(\frac{\tilde{\mathbf{w}}}{\sqrt{d}}) - \mathbb{E}\left[\phi_1\left(\frac{\tilde{\mathbf{w}}^*}{\sqrt{d}}\right)\right]\right| > \epsilon,\ \left|\phi_2(\frac{\mathbf{z}}{\sqrt{n}}) - \mathbb{E}\left[\phi_2\left(\frac{\mathbf{z}^*}{\sqrt{n}}\right)\right]\right| > \epsilon\right\}$$

(248)

where $\phi_1$ is either a square or a Lipschitz function, and $\phi_2$ is a separable, pseudo-Lipschitz function of order 2. Using the same arguments as in the proof of Lemma 16 and the assumptions on $\phi_1, \phi_2$, Gaussian concentration will give sub-exponential rates for the event $(\tilde{\mathbf{w}}^*, \mathbf{z}^*) \in D_{\tilde{\mathbf{w}}, \mathbf{z}, \epsilon}$. A similar argument to the proof of Lemma B.3 from [29] then shows that a distance of $\epsilon$ in $D_{\tilde{\mathbf{w}}, \mathbf{z}, \epsilon}$ results in a distance of $\epsilon^2$ in the event (249), leading to the following result:

**Lemma 18.** *For any $\epsilon > 0$, there exists constants $\gamma, c, C > 0$ such that the event*

$$\exists (\tilde{\mathbf{w}}, \mathbf{z}) \in \mathbb{R}^{n+d}, (\tilde{\mathbf{w}}^*, \mathbf{z}^*) \in D_{\tilde{\mathbf{w}}, \mathbf{z}, \epsilon} \text{ and } \min_{\tilde{\mathbf{w}}, \mathbf{z}} E_n(\tilde{\mathbf{w}}, \mathbf{z}) \leqslant E_n(\tilde{\mathbf{w}}^*, \mathbf{z}^*) + \gamma \epsilon^2 \tag{249}$$

*has probability at most $\frac{C}{\epsilon^2} e^{-cn\epsilon^4}$.*

which proves Theorem 5 using the fact that $\hat{\mathbf{w}}, \hat{\mathbf{z}}$ are minimizers of the initial cost function. Theorem 4 is a consequence of Theorem 5.

If the restriction on $f, g, \phi_1, \phi_2$ are relaxed to any pseudo-Lipschitz functions of finite orders, the exponential rates involving them are lost and become linear following Lemma 5.

### B.7 Relaxing the deterministic teacher assumption

The entirety of the previous proof has been done with a deterministic vector $\boldsymbol{\theta}_0$. Now, if $\boldsymbol{\theta}_0$ is assumed to be a random vector independent of all other quantities, as prescribed in the set of assumptions B.1, we can "freeze" the variable $\boldsymbol{\theta}_0$ by conditioning on it. The whole proof can then be understood as studying the value of the cost conditioned on the value of $\boldsymbol{\theta}_0$. Note that, in the Gaussian case, correlations between the teacher and student are expressed through the covariance matrices, thus leaving the possibility to parametrise the teacher with a vector $\boldsymbol{\theta}_0$ indeed independent of all the rest. To lift the conditioning in the end, one only needs to average out on the distribution of $\boldsymbol{\theta}_0$, the summability conditions of which are prescribed in the set of assumptions B.1. Thus, random teacher vectors can be treated simply by taking an additional expectation in the expressions of Theorem 5, provided $\boldsymbol{\theta}_0$ is independent of the matrices $A, B$ and the randomness in $f_0$.

As mentioned at the end of the previous section, the finite size rates will be determined by the assumptions made on the teacher vector and decay of the eigenvalues of the covariance matrices. We do not investigate in detail the limiting assumptions under which exponential rates still hold regarding the randomness of the teacher or tails of the eigenvalue distributions of covariance matrices.

### B.8 The 'vanilla' teacher-student scenario

In this section, we give the explicit forms of the fixed points equations and optimal asymptotic estimators in the case where the teacher and the student are sampled from the same distribution, i.e. $\Omega = \Phi = \Psi = \Sigma$ where $\Sigma$ is a positive definite matrix with sub-Gaussian eigenvalue decay. This setup was rigorously studied in [29] for the LASSO and heuristically in [33] for the ridge regularized logistic regression. In this case, the fixed point equations become

$$\tau_1 = \frac{1}{d} \mathbb{E}\left[ \mathbf{g}^\top \text{prox}_{\frac{\eta}{\tau_2} f(\Sigma^{-1/2}.)}\left( \frac{\eta}{\tau_2}\left( \nu \Sigma^{1/2}\boldsymbol{\theta}_0 + \kappa \mathbf{g} \right) \right) \right] \tag{250}$$

$$m\sqrt{\gamma} = \frac{1}{d} \mathbb{E}\left[ \mathbf{v}^\top \Sigma^{-1/2} \text{prox}_{\frac{\eta}{\tau_2} f(\Sigma^{-1/2}.)}\left( \frac{\eta}{\tau_2}\left( \nu \Sigma^{1/2}\boldsymbol{\theta}_0 + \kappa \mathbf{g} \right) \right) \right] \tag{251}$$

$$\tau_2 = \alpha \frac{\kappa}{\tau_1} \eta - \frac{\kappa \alpha}{\tau_1 n} \mathbb{E}\left[ \mathbf{h}^\top \text{prox}_{\frac{\tau_1}{\kappa} g(.,\mathbf{y})}\left( \frac{m}{\sqrt{\rho}} \mathbf{s} + \eta \mathbf{h} \right) \right] \tag{252}$$

$$\eta^2 + \frac{m^2}{\rho} = \frac{1}{d} \mathbb{E}\left[ \left\| \text{prox}_{\frac{\eta}{\tau_2} f(\Sigma^{-1/2}.)}\left( \frac{\eta}{\tau_2}\left( \nu \Sigma^{1/2}\boldsymbol{\theta}_0 + \kappa \mathbf{g} \right) \right) \right\|_2^2 \right] \tag{253}$$

$$\nu\sqrt{\gamma} = \alpha \frac{\kappa}{n\tau_1} \mathbb{E}\left[ \left( \frac{m}{\eta\rho} \mathbf{h} - \frac{\mathbf{s}}{\sqrt{\rho}} \right)^\top \text{prox}_{\frac{\tau_1}{\kappa} g(.,\mathbf{y})}\left( \frac{m}{\sqrt{\rho}} \mathbf{s} + \eta \mathbf{h} \right) \right] \tag{254}$$

$$\tau_1^2 = \frac{\alpha}{n} \mathbb{E}\left[ \left\| \frac{m}{\sqrt{\rho}} \mathbf{s} + \eta \mathbf{h} - \text{prox}_{\frac{\tau_1}{\kappa} g(.,y)}\left( \frac{m}{\sqrt{\rho}} \mathbf{s} + \eta \mathbf{h} \right) \right\|_2^2 \right] \tag{255}$$

and the asymptotic optimal estimators read:

$$\mathbf{w}^* = \Sigma^{-1/2} \text{prox}_{\frac{\eta^*}{\tau_2^*} f(\Sigma^{-1/2}.)}\left( \frac{\eta^*}{\tau_2^*}(\nu^* \Sigma^{1/2}\boldsymbol{\theta}_0 + \kappa^* \mathbf{g}) \right), \quad \mathbf{z}^* = \text{prox}_{\frac{\tau_1^*}{\kappa^*} g(.,\mathbf{y})}\left( \frac{m^*}{\sqrt{\rho}} \mathbf{s} + \eta^* \mathbf{h} \right) \tag{256}$$

# C  Equivalence replica-Gordon

In this Appendix, we show that the rigorous result of Theorem 5 can be used to prove the replica prediction in the case of a separable loss, a ridge penalty. For simplicity, we restrict ourselves to the case of random teacher weights with $\boldsymbol{\theta}_0 \sim \mathcal{N}(0, \mathbf{I}_p)$. We provide an exact analytical matching between the replica prediction and the one obtained with Gordon's theorem. We start by an explicit derivation of the form presented in Corollary 1 from the main result (89).

## C.1  Solution for separable loss and ridge regularization

Replacing $r$ with a ridge penalty, we can go back to step (176) of the main proof and finish the calculation without inverting the matrix $\Omega$. The assumption on the invertibility of $\Omega$ can thus be dropped in the case of $\ell_2$ regularization. Letting $G = \left( \frac{\tau_2}{\eta}\Omega + \lambda_2 \mathbf{I}_d \right)^{-1}$, we get

$$
\mathcal{E}(\tau_1, \tau_2, \kappa, \eta, \nu, m) = \frac{\kappa \tau_1}{2} - \frac{\eta \tau_2}{2} + m\nu\sqrt{\gamma} - \frac{\tau_2}{2\eta}\frac{m^2}{\rho} + \alpha\frac{1}{n}\mathbb{E}\left[ \mathcal{M}_{\frac{\tau_1}{\kappa}g(.,\mathbf{y})}\left( \frac{m}{\sqrt{\rho}}\mathbf{s} + \eta\mathbf{h} \right) \right]
$$
$$
- \frac{1}{2d}\nu^2\boldsymbol{\theta}_0^\top\Phi G\Phi^\top\boldsymbol{\theta}_0 - \frac{1}{2d}\kappa^2\mathrm{Tr}\left( \Omega^{1/2}G\Omega^{1/2} \right) \tag{257}
$$

using Lemma 5 with a separable function, the expectation over the Moreau envelope converges to:

$$
\frac{1}{n}\mathbb{E}\left[ \mathcal{M}_{\frac{\tau_1}{\kappa}g(.,\mathbf{y})}\left( \frac{m}{\sqrt{\rho}}\mathbf{s} + \eta\mathbf{h} \right) \right] = \mathbb{E}\left[ \mathcal{M}_{\frac{\tau_1}{\kappa}g(.,y)}\left( \frac{m}{\sqrt{\rho}}s + \eta h \right) \right] \tag{258}
$$

where $s$ and $h$ are standard normal random variables and $y = f_0(\sqrt{\rho}s)$. The corresponding optimality conditions then reads:

$$
\frac{\partial}{\partial\kappa} : \frac{\tau_1}{2} + \frac{1}{2\tau_1}\alpha\mathbb{E}\left[ \left( \frac{m}{\sqrt{\rho}}s + \eta h - \mathrm{prox}_{\frac{\tau_1}{\kappa}g(.,y)}\left( \frac{m}{\sqrt{\rho}}s + \eta h \right) \right)^2 \right] - \kappa\frac{1}{d}\mathrm{Tr}\left( \Omega^{1/2}G\Omega^{1/2} \right) = 0 \tag{259}
$$

$$
\frac{\partial}{\partial\nu} : m\sqrt{\gamma} - \frac{1}{d}\nu\boldsymbol{\theta}_0\Phi^\top G\Phi^\top\boldsymbol{\theta}_0 = 0 \tag{260}
$$

$$
\frac{\partial}{\partial\tau_2} : -\frac{\eta}{2} - \frac{m^2}{2\rho\eta} + \frac{1}{2}\frac{\nu^2}{\eta}\left( \Omega^{1/2}\Phi^\top\boldsymbol{\theta}_0 \right)^\top G^2\Omega^{1/2}\Phi^\top\boldsymbol{\theta}_0 + \frac{\kappa^2}{2\eta}Tr\left( G^2\Omega^2 \right) = 0 \tag{261}
$$

$$
\frac{\partial}{\partial m} : \nu\sqrt{\gamma} - \frac{\tau_2}{\rho\eta}m + \alpha\mathbb{E}\left[ \frac{\kappa}{\tau_1}\frac{s}{\sqrt{\rho}}(\frac{m}{\sqrt{\rho}}s + \eta h - \mathrm{prox}_{\frac{\tau_1}{\kappa}g(.,y)}\left( \frac{m}{\sqrt{\rho}}s + \eta h \right)) \right] = 0 \tag{262}
$$

$$
\frac{\partial}{\partial\eta} : -\frac{\tau_2}{2} + \frac{\tau_2 m^2}{2\rho\eta^2} + \alpha\mathbb{E}\left[ \frac{\kappa}{\tau_1}h\left( \frac{m}{\sqrt{\rho}}s + \eta h - \mathrm{prox}_{\frac{\tau_1}{\kappa}g(.,y)}\left( \frac{m}{\sqrt{\rho}}s + \eta h \right) \right) \right]
$$
$$
- \frac{1}{2}\frac{\tau_2\nu^2}{\eta^2}\left( \Omega^{1/2}\Phi^\top\boldsymbol{\theta}_0 \right)^\top G^2\Omega^{1/2}\Phi^\top\boldsymbol{\theta}_0 - \frac{\tau_2\kappa^2}{2\eta^2}\mathrm{Tr}(G^2\Omega^2) = 0 \tag{263}
$$

$$
\frac{\partial}{\partial\tau_1} : \frac{\kappa}{2} - \frac{\kappa}{2\tau_1^2}\alpha\mathbb{E}\left[ \left( \frac{m}{\sqrt{\rho}}s + \eta h - \mathrm{prox}_{\frac{\tau_1}{\kappa}g(.,y)}\left( \frac{m}{\sqrt{\rho}}s + \eta h \right) \right)^2 \right] = 0 \tag{264}
$$

simplifying these equations using Stein's lemma, we get:

$$\frac{\partial}{\partial \kappa} : \frac{\tau_1}{\kappa} = \frac{1}{d} \mathrm{Tr}\left( \Omega^{1/2} \left( \frac{\tau_2}{\eta}\Omega + \lambda_2 \mathbf{I_d} \right)^{-1} \Omega^{1/2} \right) \tag{265}$$

$$\frac{\partial}{\partial \nu} : m\sqrt{\gamma} = \frac{1}{d}\nu\boldsymbol{\theta}_0\Phi \left( \frac{\tau_2}{\eta}\Omega + \lambda_2 \mathbf{I}_d \right)^{-1} \Phi^\top \boldsymbol{\theta}_0 \tag{266}$$

$$\frac{\partial}{\partial \tau_2} : \eta^2 + \frac{m^2}{\rho} = \frac{1}{d}\nu^2 \left( \Omega^{1/2}\Phi^\top\boldsymbol{\theta}_0 \right)^\top \left( \frac{\tau_2}{\eta}\Omega + \lambda_2 \mathbf{I}_d \right)^{-2} \left( \Omega^{1/2}\Phi^\top\boldsymbol{\theta}_0 \right) + \frac{1}{d}\kappa^2 \mathrm{Tr}\!\left( \left( \frac{\tau_2}{\eta}\Omega + \lambda_2 \mathbf{I}_d \right)^{-2} \Omega^2 \right) \tag{267}$$

$$\frac{\partial}{\partial m} : \nu\sqrt{\gamma} = \alpha\frac{\kappa}{\sqrt{\rho}\tau_1}\left( \mathbb{E}\left[ s\mathrm{prox}_{\frac{\tau_1}{\kappa}g(.,f_0(\sqrt{\rho}s))}\left( \frac{m}{\sqrt{\rho}} + \eta h \right) \right] - \frac{m}{\sqrt{\rho}}\mathbb{E}\left[ \mathrm{prox}'_{\frac{\kappa}{\tau_1}g(.,f_0(\sqrt{\rho}s))}\left( \frac{m}{\sqrt{\rho}}s + \eta h \right) \right] \right) \tag{268}$$

$$\frac{\partial}{\partial \eta} : \frac{\tau_2}{\eta} = \alpha\frac{\kappa}{\tau_1}\left( 1 - \mathbb{E}\left[ \mathrm{prox}'_{\frac{\tau_1}{\kappa}g(.,f_0(\sqrt{\rho}s))}\left( \frac{m}{\sqrt{\rho}}s + \eta h \right) \right] \right) \tag{269}$$

$$\frac{\partial}{\partial \tau_1} : \kappa^2 = \left( \frac{\kappa}{\tau_1} \right)^2 \alpha\mathbb{E}\left[ \left( \frac{m}{\sqrt{\rho}}s + \eta h - \mathrm{prox}_{\frac{\tau_1}{\kappa}g(.,f_0(\sqrt{\rho}s))}\left( \frac{m}{\sqrt{\rho}}s + \eta h \right) \right)^2 \right] \tag{270}$$

## C.2 Matching with Replica equations

In this section, we show that the fixed point equations obtained from the asymptotic optimality condition of the scalar minimization problem 1 match the ones obtained using the replica method. In what follows we will use the same notations as in [43], and an explicit, clear match with the notations from the proof of the main theorem will be shown. The replica computation, similar to the one from [43], leads to the following fixed point equations, in the replica notations:

$$V = \frac{1}{p}\mathrm{Tr}\left( \lambda\hat{V}I_p + \Omega \right)^{-1}\Omega \tag{271}$$

$$q = \frac{1}{p}\mathrm{Tr}\left[ (\hat{q}\Omega + \hat{m}^2\Phi^\top\Phi)\Omega\left( \lambda\hat{V}I_p + \Omega \right)^{-2} \right] \tag{272}$$

$$m = \frac{1}{\sqrt{\gamma}}\frac{\hat{m}}{p}\mathrm{Tr}\left[ \Phi^\top\Phi\left( \lambda\hat{V}I_p + \Omega \right)^{-1} \right] \tag{273}$$

$$\hat{V} = \alpha\mathbb{E}_\xi\left[ \int_\mathbb{R} \mathrm{d}y\, \mathcal{Z}_y^0\left( y, \frac{m}{\sqrt{q}}, \rho - \frac{m^2}{q} \right)\partial_\omega f_g(y, \sqrt{q}\xi, V) \right] \tag{274}$$

$$\hat{q} = \alpha\mathbb{E}_\xi\left[ \int_\mathbb{R} \mathrm{d}y\, \mathcal{Z}_y^0\left( y, \frac{m}{\sqrt{q}}, \rho - \frac{m^2}{q} \right) f_g(y, \sqrt{q}\xi, V)^2 \right] \tag{275}$$

$$\hat{m} = \frac{\alpha}{\sqrt{\gamma}}\mathbb{E}_\xi\left[ \int_\mathbb{R} \mathrm{d}y\, \partial_\omega\mathcal{Z}_y^0\left( y, \frac{m}{\sqrt{q}}, \rho - \frac{m^2}{q} \right) f_g(y, \sqrt{q}\xi, V) \right] \tag{276}$$

where $f_g(y, \omega, V) = -\partial_\omega\mathcal{M}_{Vg(y,\cdot)}(\omega)$ and $\mathcal{Z}_0$ is given by:

$$\mathcal{Z}_0\left( y, \omega, V \right) = \int \frac{\mathrm{d}x}{\sqrt{2\pi V}}e^{-\frac{1}{2V}(x-\omega)^2}\delta(y - f^0(x)). \tag{277}$$

In particular we have:

$$\partial_\omega\mathcal{Z}_0\left( y, \omega, V \right) = \int \frac{\mathrm{d}x}{\sqrt{2\pi V}}e^{-\frac{1}{2V}(x-\omega)^2}\left( \frac{x - \omega}{V} \right)\delta(y - f^0(x)) \tag{278}$$

To be explicit with the notation, let's open the equations up. Take for instance the one for $\hat{m}$. Opening all the integrals:

$$\hat{m} = \int \frac{\mathrm{d}\xi}{\sqrt{2\pi}} e^{-\frac{1}{2}\xi^2} \int \mathrm{d}y \int \frac{\mathrm{d}x}{\sqrt{2\pi\left(\rho - m^2/q\right)}} e^{-\frac{1}{2}\frac{\left(x - \frac{m}{\sqrt{q}}\xi\right)^2}{\rho - m^2/q}} \left(\frac{x - \frac{m}{\sqrt{q}}\xi}{\rho - m^2/q}\right) f_g(y, \sqrt{q}\xi, V)$$

$$\stackrel{(a)}{=} \int \frac{\mathrm{d}\xi}{\sqrt{2\pi}} e^{-\frac{1}{2}\xi^2} \int \frac{\mathrm{d}x}{\sqrt{2\pi\left(\rho - m^2/q\right)}} e^{-\frac{1}{2}\frac{\left(x - \frac{m}{\sqrt{q}}\xi\right)^2}{\rho - m^2/q}} \left(\frac{x - \frac{m}{\sqrt{q}}\xi}{\rho - m^2/q}\right) f_g(f_0(x), \sqrt{q}\xi, V)$$

(279)

where in $(a)$ we integrated over $y$ explicitly. A direct comparison between the two sets of equations suggests the following mapping to navigate between the replica derivation and the proof using Gaussian comparison theorems. We denote replica quantities with *Rep* indices:

$$V_{Rep} \iff \frac{\tau_1}{\kappa}, \qquad \hat{V}_{Rep} \iff \frac{\tau_2}{\eta}, \qquad q_{Rep} \iff \eta^2 + \frac{m^2}{\rho}$$

$$\hat{q}_{Rep} \iff \kappa^2, \qquad m_{Rep} \iff m, \qquad \hat{m}_{Rep} \iff \nu \qquad (280)$$

with these notations, we get :

$$\frac{\partial}{\partial\kappa} : V = \frac{1}{d}\mathrm{Tr}((\hat{V}\Omega + \lambda_2\mathbf{I}_\mathrm{d})^{-1}\Omega) \tag{281}$$

$$\frac{\partial}{\partial\nu} : m = \frac{1}{\sqrt{\gamma}}\frac{\hat{m}}{d}\mathrm{Tr}((\hat{V}\Omega + \lambda_2\mathbf{I}_\mathrm{d})^{-1}\Phi^\top\Phi) \tag{282}$$

$$\frac{\partial}{\partial\tau_2} : q = \frac{1}{d}\mathrm{Tr}((\hat{q}\Omega + \hat{m}^2\Phi^\top\Phi)\Omega(\hat{V}\Omega + \lambda_2\mathbf{I}_\mathrm{d})^{-2}) \tag{283}$$

$$\frac{\partial}{\partial m} : \hat{m} = \frac{\alpha}{\sqrt{\gamma}}\frac{1}{V}\left(\mathbb{E}\left[\frac{s}{\sqrt{\rho}}\mathrm{prox}_{Vg(.,f_0(\sqrt{\rho}s))}\left(\frac{m}{\sqrt{\rho}}s + \sqrt{q - \frac{m^2}{\rho}}h\right)\right]\right.$$

$$\left. - \frac{m}{\rho}\mathbb{E}\left[\mathrm{prox}'_{Vg(.,f_0(\sqrt{\rho}s))}\left(\frac{m}{\sqrt{\rho}}s + \sqrt{q - \frac{m^2}{\rho}}h\right)\right]\right) \tag{284}$$

$$\frac{\partial}{\partial\eta} : \hat{V} = \frac{\alpha}{V}\left(1 - \mathbb{E}\left[\mathrm{prox}'_{Vg(.,f_0(\sqrt{\rho}s))}\left(\frac{m}{\sqrt{\rho}}s + \sqrt{q - \frac{m^2}{\rho}}h\right)\right]\right) \tag{285}$$

$$\frac{\partial}{\partial\tau_1} : \hat{q} = \left(\frac{\alpha}{V^2}\right)\mathbb{E}\left[\left(\frac{m}{\sqrt{\rho}}s + \sqrt{q - \frac{m^2}{\rho}}h - \mathrm{prox}_{Vg(.,f_0(\sqrt{\rho}s))}\left(\frac{m}{\sqrt{\rho}}s + \sqrt{q - \frac{m^2}{\rho}}h\right)\right)^2\right] \tag{286}$$

The first three equations match the replica prediction, the last three can be exactly matched using the following change of variable and Gaussian integration:

$$\tilde{x} = \frac{x}{\sqrt{\rho}} \quad \tilde{\xi} = \left(\frac{\rho}{\rho - \frac{m^2}{q}}\right)^{1/2}\left(\frac{m}{\sqrt{q\rho}}\tilde{x} - \xi\right) \tag{287}$$

# D Details on the simulations

In this Appendix we give full details on the numerics used to generate the plots in the main manuscript. An implementation of all the pipelines described below is available at `https://github.com/IdePHICS/GCMProject`.

## D.1 Ridge regression on real data

Consider a real data set $\{\boldsymbol{x}^\mu, y^\mu\}_{\mu=1}^{n_{\text{tot}}}$, where $n_{\text{tot}}$ denote the total number of samples available. In Figs. 4 and 4 we work with the MNIST and fashion MNIST data sets for which $n_{\text{tot}} = 6 \times 10^4$ and $D = 28 \times 28 = 764$. In both cases, we center the data and normalise by dividing it by the global standard deviation. We work with binary labels $y^\mu \in \{-1, 1\}$, with $y^\mu = 1$ for even digits (MNIST) or clothes above the waist (fashion MNIST) and $y^\mu = -1$ for odd digitis (MNIST) or clothes below the waist (fashion MNIST). In a ridge regression task, we assume $y^\mu = \boldsymbol{\theta}_0^\top \boldsymbol{u}^\mu$ for a teacher feature map $\boldsymbol{u}^\mu = \boldsymbol{\varphi}_t(\boldsymbol{x}^\mu)$ and we are interested in studying the performance of the estimator $\hat{y} = \boldsymbol{v}^\top \hat{\boldsymbol{w}}$ where $\boldsymbol{v} = \boldsymbol{\varphi}_s(\boldsymbol{x})$ obtained by solving the empirical risk minimisation problem in eq. (24) with the squared loss $g(x, y) = \frac{1}{2}(y - x)^2$ and $\ell_2$ regularisation $\lambda > 0$.

**Simulations:** First, we discuss in detail how we conducted the numerical simulations in Figs. 4 and 4 in the main manuscript.

In Fig. 4, the student feature maps $\boldsymbol{\varphi}_s$ is taken to be different transforms used in the literature. For the scattering transform, we have used the out-of-the-box python package `Kymatio` [18] with hyperparameters $J = 3$ and $L = 8$, which defines a feature map $\boldsymbol{\varphi}_s : \mathbb{R}^{28 \times 28} \to \mathbb{R}^{217 \times 3 \times 3}$, and thus $d = 1953$. For the random features, a random matrix $\mathrm{F} \in \mathbb{R}^{d \times 784}$ with i.i.d. $\mathcal{N}(0, 1/784)$ entries is generated and fixed. Note that the number of features $d = 1953$ is chosen to match the ones for the scattering transform. The random feature map is then applied to the flattened MNIST image as $\boldsymbol{\varphi}_s(\boldsymbol{x}) = \mathrm{erf}\,(\mathrm{Fx})$. Finally, we have chosen a kernel corresponding to the limit of this random feature map [23]:

$$K(\boldsymbol{x}_1, \boldsymbol{x}_2) = \frac{2}{\pi} \sin^{-1} \left( \frac{2\boldsymbol{x}_1^\top \boldsymbol{x}_2}{\sqrt{(1/d + 2||\boldsymbol{x}_1||_2^2)\,(1/d + 2||\boldsymbol{x}_2||_2^2)}} \right). \tag{288}$$

In Fig. 4, the feature $\boldsymbol{\varphi}_s^t$ is taken from a learned neural network at different epochs $t \in \{0, 5, 50, 200\}$ of training. For this experiment, we chose the following architecture implemented in `Pytorch`:

```
Sequential(
  (0): Linear(in_features=784, out_features=2352, bias=False)
  (1): ReLU()
  (2): Linear(in_features=2352, out_features=2352, bias=False)
  (3): ReLU()
  (4): Linear(in_features=2352, out_features=1, bias=False)
)
```

The first two layers of the network therefore defines a feature map $\boldsymbol{\varphi}_s : \mathbb{R}^{784} \to \mathbb{R}^{2352}$ acting on flattened fashion MNIST images. The network was initialized using the `pyTorch`'s default Kaiming initialisation [75] and was trained on the full data set ($n_{\text{tot}}$ samples) with Adam [76] optimiser (learning rate $10^{-3}$) on the MSE loss for a total of 500 epochs. Snapshots were taken at epochs $t \in \{0, 5, 50, 200\}$, defining the feature maps $\boldsymbol{\varphi}_s^t(\cdot)$ at each of these epochs.

In both experiments, we ran ridge regression at fixed regularisation $\lambda > 0$ by sub-sampling $n$ samples from the data set $\mathcal{D} = \{\boldsymbol{v}^\mu, y^\mu\}_{\mu=1}^{n_{\text{tot}}}$, $\boldsymbol{v}^\mu = \boldsymbol{\varphi}_s(\boldsymbol{x}^\mu)$, with the estimator given by the closed-form expression:

$$\hat{\boldsymbol{w}} = \begin{cases} \left(\lambda \mathrm{I}_d + \mathrm{V}^\top \mathrm{V}\right)^{-1} \mathrm{V}^\top \boldsymbol{y}, & \text{if } n \geq d \\ \mathrm{V}^\top \left(\lambda \mathrm{I}_n + \mathrm{V}\mathrm{V}^\top\right)^{-1} \boldsymbol{y}, & \text{if } n < d \end{cases} \tag{289}$$

where $\mathrm{V} \in \mathbb{R}^{n \times d}$ is the normalised matrix obtained by concatenating $\{\boldsymbol{v}^\mu / \sqrt{d}\}_{\mu=1}^n$. A similar closed-form expression in terms of the Gram matrix was used in the kernel case. The averaged training and test errors were computed over 10 independent draws sub-samples of $\mathcal{D}$. To reduce the

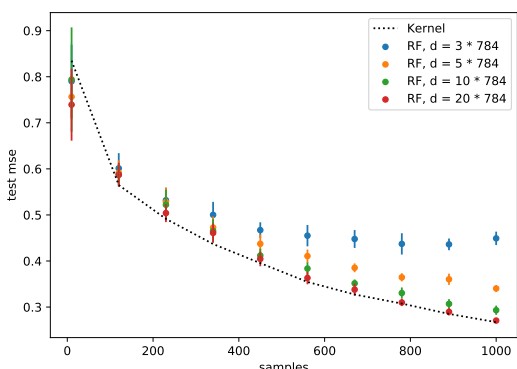

Figure 5: Test error as a function of the number of samples for kernel ridge regression task on MNIST odd vs. even data, with $\lambda = 10^{-1}$. The different curves compare the performance of a random features approximation $\varphi_s(x) = \text{erf}(Fx)$ with the performance of the limiting kernel eq. (288). Different curves correspond to different aspect ratios of the Gaussian projection matrix $F \in \mathbb{R}^{d \times 784}$.

effect spurious correlations due to the sampling of a finite universe $\mathcal{D}$, we have always evaluated the test error on the whole universe $\mathcal{D}$. The code for these two experiments is available in `https://github.com/IdePHICS/GCMProject`.

**Self-consistent equations:** For the theoretical curves, we need to provide the population covariances $(\Omega, \Phi, \Psi)$ and the teacher weights $\theta_0 \in \mathbb{R}^p$ corresponding to the task of interest. Since when dealing with real data we have a limited number of samples $n_{\text{tot}}$ at our disposal, we estimate the population covariances by the empirical covariances on the whole universe:

$$\Psi = \frac{1}{n_{\text{tot}}} \sum_{\mu=1}^{n_{\text{tot}}} u^\mu u^{\mu\top}, \qquad \Phi = \frac{1}{n_{\text{tot}}} \sum_{\mu=1}^{n_{\text{tot}}} u^\mu v^{\mu\top}, \qquad \Omega = \frac{1}{n_{\text{tot}}} \sum_{\mu=1}^{n_{\text{tot}}} v^\mu v^{\mu\top}. \qquad (290)$$

In principle, the teacher weights need to be estimated by inverting $y = U\theta_0$. However, as explained in $n_{\text{tot}}$ in Sec. 3.4, one can avoid doing so by noting the teacher weights only appear in the self-consistent equations 13 through $\rho = \frac{1}{k}\theta_0^\top \Psi \theta_0$ and $\Phi^\top \theta_0$. Therefore, *all teacher vector $\theta_0$ and feature map $\varphi_s$ that linearly interpolate the data set $\{x^\mu, y^\mu\}_{\mu=1}^{n_{\text{tot}}}$ are equivalent*, since we can write:

$$\rho = \frac{1}{n_{\text{tot}}} \sum_{\mu=1}^{n_{\text{tot}}} (y^\mu)^2, \qquad\qquad \Phi^\top \theta_0 = \frac{1}{n_{\text{tot}}} \sum_{\mu=1}^{n_{\text{tot}}} v^\mu y^\mu. \qquad (291)$$

which is independent from $(\varphi_t, \theta_0)$. In particular, note that for our binary labels $y^\mu \in \{+1, -1\}$, we have $\rho = 1$. In both Fig. 4 and 4 of the main, we estimated the covariance $\Omega$ as in eq. (290) by applying the feature maps $\varphi_s$ described above to the whole data set, took $\rho = 1$ (since in both we have binary labels) and used eq. (291) to estimate $\Phi^\top \theta_0$. This was then fed to our iterator package (`https://github.com/IdePHICS/GCMProject`) to compute the curves. For the kernel curve, we used the random features approximation of eq. (288) with a $d = 20 \times 1953$ dimensional feature space to estimate the covariance $\Omega$. We have checked that this indeed provide a good approximation of $K$ for the sample range considered, see Fig.5.

**Limitations:** As we have discussed above, a key ingredient of our theoretical analysis is the estimation of the population covariances. For real data, this relies on the empirical covariance of the whole data set with $n_{\text{tot}}$ samples. We expect this approximation to be good only for $n \ll n_{\text{tot}}$ samples, as it is the case for the ranges plotted in Figs. 4 and 4. Indeed, as $n \approx n_{\text{tot}}$ we start observing deviations between the theoretical prediction and the simulations. In Fig. 6 (right) we show an example of a NTK kernel regression task on 8 vs 9 MNIST digit classification, for which $n_{\text{tot}} = 7000$. Note that while the theoretical prediction reach perfect generalisation at $n \approx n_{\text{tot}}$, the simulated error approaches a plateau. Alternatively, instead of varying the sample range, in Fig. 6 (left) we show how the matching betweem theory and simulation degrades by varying $n_{\text{tot}}$ on a fixed sample range for a MNIST odd vs. even task.

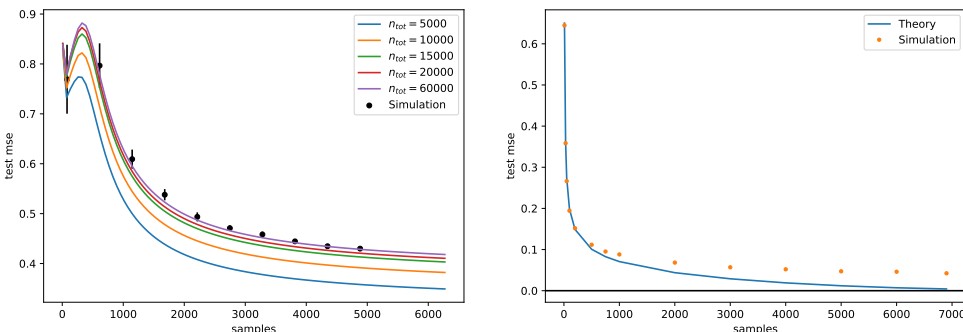

Figure 6: (Left) Test mse for ridge regression on MNIST odd vs. even task and $\lambda = 0.01$. Different curves show the theoretical prediction when the population covariances are estimated using a smaller number of samples $n_{\text{tot}}$ in the universe. (Right) Test mse for NTK kernel regression on MNIST 8 vs. 9 task with $\lambda = 0.01$. Note that for this task we have $n_{\text{tot}} = 7000$, and while the theoretical result predicts perfect generalisation as the number of samples approach $n_{\text{tot}}$, the true test error goes to a constant.

As it was discussed in Sec. 3.4 of the main manuscript, the universality argument sketched above is only valid in the case of a linear student. For instance, applying the same construction to a binary classification task with $f_0(x) = \hat{f}(x) = \text{sign}(x)$ lead to a mismatch between theory and experiments, as exemplified in Fig. 3 of the main for a logistic regression task on CIFAR10 gray-scale images. Interestingly, this is even the case for binary classification with the square loss $g(x, y) = \frac{1}{2}(x - y)^2$, in which the estimator $\hat{w}$ is the same as for ridge regression. In other words, by simply changing the predictor $\hat{f}(x) = \text{sign}(x)$, we have a breakdown of universality, as shown in Fig. 7.

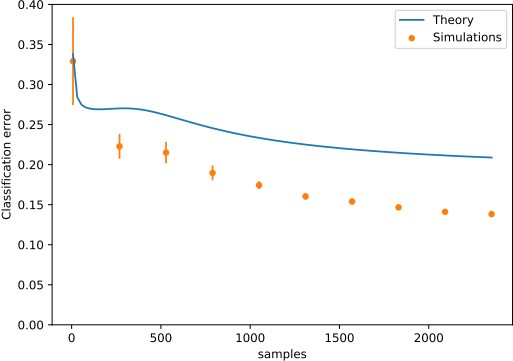

Figure 7: Classification error for binary classification task with the square loss on MNIST odd vs. even task and $\lambda = 0.01$

## D.2 Binary classification on GAN generated data

For our purposes, a generative adversarial network (GAN) is a pre-trained neural network defining a map $\mathcal{G}$ taking a Gaussian i.i.d. vector $z \sim \mathcal{N}(0, I)$ (a.k.a. the latent representation) into a realistic looking input image $x \in \mathbb{R}^D$. In both Figs. 3 and 3, we have used a deep convolutional GAN (dcGAN) [59] with the following architecture

and which has been trained on the full CIFAR10 data set. It therefore takes a 100-dimensional latent vector and returns a $D = 32 \times 32 \times 3 = 3072$ CIFAR10-looking image. The GAN was trained on the original CIFAR10 data set without data augmentation for 50 epochs. Both the discriminator and the generator were trained using Adam, with Adam parameters $\beta_1 = 0.5$ and $\beta_2 = 0.999$. In practice, the advantage of working with a GAN is that we have a generative process to sample as

many independent data points as we need, both for the simulations and for the estimation of the population covariances.

**Learning the teacher:** As discussed in Sec. 3.3 of the main manuscript, to label the GAN generated CIFAR10-looking images we learn a teacher feature map $\varphi_t$ and weights $\boldsymbol{\theta}_0 \in \mathbb{R}^p$. For the experiments shown in Figs. 3, we have trained with a fully-connected neural network on the full CIFAR10 data set with the following squared architecture:

```
Generator(
  (main): Sequential(
    (0): ConvTranspose2d(100, 512, kernel_size=(4, 4), stride=(1, 1),
                                   bias=False)
    (1): BatchNorm2d(512, eps=1e-05, momentum=0.1, affine=True,
                                   track_running_stats=True)
    (2): ReLU(inplace=True)
    (3): ConvTranspose2d(512, 256, kernel_size=(4, 4), stride=(2, 2),
                                   padding=(1, 1), bias=False)
    (4): BatchNorm2d(256, eps=1e-05, momentum=0.1, affine=True,
                                   track_running_stats=True)
    (5): ReLU(inplace=True)
    (6): ConvTranspose2d(256, 128, kernel_size=(4, 4), stride=(2, 2),
                                   padding=(1, 1), bias=False)
    (7): BatchNorm2d(128, eps=1e-05, momentum=0.1, affine=True,
                                   track_running_stats=True)
    (8): ReLU(inplace=True)
    (9): ConvTranspose2d(128, 64, kernel_size=(4, 4), stride=(2, 2),
                                   padding=(1, 1), bias=False)
    (10): BatchNorm2d(64, eps=1e-05, momentum=0.1, affine=True,
                                   track_running_stats=True)
    (11): ReLU(inplace=True)
    (12): ConvTranspose2d(64, 3, kernel_size=(1, 1), stride=(1, 1),
                                   bias=False)
    (13): Tanh()
  )
)
```

The teacher feature map $\varphi_t : \mathbb{R}^D \to \mathbb{R}^p$ was then taken to be the first 2-layers, and the teacher weights $\boldsymbol{\theta}_0$ the weights of the last layer, where $D = p = 32 \times 32 \times 3 = 3072$. We used the same architecture for the experiment in Fig. 3, but with $D = p = 32 \times 32 = 1024$ on gray-scale CIFAR10 images. Both teachers were trained on the odd-even discrimination task on CIFAR10 discussed above with the mean-squared error for 50 epochs, starting from pyTorch's default Kaiming initialisation [75] . Optimisation was performed using SGD with momentum 0.9 and weight decay $5 \cdot 10^{-4}$. We started with a learning rate of $0.05$, which decayed by a factor $0.1$ after 25 and 40 epochs. The resulting trained teacher achieved a $78\%$ classification accuracy on this task. See Fig. 8 for an illustration of this pipeline.

**Simulations:** The experiment shown in Fig. 3 follow a similar pipeline as the one described in Sec. D.1. The student feature maps $\varphi_s^t$ are obtained by removing the last layer of a trained a 3-layer student network with architecture:

```
Sequential(
  (0): Linear(in_features=1024, out_features=2304, bias=False)
  (1): ReLU()
  (2): Linear(in_features=2304, out_features=2304, bias=False)
  (3): ReLU()
  (4): Linear(in_features=2304, out_features=1, bias=False)
)
```

Training was performed on a data set composed of $n = 30000$ independent samples drawn from the dcGAN described above, with labels $y^\mu \in \{+1, -1\}$ assigned by the learned teacher $y^\mu = \text{sign}\left(\boldsymbol{u}^\top \boldsymbol{\theta}_0\right)$, $\boldsymbol{u}^\mu = \varphi_t\left(\boldsymbol{x}^\mu\right)$. The network was trained for 300 epochs using Adam optimiser on the MSE loss and pyTorch's default Kaiming initialisation, and snapshops of the weights were

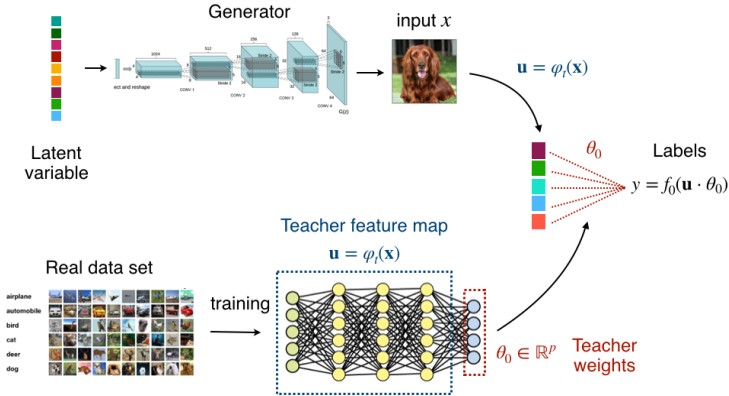

Figure 8: Illustration of the pipeline to generate synthetic realistic data. A dcGAN is first trained to generate CIFAR10-looking images from i.i.d. Gaussian noise. Then, a teacher trained to classify real CIFAR10 images is used to assign labels to the dcGAN generated images.

extracted at epochs $t \in \{0, 5, 50, 200\}$. Finally, logistic regression was performed on the learned features $v = \varphi_s(x)$ on fresh pair of dcGAN generated samples and labels using the out-of-the-box `LogisticRegression` solver from `Scikit-learn`. The points and error bars in Fig. 3 were computed by averaging over 10 independent runs. The same pipeline was used for Fig. 3, but for $\varphi_s = \mathrm{id}$ and on dcGAN generated CIFAR10 gray-scale images.

**Self-consistent equations:** As before, the self-consistent eqs. 13 require the population covariances $(\Omega, \Phi, \Psi)$ and the teacher weights $\theta_0$. For synthetic GAN data, the population covariances of the feature maps $(\varphi_t, \varphi_s^t)$ used in the simulations can be estimated as well as needed with a Monte Carlo sampling algorithm. For the curves shown in Figs. 3 and 3, the covariances were estimated with $n = 10^6$ samples with a precision of the order of $10^{-5}$. Together with the teacher weights $\theta_0$ used to generate the labels, this provides everything needed to compute the theoretical learning curves from the self-consistent equations.

# E    Ridge regression with linear teachers

In this Appendix we discuss briefly random matrix theory, and consider heuristic reasons behind the validity of our asymptotic result beyond Gaussian covariates $(\mathbf{u}, \mathbf{v})$ in the context of ridge regression, with linear teacher. As is well known, the computation of the training and test MSE for ridge regression can be written as a random matrix theory problem. We do not attempt a rigorous approach, but rather to motivate with simple arguments, many of them actually well known, the observed universality and its limits.

First, let us remind the definition of the model and introduce some simplifications that arise in ridge regression task. We have Gaussian covariates vectors $\mathbf{u} \in \mathbb{R}^p$ and $\mathbf{v} \in \mathbb{R}^d$, with correlations matrices $\Psi, \Omega$ and $\Phi$, from which we draw $n$ independent samples:

$$\begin{bmatrix} \mathbf{u} \\ \mathbf{v} \end{bmatrix} \in \mathbb{R}^{p+d} \sim \mathcal{N} \left( 0, \begin{bmatrix} \Psi & \Phi \\ \Phi^\top & \Omega \end{bmatrix} \right) . \tag{292}$$

We assume the existence of a linear teacher generating the labels $\mathbf{y} = U\boldsymbol{\theta}_0$, and recall the student performs ridge regression on the data matrix $\mathcal{V}$.

Note that since ridge regression can be performed in any basis, we might as well work in the basis where the population covariance $\Psi$ of the vector $\mathbf{u}$ is diagonal. Additionally, we shall use the fact that one can consider a $\boldsymbol{\theta}_0$ to be an i.i.d. Rademacher vector, i.e. a random vector of $\pm 1$ *without loss of generality*. Indeed, the statistical properties of the random variable $\mathbf{u} \cdot \boldsymbol{\theta}_0$, for a generic $\boldsymbol{\theta}_0$, and of the random variable $\tilde{\mathbf{u}} \cdot \boldsymbol{\theta}$, with $\boldsymbol{\theta}$ a Rademacher vector are identical provided a change in the (diagonal) covariance:

$$\Psi = \begin{pmatrix} \Psi_1(\boldsymbol{\theta}_0)_1{}^2 & 0 & \dots & 0 \\ 0 & \Psi_2(\boldsymbol{\theta}_0)_2{}^2 & \dots & 0 \\ \vdots & \vdots & \ddots & \vdots \\ 0 & 0 & \dots & \Psi_p(\boldsymbol{\theta}_0)_p{}^2 . \end{pmatrix} \tag{293}$$

The Gaussian model we consider can therefore be rewritten with a Rademacher vector $\boldsymbol{\theta}$ provided we change the correlation matrix $\Psi$ (as well as the cross-correlation $\Phi$) accordingly.

We now come back on the problem. Given the vector $\mathbf{y}$ and the data $\mathcal{V} \in \mathbb{R}^{n \times d}$, the ridge estimator has the following closed-form solution:

$$\hat{\mathbf{w}} = \left( \frac{1}{n} \mathcal{V}^\top \mathcal{V} + \lambda \mathrm{I}_d \right)^{-1} \mathcal{V}^\top \mathbf{y} = \left( S_{v,v} + \lambda \mathrm{I}_d \right) S_{u,v}^\top \boldsymbol{\theta} \tag{294}$$

where we have defined the *empirical* covariance matrices

$$S_{u,u} \equiv \frac{1}{n} U^\top U, \qquad S_{u,v} \equiv \frac{1}{n} U^\top \mathcal{V} \qquad S_{v,v} \equiv \frac{1}{n} \mathcal{V}^\top \mathcal{V}. \tag{295}$$

Given this vector, one can now readily write the expected value of the training and test losses as follows:

$$\begin{aligned} \mathcal{E}_{\text{train.}} &= \mathbb{E}_{U,\mathcal{V},\boldsymbol{\theta}} \left[ \frac{1}{n} \| U\boldsymbol{\theta} - \mathcal{V}\hat{\mathbf{w}}(U,\mathcal{V}) \|_2^2 \right] \\ &= \mathbb{E}_{U,\mathcal{V},\boldsymbol{\theta}} \left[ \frac{1}{n} \boldsymbol{\theta}^\top U^\top U\boldsymbol{\theta} \right] + \mathbb{E}_{U,\mathcal{V}} \left[ \frac{1}{n} \hat{\mathbf{w}}(U,\mathcal{V})^\top \mathcal{V}^\top \mathcal{V}\hat{\mathbf{w}}(U,\mathcal{V}) \right] - 2\mathbb{E}_{U,\mathcal{V}} \left[ \frac{1}{n} \boldsymbol{\theta}^\top U^\top \mathcal{V}\hat{\mathbf{w}}(U,\mathcal{V}) \right] \\ &= \mathbb{E} \left[ \mathrm{Tr}\, S_{u,u} \right] + \mathbb{E} \left[ \mathrm{Tr}\, S_{u,v} \left( S_{v,v} + \lambda \mathrm{I}_d \right)^{-1} S_{v,v} \left( S_{v,v} + \lambda \mathrm{I}_d \right)^{-1} S_{u,v}^\top \right] - 2\mathbb{E} \left[ \mathrm{Tr}\, S_{u,v} \left( S_{v,v} + \lambda \mathrm{I}_d \right)^{-1} S_{u,v}^\top \right] \end{aligned} \tag{296}$$

and

$$\begin{aligned} \mathcal{E}_{\text{gen.}} &= \mathbb{E}_{U,\mathcal{V},\mathbf{u},\mathbf{v},\theta} \left[ \frac{1}{n} \| \mathbf{u}^\top \boldsymbol{\theta} - \mathbf{v}^\top \hat{\mathbf{w}}(U,\mathcal{V}) \|_2^2 \right] \\ &= \mathbb{E}_{U,\mathcal{V},\mathbf{u},\mathbf{v},\theta} \left[ \frac{1}{n} \theta^\top \mathbf{u}\mathbf{u}^\top \boldsymbol{\theta} \right] + \mathbb{E}_{U,\mathcal{V}} \left[ \frac{1}{n} \hat{\mathbf{w}}(U,\mathcal{V})^\top \mathbf{v}\mathbf{v}^\top \hat{\mathbf{w}}(U,\mathcal{V}) \right] - 2\mathbb{E}_{U,\mathcal{V}} \left[ \frac{1}{n} \boldsymbol{\theta}^\top \mathbf{u}\mathbf{v}^\top \hat{\mathbf{w}}(U,\mathcal{V}) \right] \\ &= \mathrm{Tr}\, \Sigma_{u,u} + \mathbb{E} \left[ \mathrm{Tr}\, S_{u,v} \left( S_{v,v} + \lambda \mathrm{I}_d \right)^{-1} \Sigma_{v,v} \left( S_{v,v} + \lambda \mathrm{I}_d \right)^{-1} S_{u,v}^\top \right] - 2\mathbb{E} \left[ \mathrm{Tr}\, \Sigma_{u,v} \left( S_{v,v} + \lambda \mathrm{I}_d \right)^{-1} S_{u,v}^\top \right] \end{aligned} \tag{297}$$

where we have denoted the *population* correlation matrices $\Psi \equiv \Sigma_{u,u}, \Phi \equiv \Sigma_{u,v}, \Omega \equiv \Sigma_{v,v}$ for readability and a direct comparison with their empirical counterpart. The traces appears by the left and right multiplication by the random vector $\boldsymbol{\theta}$.

At this point, the entire problem has been mapped to a random matrix theory exercise: assuming data are indeed Gaussian, one can use RMT to compute the six traces that appears in (296,297). Indeed, this is the canonical approach used in most rigorous works for the ridge regression task in the teacher-student framework, instance in [12, 36–38]. Remarkably, the replica (and the rigorous Gordon counterpart) allow to find the same result without the explicit use of RMT.

We now discuss, heuristically, why these results are valid even though the distribution of $[\mathbf{u}, \mathbf{v}]$ is not actually Gaussian, and in some instances even for real data. Indeed, that both $(\mathcal{E}_{\text{train.}}, \mathcal{E}_{\text{gen.}})$ *do not depend* explicitly on the distribution of the data, but —assuming some concentration (or self-averaging)— only on:

1. The spectrum of the population covariances $\Sigma_{u,u}, \Sigma_{u,v}, \Sigma_{v,v}$.
2. The spectrum of the empirical covariances $\mathrm{S}_{u,u}, \mathrm{S}_{u,v}, \mathrm{S}_{v,v}$.
3. The expectation of the trace of products between empirical and population covariances.

We expect that asymptotically the prediction from the theory will thus be valid for much more generic distributions $[\mathbf{u}, \mathbf{v}] \sim P_{\mathbf{u}, \mathbf{v}}$, provided they share the same population covariances $\Sigma_{u,u}, \Sigma_{u,v}, \Sigma_{v,v}$ (which we call $\Psi, \Phi, \Omega$). To see this, we need to check how this change in distribution would affect points (1),(2) and (3). Fixing the population covariances, the first bullet point (1) is automatically taken into account. Point (2) and (3) are, however, less trivial: in order to have universality we need that a) the spectrum of the *empirical* covariances of the non-Gaussian distribution to converge the one obtained with the Gaussian one; and b) the trace of products between the *empirical* and the *population* covariances also to converge to the universal values computed from Gaussians data.

These two last points have been investigated in RMT [77], and it is a classical result that such quantities are universal and converge to the Gaussian-predicted values for many distribution, way beyond the Gaussian assumption (in which case the spectral densities are known as the Wigner and Wishart model, or Marcenko-Pastur distribution [78]): this powerful universality of RMT is at the origin of the applicability of the model beyond Gaussian data. For instance, [64] showed that these assumptions are verified for any data generated as $\mathbf{u} = \boldsymbol{\Sigma}_{\mathbf{u},\mathbf{u}}^{1/2}\boldsymbol{\omega}$, assuming the components of the vector $\boldsymbol{\omega}$ are drawn i.i.d. from *any* distribution (with some assumption on the larger moments). While this is still restrictive, stronger results can be shown, and [49, 63, 65, 79] extended them (also loosening the independence assumption) for a very generic class of distributions of correlated random vectors $\mathbf{u}$.

Let us give a concrete example. For simplicity, consider the restricted case where $\mathbf{u} = \mathbf{v}$, i.e. the teacher acts on the same space as the student. In this case, eqs. (296,297) simplify (this is essentially the analysis in [36]) to:

$$
\begin{aligned}
\mathcal{E}_{\text{train.}} &= \mathbb{E}_{\mathrm{U},\boldsymbol{\theta}}\left[\frac{1}{n}\|\mathrm{U}\boldsymbol{\theta} - \mathrm{U}\hat{\mathbf{w}}(\mathrm{U})\|_2^2\right] \\
&= \mathbb{E}_{\mathrm{U},\boldsymbol{\theta}}\left[\frac{1}{n}\boldsymbol{\theta}^\top \mathrm{U}^\top \mathrm{U}\boldsymbol{\theta}\right] + \mathbb{E}_{\mathrm{U}}\left[\frac{1}{n}\hat{\mathbf{w}}(\mathrm{U})^\top \mathrm{U}^\top \mathrm{U}\hat{\mathbf{w}}(U)\right] - 2\mathbb{E}_{\mathrm{U}}\left[\frac{1}{n}\boldsymbol{\theta}^\top \mathrm{U}^\top \mathrm{U}\hat{\mathbf{w}}(\mathrm{U})\right] \\
&= \mathbb{E}\left[\text{Tr } S\right] + \mathbb{E}\left[\text{Tr } S\left(\mathrm{S} + \lambda \mathrm{I}_d\right)^{-1} S\left(\mathrm{S} + \lambda \mathrm{I}_d\right)^{-1}\mathrm{S}^\top\right] - 2\mathbb{E}\left[\text{Tr } S\left(\mathrm{S} + \lambda \mathrm{I}_d\right)^{-1}\mathrm{S}^\top\right]
\end{aligned}
\tag{298}
$$

and

$$
\begin{aligned}
\mathcal{E}_{\text{gen.}} &= \mathbb{E}_{U,\mathbf{u},\theta}\left[\frac{1}{n}\|\mathbf{u}^\top \boldsymbol{\theta} - \mathbf{u}^\top \hat{\mathbf{w}}(\mathrm{U})\|_2^2\right] \\
&= \mathbb{E}_{\mathrm{U},\mathbf{u},\theta}\left[\frac{1}{n}\boldsymbol{\theta}^\top \mathbf{u}\mathbf{u}^\top \boldsymbol{\theta}\right] + \mathbb{E}_{\mathrm{U}}\left[\frac{1}{n}\hat{\mathbf{w}}(\mathrm{U})^\top \mathbf{u}\mathbf{u}^\top \hat{\mathbf{w}}(U)\right] - 2\mathbb{E}_{\mathrm{U}}\left[\frac{1}{n}\boldsymbol{\theta}^\top \mathbf{u}\mathbf{u}^\top \hat{\mathbf{w}}(U)\right] \\
&= \text{Tr } \Sigma_{u,u} + \mathbb{E}\left[\text{Tr } S\left(\mathrm{S} + \lambda \mathrm{I}_d\right)^{-1}\Sigma\left(\mathrm{S} + \lambda \mathrm{I}_d\right)^{-1}\mathrm{S}^\top\right] - 2\mathbb{E}\left[\text{Tr } \Sigma\left(\mathrm{S} + \lambda \mathrm{I}_d\right)^{-1}\mathrm{S}^\top\right] \quad (299)
\end{aligned}
$$

In the expression of the training loss eq. (298), we see terms such as

$$
\mathcal{A} = \mathbb{E}\left[\text{Tr } S\left(\mathrm{S} + \lambda \mathrm{I}_d\right)^{-1}\mathrm{S}^\top\right],
\tag{300}
$$

depend only on the limiting distribution of eigenvalues of $S \in \mathbb{R}^{d \times d}$. This is a very well known problem when the dimension $d$ and the number of samples $n$ are send to infinity with fixed ratio $\alpha = n/d$, and the limiting spectral density is known as the Marcenko-Pastur law. This is a very robust distribution that is valid way beyond the Gaussian hypothesis [49, 63, 65, 79].

In the expression of the generalisation loss eq. (299), however, terms such as

$$\mathcal{B} = \mathbb{E}\left[\operatorname{Tr} \Sigma \left(S + \lambda I_d\right)^{-1} S^\top\right] . \tag{301}$$

appears. These can be computed using classical RMT results on the concentration of the inverse of the covariance [64, 80]. The strongest result we are aware of for such problems is from the remarkable work of [49]. This universality of random matrix theory is thus at the origin of the surprisingly successful application of our Gaussian theory to real data with arbitrary feature maps. Of course the discussion here is limited to the case where $\mathbf{u} = \mathbf{v}$ and a concrete mathematical statement would require the generalisation of these arguments to the more generic case of eqs.(296,297), which are closer to the work of [37]. We leave this discussion to future works.

A similar universality has been discussed for kernel methods in very recent works, but for the slightly different setting in which data is drawn from a Mixture of Gaussians [38, 50] (in which case there is no teacher, the label depends on which Gaussian has been chosen). The universality observed here for ridge regression with linear student, albeit different, is of a similar nature, and it would be interesting to discuss the link between these two approaches.