# OpenReview forum: "Learning curves of generic features maps for realistic datasets with a teacher-student model"
_NeurIPS.cc/2021/Conference — NeurIPS 2021 Poster_

### Official Review · Reviewer_mfPa · 2021-07-10

**Rating:** 6
**Confidence:** 3

**Summary:**

This paper investigates the performance of a class of teacher-student models in the context of supervised learning, and theoretically derives learning curves in the high-dimensional limit. An important contribution of this paper is to have unified a number of related previous works by introducing a new model called Gaussian covariate model. In this model, two covariates, u and v, are assumed to be generated from a common input data in different ways; u and v are given to the teacher and student, respectively, and are assumed to be mutually correlated Gaussian characterized by generic (though some regularity is assumed) covariance. The introduction of the conversion step from the input to the covariates makes it possible to handle more realistic datasets in the teacher-student framework, while keeping the model analytically tractable by the Gaussian assumption of the covariates. Numerical experiments well support the theoretical prediction. Especially, the experiment using GAN, in which the GAN generated data exhibits a good consistency with the theory while the original one used for training the GAN shows an inconsistency, is interesting since it simultaneously shows the possibilities and limitations of the proposed theory.


**Limitations And Societal Impact:**

Not relevant to this paper.

**Main Review:**

Originality: The above idea introducing the conversion from the input to the covariates is new and interesting. Methods mainly used in the analysis are the replica method and the Gordon's theorem; they are not new.

Quality: The results are well supported by the consistency between the analysis and the numerical experiment. The theoretical computations are also described and I do not find any serious error in it. The quality is high in this sense.

Clarity: I think the presentation is not clear: there are too many claims in the manuscript, which makes it hard to see what the important points are. In my opinion, the crucial contribution of this paper is the introduction of the Gaussian covariate model which is flexible and opens a possibility to treat modern realistic datasets in the teacher-student scenario; thus, it is better to mainly focus on this point. In this context, I do not understand why two computations by the replica method and by the Gordon's theorem are both necessary in the present manuscript. The position of the non-asymptotic case results is also unclear.
Also, I found small but many typos, unexplained details of the model/problem setting, misleading expressions, undeclared abbreviations, and redundant repetitions, especially in the Supplementary Material. (I will give a very brief list of these things below.) These make the manuscript hard to read. Overall, I think the level of clarity is low.

Significance: I think the analysis based on the above idea potentially has a large significance, since it provides a possibility to make the theory closer to the practice in real data analysis. Unfortunately, this point is not fully pursued in the present manuscript. Overall, I think the significance level is medium.

The list of typos, misleading expressions, etc.:

%%

Line 44: r and g are assumed to be convex and thus they are continuous, implying that ``lower-semicontinuous'' assumption is unnecessary.

Lines 64,66: the mathematical characters become non-italic here.

Lines 190: I think id is the abbreviation of identity, but it is better to explain at the first appearance.

Eq. (27): It is better to define Z.

A sentence below Eq. (29):
"which the deterministic case in which this is fixed given"
Is this grammatically correct?

A sentence below Eq. (38):
Inserting this ansatz in eq. (34) allow us
->
Inserting this ansatz in eq. (34) allows us
(Similar error of missing the third-person singular -s are found many times)

etc.

%%

Some other suggestions and comments:

%%

Section 3: More explanation is desired for each concrete setting.

Theorem 3: Universality of the linear model should be explained more deeply somewhere, not only by stating the theorem.
About regularization: In this manuscript, mainly the $\ell_2$ regularization is considered. What happens if the $\ell_1$ is used? I think the $\ell_1$ regularization breaks the rotational symmetry and thus the computation does not proceed well in this case. If so, it is better to state this as one of limitations of the theory.
%%


**Time Spent Reviewing:**

12 hours

---

> ### Author Response · Authors · 2021-08-09
> **Reply to mfPa**
>
> We thank the reviewer for pointing out a list of items, which will help us to improve and clarify the revised version.
>
> We completely agree with the characterisation of the reviewer: the main point of our paper is the introduction of the Gaussian covariate model, since it opens up a way to analyze the performance of simple neural networks on realistic data within the teacher-student setup.
>
> Our rigorous results are indeed obtained using Gordon's theorem. The referee is right that this would be enough to have a self-contained paper. However, we have decided to add the corresponding replica results in the interest of those members of the NeurIPS committee whose background is in statistical physics. Furthermore, there is a series of works exploring the extent to which the non-rigorous predictions of statistical physics can be made rigorous using different tools, such as Gordon's inequalities. The equivalence between replica and Gordon is thus an interesting result in itself, adding to a line of research that is of interest to a subset of the NeurIPS community.
>
> Note that establishing new replica and Gordon results to a new model are important and non-trivial contributions even though the methods are not new. There are many papers applying these same methods to new problems, and in particular these works are published every year at venues such as NeurIPS & ICML.
>
> Below we address some specific points raised:
>
> 1. The non-asymptotic results of Theorem 2 show how the closed-form asymptotics of Theorem 1 can be made more precise by quantifying how fast the different quantities concentrate towards the asymptotic representation with the dimension of the problem. It is also there to support the fact that, in the figures, dimensions of only a few hundreds are required to obtain clear curves. The most generic form of the theorem could be stated directly using these rates, but in our opinion this formulation hides the closed-form results that include the effect of the covariance matrices (through their eigenvalue distribution) in the $l_2$ case, presented in Theorem 1, and which are easier to interpret. We thus decided to present both formulations.
>
>
> 2. *Comment on $\ell_1$ regularisation:* The most general form of our theorem (Theorem 2 in the main body, Theorem 5 of the appendix) includes any non-zero convex regularization, and in particular the $\ell_1$ penalty. Theorem 2 gives the asymptotic model describing the estimator $w$ in the form of the quantity $w^{\star}$ at lines 165-166. The equation defining $w^{\star}$ involves the proximal of the regularization composed with the covariance $\Omega$, and depends on the vector $\mathbf{t} = \Omega^{-1/2}\Phi^{\top}\theta_{0}$. This shows that for $\ell_1$-regularization, a soft-thresholding will be composed with the covariance matrices and indeed break rotational symmetry. We agree that further investigation of sparse problems in this setting is an interesting future research direction.
>
>
> 3. *Comment on significance:* We agree that the model provides a possibility to make the theory closer to practice in real data analysis, and we give a clear evidence of that in the result section. We also agree that there are potentially many interesting directions in which this can be pursued. On the other hand, as pointed out by the referee the paper is already dense with results, and we fear that adding more would hurt clarity further. We welcome the referee's suggestions to organize better the current results and to improve clarity of exposition.
>
> 4. *Comment on lower-semicontinuity:* We note that lower-semicontinuity is a generic assumption in convex analysis, see e.g. Chapter 9 "Lower-semicontinuous convex functions" of [28]. However we agree that instances of convex functions that are not continuous lead to an ill-defined problem with little use in machine learning.

---

> > ### Comment · Reviewer_mfPa · 2021-08-23
> > **Thank you for your rebuttal**
> >
> > The authors have indeed responded to all my concern, but it seems that the current presentation style would not be largely changed to improve the clarity which is my largest concern. Thus, I have no good reason to raise my score and would like to keep the original one which is in the positive side.
> >
> > The below is my response to some of the authors comments:
> >
> > > Our rigorous results are indeed obtained using Gordon's theorem. The referee is right that this would be enough to have a self-contained paper. However, we have decided to add the corresponding replica results in the interest of those members of the NeurIPS committee whose background is in statistical physics. Furthermore, there is a series of works exploring the extent to which the non-rigorous predictions of statistical physics can be made rigorous using different tools, such as Gordon's inequalities. The equivalence between replica and Gordon is thus an interesting result in itself, adding to a line of research that is of interest to a subset of the NeurIPS community.
> > > Note that establishing new replica and Gordon results to a new model are important and non-trivial contributions even though the methods are not new. There are many papers applying these same methods to new problems, and in particular these works are published every year at venues such as NeurIPS & ICML.
> >
> > I know that inventing the proof techniques and inventing the computational techniques are different, and both are important. I understood the present manuscript emphasizes these points.
> >
> >
> > > The non-asymptotic results of Theorem 2 show how the closed-form asymptotics of Theorem 1 can be made more precise by quantifying how fast the different quantities concentrate towards the asymptotic representation with the dimension of the problem. It is also there to support the fact that, in the figures, dimensions of only a few hundreds are required to obtain clear curves. The most generic form of the theorem could be stated directly using these rates, but in our opinion this formulation hides the closed-form results that include the effect of the covariance matrices (through their eigenvalue distribution) in the  case, presented in Theorem 1, and which are easier to interpret. We thus decided to present both formulations.
> >
> > Thank you for the explanation. I understand the importance of this Theorem.
> >
> > > Comment on $\ell_1$ regularisation:
> >
> > I understand that Theorem 2 still holds even for the case of the $\ell_1$ regularization. My point was that the simple closed-form result like Theorem 1 cannot be obtained for the case of the $\ell_1$ regularization since the integration w.r.t $w$ in Eq. (43) in Appendix has no compact analytical formula. I found this point is stated when re-reading it, but it is not clear.
> >
> > > Comment on significance:
> >
> > I think we agree on the interesting point of the present paper. I think the presentation will be clearer and the insight will be more direct if only the closed-form result is shown and its consequence is intensively studied. But now I understand the aim of the present paper is to establish the replica and Gordon results in a new model, and hence there is nothing more I can say.
> >
> > > Comment on lower-semicontinuity:
> >
> > Thank you for your comment. Concerning to this point, there was a lack of understanding on me, and now I fully agree with the assumptions on the theorems.

---

> > > ### Author Response · Authors · 2021-08-23
> > > **Addressing your concerns on clarity**
> > >
> > > Thank you for your reply to our rebuttal. We are happy that we could clarify most of the points, and that you think the results have *“potentially large significance”*.
> > >
> > > We do hear that your biggest concern remains clarity. While you say *“it seems that the current presentation style would not be largely changed to improve the clarity”* we want to stress that we certainly appreciated your (and other referees) specific pointers about which parts and reasoning were not clear, and we plan to spend time on the camera ready version to implement those very carefully. You and the other referees have spent time giving us very useful feedback and we can assure you all that we will not waste that time and will improve the presentation of our paper accordingly.
> > >
> > > We are aware that if the paper is accepted this discussion will be made public and we would not be making it if we were not to stand by it.

---

### Official Review · Reviewer_Yv7K · 2021-07-16

**Rating:** 6
**Confidence:** 2

**Summary:**

In this papers the author proposed a new Gaussian covariate teacher-student model that is capable of accurately estimating the training and generalization errors for its learned linear coefficients of its student part with an exponentially decaying concentrate bound. Their framework covers a wide range of learning problems that either are either directly in teacher-student style or can be easily adapted into such setup. Their empirical study demonstrated the effectiveness of their framework for a variety of problems even when letting loose some of the theoretical assumptions required.



**Limitations And Societal Impact:**

Yes

**Main Review:**

The authors deliver a strong theoretical study on their new model of framing a wide range of learning problems into a Gaussian covariate teacher and student model. The guaranteed training and generalization errors and the versatility of their framework are all promising results, and the paper is relatively sound in its theoretical development. That said, the presentation of the results could be clarified.

The reviewer has a few major concerns regarding the theoretical results, as follows:

1. Regarding Theorem 1, how closed-form it is seems to depend on the optimization problem of prox_{V, g} and the iterative steps to find the fixed points of the parameters for the projected 2d Gaussian distribution. The former part would not be closed form should the loss function do not have closed form solution. The latter part seems unlikely to finish exactly in finite steps. It would also be nice to have more discussions on prox_{V, g} - it is quite strong in the theorem in the sense that it technically solves the problem in the projected space therefore determines the training error term.

2. Regarding Theorem 3, the fact that the result is independent of all perfect linear teachers somehow suggests that the teacher part of the framework can be omitted in practice since all its influence is through the covariance structure after the feature map, while this structure, as shown in the theorem, can be estimated directly from the label and the student's feature map.

The reviewer also thinks the deliver of the paper, especially the theoretical development, could be more clarified and use more explanations. There are typos that prevent a smooth pass through the theory in its current shape. To name a few:
- Line 120, \hat{\mu}_d is defined but never used.
- \mu is double used as both a sample index and a limiting distribution
- Before Theorem 1, line 129, C_2 and C_3 are mentioned but never used.
- Theorem 1, z is double used (consider directly define z as prox).
- Theorem 2, Assumption (B.1) is not numbered in the context.
- Line 167, (V*, .. \hat{m}*) could have their definition in the main body.

====
Post rebuttal:

The reviewer would like to thank the authors for addressing most of the concerns, and would like to raise the rating to a 6.

**Time Spent Reviewing:**

6

---

> ### Author Response · Authors · 2021-08-09
> **Reply to Reviewer Yv7K**
>
>
> We thank the reviewer for his/her detailed comments, and welcome the suggestions of the reviewer for presenting the results more clearly. The comments are addressed point-by-point below.
>
> 1. *Regarding Theorem 1, how closed-form it is seems to depend on the optimization problem of $\mathrm{prox}_{Vg}$ and the iterative steps to find the fixed points of the parameters for the projected 2d Gaussian distribution.*
>
> The referee is right in pointing out that our expressions for the asymptotic errors in Theorem 1 are given in terms of an optimisation problem. However, in contrast to the original empirical risk minimisation problem (which scales with the dimension of the problem, of the feature space, and sample size), our result is written in terms of a few real variables. Therefore, while the original problem is high-dimensional, our formula for the limit $n,p,d\to \infty$ is closed-form in the sense that the low-dimensional optimisation problem is readily solvable in a computer. In fact, the theoretical predictions shown by solid lines in all the figures of the main text were obtained by solving these low-dimensional optimisation problems. This type of result is common to many theoretical works in high-dimensional statistics and theory of machine learning, e.g. our references [4, 5, 11, 14, 24, 27, 31, 32, 35, 36, 42, 43, 47, 48] This kind of characterization is also common in information and coding theory where it is referred to as "single letter" characterization, as well as in statistical physics where the description of a high-dimensional limit in terms of few variables is referred to as "exact solvability" of a model. We welcome the comment and will be stressing this in the revised version.
>
> 2. *It would also be nice to have more discussions on $\mathrm{prox}_{Vg}$*
>
> Proximal operators (of convex functions) are well-defined objects with a substantial literature, both theoretical (e.g. [28]) and practical (e.g. [PB14] below). They are akin to a gradient step (with step-size the parameter of the proximal, here V) on a smoothed version of the initial function (with the same minimizers, the Moreau envelopes, see e.g. [28,PB14]). They always appear explicitly in asymptotic characterization of GLMs (see references above) of the flavour of the one presented here.  They are easy to evaluate (the optimization problem defining them is strongly convex, and therefore the solution exists and is unique), one-to-one mappings, almost contractive (firmly non-expansive) and their fixed points coincide with the minimizers of the original function. They can, as pointed out by the referee, be interpreted as a projection on the level lines of the original function.
>
> In the present case, we show that both the estimator $\hat{w}$ and the output of the linear model estimator*data $\hat{z}$ are Gaussian random variables centered around two concrete quantities linked to the teacher vector $\theta_{0}$, deformed by a function that depends on the loss (for $\hat{z}$) and regularization (for $\hat{w}$). The functions gauging this deformation are precisely the proximal operators. More specifically, the proximal of the loss parametrizes the degree of non-linearity in the loss function. This interpretation is in line with robust regression, where non-square losses are introduced to account for the non-linearity of the generative model. Concerning the proximal for the regularization, in the $\ell_2$ case it is a linear function, giving Theorem 1. Another instructive example is the $\ell_1$ case, for which the proximal operator corresponds to the soft-thresholding function: this means $\hat{w}$ becomes (in distribution) a thresholded Gaussian vector centered around the teacher vector, hence the sparsity of the solution. This is summarized by the equations defining $w^{\star},z^{\star}$ at line 165-166.
>
> 3. Theorem 3
>
> What the referee says is correct, but only for the specific cases for which Theorem 3 applies, i.e. the Gaussian covariate model that uses a loss with $l_2$ regularization and the special case of linear interpolating teachers. (we will make sure to make this is clear in the revised manuscript). By invoking Theorem 3 we are thus implicitly assuming that the input is Gaussian, the penalty is $l_2$ and there exists a linear interpolating teacher. The independence of the teacher does not hold for non-linear models, and the teacher part of the framework cannot be omitted. Theorems 1 and 2 hold for a broader range of teachers, and the appendix gives the analogous theorems for other penalties, including for instance lasso and elastic nets.
>
>
> [PB14] *N. Parikh and S. Boyd, Proximal Algorithms, 2014.*
>
> **Minor comments of the referee**
>
> 1. We will reference assumptions (A1-A7) instead of the section (B.1) in the revised version.
>
> 2. "Line 120, $\hat{\mu}_d$ is defined but never used."  the asymptotic limit $\mu$ of the joint distribution eq. (7) is used in Theorem 1. We will make this move visible.
>
> 3. The constants $C_2,C_3$ are indeed not used, we will correct this and other notational inconsistencies and suggestions pointed out by the reviewer.

---

> > ### Author Response · Authors · 2021-08-30
> > **Thank you for the feedback**
> >
> > We thank again for this review that gives us valuable feedback on our paper.
> >
> > As the discussion period comes to an end, we would like to make sure our answer provided clarification on the questions raised by the reviewer.
> >
> > Sincerely yours,
> >
> > The authors.

---

### Official Review · Reviewer_Dp3D · 2021-07-17

**Rating:** 6
**Confidence:** 3

**Summary:**

This paper introduces a Gaussian covariate teacher-student model, allowing for the teacher and student to work on different spaces introduced by different feature maps. The authors demonstrate close-form expressions for the training and generalization error in an asymptotic setting. These predicted results are compared with actual performances for kernel methods with ridge regression. The predicted quantities are shown to fit well to observed learning curves, even in the ridge regression case with real data. However, it is also observed that the prediction is limited in general for real datasets.

**Ethical Concerns:**

No ethical concern

**Limitations And Societal Impact:**

The paper has no direct societal impact.

**Main Review:**

Originality: The paper presents an interesting step forward compared to the current literature based on teacher-student models. The proposed extension makes it possible in particular to have different working spaces for the teacher and the student, which is an important characteristic. In particular, this echoes the paradigm of learning with privileged information, which is not discussed in the paper but could be an interesting application case. Otherwise, the paper is will located in the state of the art and the authors are very explicit about related works, and their contributions.


Quality:  The paper seems of a very high technical quality, with some reservation which I will discuss later. I particularly appreciated that the main paper focuses on a the explanation and intuition aspect of the work, proposing extremely pertinent discussions on the presented results. Unlike in many other papers, here the authors have a rigorous approach, presenting their observations and analyses of these observations (a typical example is the final discussion on why ridge regression predictions work so well even on real datasets which are not Gaussian). Based on the main paper only, my opinion is that the paper is a sound and rigorous research.
My main concern however is that the results presented in the paper all depend on proofs which are all detailed in appendix. I could not complete a thorough review of the proofs in the reviewing time. What I read and reviewed seems correct, but I do not think it is possible to accept the paper with a doubt on the technical validity of the presented results. Actually, the problem here is that the technical content contained in the appendix is at least as important as the content of the main paper, and not additional information. For this reason, I tend to think that the paper would be more adapted to a journal. I will continue reviewing the proof until the end of the reviewing process and hope that some other reviewers could have checked the proofs already. If I can confirm the validity of the appendix (and depending on errors that other reviewers may have noticed), then I will change my score to a 8.

Minor comments and typos:
- l.39: mention that mu corresponds to the index of a data point
- l.44: “and” rather than “&” between mathematical symbols
- Figures: mention the number of runs used to compute the standard deviation
- Equation 14: Is there an intuitive interpretation of V*?
- l.171: Assumption (B1) is presented in the appendix only. Please introduce it in the paper.
- l.241: that follows => that follow
- l.316-319: Explanations unclear. Could you please rephrase it?
- l. 339: two first => first two

A very open question: Do you think a general teacher able to reproduce learning curves for more general cases could exist? I can imagine here that some no-free-lunch result could be proven, showing that no such general teachers could be found. What is your opinion on this question?

Clarity: The paper is extremely well-written and clear to read.

Significance: The proposed work tackles important problems and is an important theoretical contribution.


**Time Spent Reviewing:**

7

---

> ### Author Response · Authors · 2021-08-09
> **Reply to Reviewer Dp3D**
>
> We thank the reviewer for his/her detailed comments. We are particularly grateful for your critical reading of the proofs, and are happy to help with this process by answering questions that may arise during the discussion period. We give below pointers to what we think are the key parts of the proof to ease your review. We also thank the referee for the list of minor comments and typos that help us polish our manuscript, and on which we comment below.
>
> **Comments on the proof:** The core part of the proof is finding the decoupled problem ("auxiliary problem") corresponding to the initial optimization problem ("primary problem"), along with the study of its asymptotic properties and the existence and uniqueness of its solution. This part follows the typical sketch of proof of Gordon-type methods (e.g. [27]), but each step considers a more complex problem than the ones appearing in the current literature. The most important (and new) parts are section B.4 ("Determining a candidate ...") and B.5 ("Study of the ..."). Once the asymptotics and the uniqueness of the solution of the decoupled auxiliary problem are proved, the main theorem is shown using the methods of Refs. [51], [29]. As previously mentioned, we are happy to discuss further details during the discussion period. Please do not hesitate to write us with questions.
>
> **Other comments**
>
> 1. *Is there an intuitive interpretation of $V^{\star}$?*
>
>  $V^{\star}$ parametrizes the deformation that must be applied to a Gaussian field specified by the solution of the fixed point equations to obtain the asymptotic behaviour of $\hat{z}$. This interpretation for $V^{\star}$ (and also $\hat{V}^{\star}$) is better understood through Theorem 2 and the equations for $w^{\star},z^{\star}$ at line 165-166. The theorem shows that $\hat{w},\hat{z}$ (defined line 163) asymptotically behave as deformations of Gaussian fields centered around quantities linked to the teacher vector. These deformations are given by the proximal operators of the loss and regularization. Proximal operators are parametrized by a positive constant - which here are precisely $V^{\star},\hat{V}^{\star}$. This has a clear interpretation for the regularization : in the $\ell_2$ case, the proximal gives a linear correction with slope that depends on $\hat{V}^{\star}$, in the $\ell_1$ case it gives a soft-thresholding function, with the threshold value set by $\hat{V}^{\star}$. $V^{\star}$ plays the same role for the loss function and the linear output of the learned model $\hat{z}$. It prescribes the degree of non-linearity given to this output by the loss. This is coherent with a robust regression viewpoint, where one introduces non-square losses to deal with the potential non-linearity of the generative model. In specific cases,
>
> For specific cases these parameters can have an additional interpretation. For instance, for the square loss $V^{\star}$ can be seen as parametrizing the generalization gap, see discussion in Appendix A.5, p19,line 672-691 below eq. (59).
>
>
> 2. *Clarity of lines 316-319: "This is to be expected since in that case the teacher can be learned with zero test error, while the test error on the real data set saturates"*
>
> The deviation that we see between the learning curve on real data vs the learning curve obtained from the teacher-student model comes from the fact that the student can, in principle, express the same function as the teacher if it recovers its weights exactly. Recovering the teacher weights becomes possible with a large training set. In that case, its test error will be zero. However, in our setup the test error on real data remains finite even as more training data is added, leading to the discrepancy between teacher-student learning curve and real data. We will clarify this in the revised manuscript; we also discuss this point in more detail together with Fig. 6 in the appendix.
>
> 3. *"[Is there] a general teacher able to reproduce learning curves for more general cases [...]?"*
>
> This is a question we kept thinking about after submitting our work. We found that for classification if we consider a Gaussian mixture where every class is represented by one Gaussian with generic mean and covariance we can get a much better agreement with the real learning curves than with the present Gaussian covariate model. It is still not perfect, but much closer and improved if a larger number of Gaussians is considered. We also managed to analyse the corresponding Gaussian mixture model. We are happy to discuss further details with the reviewer.
>
>
> 4. *This [work] echoes the paradigm of learning with privileged information, which is not discussed in the paper but could be an interesting application case.*
>
> Thank you for pointing out the connection to learning with privileged information, we are not aware of this literature but will certainly study it.

---

> > ### Author Response · Authors · 2021-08-30
> > **Thank you for the feedback**
> >
> > We thank again for this review that gives us valuable feedback on our paper.
> >
> > As the discussion period comes to an end, we would like to make sure our answer provided clarification on some of the questions raised by the reviewer, and in particular the proof, which was her/his main point of concern. Indeed, we got tempted by the mentioned possibility of raising the score to 8 and are eager to answer questions in that respect.
> >
> > Sincerely yours,
> >
> > The authors.

---

### Official Review · Reviewer_azvB · 2021-07-20

**Rating:** 6
**Confidence:** 3

**Summary:**

The paper introduces a teacher student method that does not require the input data to follow the i.i.d. assumption. Authors provide a closed form expression for the asymptotic training and generalization error and provide strengths and limitations of the proposed method under empirical settings.
The paper covers the related work and the theoretical arguments are sound. The method is tested on synthetic and real world data.

**Limitations And Societal Impact:**

Societal impact from this work is very comparable to any other machine learning model that my suffer from bias or skewed training data and not necessarily related to the proposed method.

**Main Review:**

Originality: The paper builds on existing teacher student setting and take it to the next level by dropping the requirement of iid data which makes the method applicable to real data.

Quality: The theoretical discussions in the paper are solid and experimental results back the claims. Authors have tested the method under different synthetic and real data settings.

Clarity: My main concern with this submission is clarity. The reader needs to go back and forth between the main body and the appendix to follow the theoretical discussions which is time consuming. The flow of the paper is not well organized in parts and theoretical discussions and experimental results are mixed. Authors could have double clicked on some interesting observations in the experimental results (e.g. Fig 4 vs. Fig 3 and the effect of the number of epochs on the classification errors across data sets with different characteristics).

Significance: The contributions are fairly significant. The authors could have shed more light into the trade offs between the computational overhead of the proposed approach versus the classification error improvement.


Minor:

line 40: such as, for instance => such as

line 120: The use case for joint empirical density is not clear.

line 154: "the one of a", consider rewording.

contributions are scattered all over introduction and can be unified.



**Time Spent Reviewing:**

4

---

> ### Author Response · Authors · 2021-08-09
> **Reply to Reviewer azvB**
>
> If we understood correctly, the referee raised three main points:
>
> 1. *Point 1: Clarity.*
>
> As pointed out by another reviewer, our work is dense, and combines a set of rigorous theoretical results and different experimental applications. We agree with the referee that this poses a challenge in terms of the presentation of the results within the space constraints. The paper is roughly divided in theoretical and experimental results, although we have decided to include one result early on the introduction with the purpose of captivating the reader who might only be interested in the applications. In the theory part, we have opted to leave part of the general result in the appendix, choosing to present in the main manuscript two self-contained theorems which we believe are the most relevant to the experiments that follow. On the same line, in the experiments section we have decided to focus in presenting a diverse set of scenarios where the theory might be useful, rather than focusing in the phenomenology of specific scenarios. We understand and agree with the referee that these choices might not fulfil a reader who wants to go deeper either in the rigorous math or in the observed phenomenology - but we do not see how to fulfill such reader within the page limit for the main paper. In the revised version we welcome the referee's suggestions to improve the clarity of exposition, we will concentrate on that in the revised version.
>
> 2. *"The authors could have shed more light into the trade offs between the computational overhead of the proposed approach versus the classification error improvement."*.
>
> We agree with the referee that we have not commented on the computational trade-offs of evaluating our theory (i.e. solving the equations (13)) and actually running the finite-size instances of the problem. The great advantage of the asymptotic formula (13) is that it is written only in terms of scalar quantities, and requires at most performing double numerical integrals. The sample complexity $\alpha$ (\# of samples / \# of parameters) and the aspect ratio $\gamma$ (\# of teacher parameters / \# student parameters) are simply constants in the formula. Therefore, different from using standard solver on a finite-size instance, the computational time in solving our formula doesn't scale with the dimensions.  This allow us to access interesting regimes such as \# parameters $\gg$ \# samples (overparametrisation) in a laptop effortlessly. We will add this discussion in the revised manuscript.
>
>
> Our goal was not to improve classification error, but analyze the performance of existing estimators. There is thus no trade-off between the computational overhead and error improvement relevant to our work.
>
>  3. *"The use case for joint empirical density is not clear."*
>
> Formally, the definition of the joint empirical density eq.~(7) is required in order to properly define its asymptotic limit (line 125), which appears in the left of the self-consistent equations (13) for the errors. In practice, there are two scenarios: 1) if possible, compute the asymptotic spectral density analytically, and therefore perform the averages in equation (13) explicitly; 2) if not possible, approximate the integral by a sum over a draw of the spectrum, which is equivalent to using directly the empirical density. In the later, this involves a pre-processing step which, due to concentration of the quantities involved, is done once for all before iterating the self-consistent equations. Both cases are exemplified in the public GitHub repository which will be released with this work.

---

### Decision · Program_Chairs · 2021-09-27

**Decision:**

Accept (Poster)

**Comment:**

This paper received four reviews in total, with the scores/confidences (after the author response) being 6/3, 6/3, 6/2, and 6/3. Upon reading by myself the reviews and the author response, as well as the paper itself, my judgement on this paper is that, whereas it has a clarity issue as pointed out by most reviewers, it presents an interesting piece of theoretical work. Overall, the reviewers evaluated quite positively this paper's contributions such as:
- Crystallizing those problem settings in some recent pieces of work into the formulation which the authors call the novel Gaussian covariance teacher-student model.
- Providing a rigorous characterization of the properties of the proposed model (summarized in Theorems 1 ($d\to\infty$) and 2 (non-asymptotic)) under fairly generic settings (although the reviewers would not have checked all the details of the proofs in SM).
- Demonstrating applicability of the theoretical predictions to more realistic scenarios. The problem setting in the proposed model would seem rather simplistic as both the teacher and the student are essentially assumed to be single-layer perceptrons. Nevertheless, the authors argue via what is called the Gaussian equivalence conjecture (GEC; Conjecture 1) that the analytical results would be able to capture asymptotic behaviors of a wider class of models beyond the proposed model. The GEC was empirically confirmed on the teacher-student setting in this paper via the experiments on trained generative networks (Section 3.3), where one focuses on the last layers of the trained teacher and student networks which allows us to cast the problem into the analytical framework of this paper except the Gaussianity, as well as on ridge regression (Section 3.4), where not the Gaussianity but second-order statistics determine the learning behaviors. The empirical validation of the GEC on the teacher-student setting is another main contribution of this paper.

On the basis of the positive scores as well as the above evaluations, I am happy to recommend acceptance of this paper.

I agree with the authors that some comments on computational aspects as non-essential. As for the clarity issue, I also agree with the authors that this paper contains a lot of theoretical and experimental results so that it may be somehow unavoidable for such a paper to be dense. Still, my impression is that there exist some gaps between the arguments given in natural language and those given in the form of mathematical expressions. As an example, one could explain significance of the density $\hat{\mu}_d$ in equation (7) in terms of a kind of gauge transform, as follows:
> As the teacher sees its input $u$ through its one-dimensional projection $\theta_0^Tu$, the distribution of the teacher input is characterized by the variance of the projection $\theta_0^Tu$, which is proportional to $\rho$. Also, one can rotate the coordinates of the student input $v$ so that its elements are independent. Thus the joint input distribution is fully characterized under the teacher-student setting in this paper by the variances of the independent elements of the student input, which are proportional to $\bar{\theta}_i$s, and their covariances with the projection $\theta_0^Tu$, which are proportional to $\omega_i$s.

Such an explanation should be beneficial not only as it clarifies the meaning of these quantities but also as it would furthermore suggest how to extend the theoretical discussion in this paper to more general cases, for example, to the case with a committee machine as the teacher.

Some very minor points:
- Font styles in mathematical formulas are sometimes inconsistent. For example, vector quantities such as $x,u,v$ are in some occasions written in bold roman, whereas in other occasions they appear in bold italic.
- Equation (4): The argument of $f_0$ should be divided by $\sqrt{p}$.
- Line 51: In Theorem(s)
- Lines 52-53: training and generalisation error(s)
- Lines 152, 233: $x$ and $y$ should be italic.
- Line 157: the training (loss $\to$ error)
- Line 162: such as (e.g.)
- Line 188: kitchen sink(s) model
- Line 193: all-one(s) vector
- Line 196: random sink(s) models
- Line 241: that follow(s)
- Line 252: five-layer(s) Deep convolutional GAN
- Line 274: the covariance(s) matrices
- Line 283: that fit(s) the true labels
- Line 314: Fig(s). 4 and 1